# Convergence Analysis of Policy Gradient Methods with Dynamic Stochasticity

**Alessandro Montenegro** [1]   **Marco Mussi** [1]   **Matteo Papini** [1]   **Alberto Maria Metelli** [1]

## Abstract

*Policy gradient* (PG) methods are effective *reinforcement learning* (RL) approaches, particularly for continuous problems. While they optimize stochastic (hyper)policies via action- or parameter-space exploration, real-world applications often require deterministic policies. Existing PG convergence guarantees to deterministic policies assume a fixed stochasticity in the (hyper)policy, tuned according to the desired final suboptimality, whereas practitioners commonly use a dynamic stochasticity level. This work provides the theoretical foundations for this practice. We introduce `PES`, a phase-based method that reduces stochasticity via a deterministic schedule while running PG subroutines with fixed stochasticity in each phase. Under gradient domination assumptions, `PES` achieves last-iterate convergence to the optimal deterministic policy with a sample complexity of order $\widetilde{\mathcal{O}}(\epsilon^{-5})$. Additionally, we analyze the common practice, termed `SL-PG`, of jointly learning stochasticity (via an appropriate parameterization) and (hyper)policy parameters. We show that `SL-PG` also ensures last-iterate convergence with a rate $\widetilde{\mathcal{O}}(\epsilon^{-3})$, but to the optimal stochastic (hyper)policy only, requiring stronger assumptions compared to `PES`.

## 1. Introduction

Among *reinforcement learning* (RL, Sutton & Barto, 2018) approaches, *policy gradient* (PG, Deisenroth et al., 2013) methods achieved significant success in addressing real-world scenarios thanks to their ability to handle continuous state and action spaces (Peters & Schaal, 2006), resilience to sensor and actuator noise (Gravell et al., 2020), and robustness in partially-observable environments (Azizzadenesheli et al., 2018). Additionally, they enable the incorporation of expert knowledge in the policy design phase (Ghavamzadeh & Engel, 2006), improving the efficacy, safety, and interpretability of the learned policy (Peters & Schaal, 2008).

PG methods optimize directly over the parameter space of *parametric policies* in order to improve a performance function (e.g., the expected return). In RL, addressing the *exploration* problem is crucial. Agents must try different actions to gather information on long-term outcomes, rather than solely maximizing immediate rewards. In PGs, exploration is typically achieved by injecting noise into either the agent's actions or the policy parameters. These two exploration strategies are known as *action-based* (AB) and *parameter-based* (PB) exploration (Metelli et al., 2018), respectively. In particular, AB exploration, whose prototypical algorithms are REINFORCE (Williams, 1992) and GPOMDP (Baxter & Bartlett, 2001), keeps the exploration at the action level by leveraging *stochastic policies* (e.g., Gaussian). Instead, PB approaches, whose prototype is PGPE (Sehnke et al., 2010), explore at the parameter level via *stochastic hyperpolicies*, used to sample the parameters of an underlying (typically deterministic) policy.

From a theoretical perspective, significant work has focused on the convergence guarantees of PG methods, particularly for AB exploration (Zhao et al., 2011; Papini et al., 2018; Yuan et al., 2022; Fatkhullin et al., 2023; Bhandari & Russo, 2024). However, these methods produce parameters of *stochastic (hyper)policies*[1] that often fail to meet reliability, safety, and traceability requirements in real-world applications. The PG literature traditionally addressed learning *deterministic policies* through *deterministic policy gradient* algorithms (DPG, Silver et al., 2014), which inspired successful deep RL methods (e.g., DDPG, Lillicrap et al., 2016; Fujimoto et al., 2018). However, these approaches are inherently *off-policy* and rely on *actor-critic* architectures. These algorithmic complexities make their convergence guarantees difficult to establish and currently available under demanding assumptions only (Xiong et al., 2022).

Recently, Montenegro et al. (2024) introduced a general framework for assessing last-iterate global convergence to the optimal deterministic policy using stochastic (hyper)policies. Similarly to much of the PG literature, they

---

[1]Politecnico di Milano, Piazza Leonardo Da Vinci 32, 20133, Milan, Italy. Correspondence to: Alessandro Montenegro <alessandro.montenegro@polimi.it>.

*Proceedings of the 42nd International Conference on Machine Learning*, Vancouver, Canada. PMLR 267, 2025. Copyright 2025 by the author(s).

---

[1]The term (hyper)policy refers jointly to AB and PB explorations.

consider the simplest form of exploration that injects noise from a white noise distribution (e.g., zero-mean Gaussian) with a *static* stochasticity parameter $\sigma$ (i.e., fixed throughout learning). By setting $\sigma$ proportional to the desired suboptimality $\epsilon$, convergence to the optimal deterministic policy is achieved. However, this approach has the drawback of requiring to use a very small $\sigma$ *from the start* of learning. This limits the practicality of static stochasticity methods.

Static stochasticity conflicts with the common practice of *dynamically* adjusting stochasticity during learning. In AB exploration, deep RL methods (Schulman et al., 2015; Duan et al., 2016) typically address this by jointly optimizing the policy parameters and variance (via a suitable parameterization) through gradient ascent. Additionally, entropy regularization (Haarnoja et al., 2018; Ahmed et al., 2019) further promotes exploration by modifying the reward function to favor more stochastic policies. In PB exploration, Gaussian hyperpolicies are often used, with the variance learned alongside the hyperpolicy mean via gradient ascent (e.g., Wierstra et al., 2014; Likmeta et al., 2020).

Although dynamically adjusting stochasticity has been successful in practical RL applications, its theoretical understanding remains limited. This theory-practice gap raises the question: *Can PG methods guarantee convergence to the optimal deterministic policy when using dynamic stochasticity?* In this paper, we answer this question positively.

**Original Contribution.** In this work, we make a step towards the theoretical understanding of employing dynamic stochasticity in PGs. The main contributions are:

- In Section 3, we present PES (Phased Exploration Schedule), a phase-based algorithm reducing stochasticity through a *pre-determined schedule* across phases, in which a PG subroutine with fixed stochasticity is run. After outlining the convergence conditions for PG subroutines in Section 4, we discuss in Section 5 the impact of varying stochasticity on the performance index. Finally, in Section 6, we demonstrate that, under these conditions and a gradient domination assumption, PES achieves last-iterate global convergence to the optimal deterministic policy with an $\widetilde{\mathcal{O}}(\epsilon^{-5})$ sample complexity.
- In Section 7, we analyze the convergence for the common practice of jointly learning (hyper)policy parameters and stochasticity, referred to as SL-PG (Stochasticity-Learning Policy Gradient). We show that last-iterate global convergence is achievable with a sample complexity of $\widetilde{\mathcal{O}}(\epsilon^{-3})$, but to stochastic (hyper)policies only and requiring stronger assumptions than PES. We conclude by discussing the possibility of achieving convergence to *deterministic* policies at the same rate as PES.

In Section 9, we numerically validate the proposed algorithm. Related work is discussed in Section 8. The proofs of the results are deferred to the appendix.

## 2. White Noise Exploring Policy Gradients

**Notation.** For a measurable set $\mathcal{X}$, we denote with $\Delta(\mathcal{X})$ the set of probability measures over $\mathcal{X}$. For $P \in \Delta(\mathcal{X})$, we denote with $p$ its density function. For $n, m \in \mathbb{N}$ with $n \geqslant m$, we denote $[\![n]\!] := \{1, \ldots, n\}$ and $[\![m, n]\!] := \{m, m+1, \ldots, n\}$. For $x \in \mathbb{R}$, we denote $(x)^+ := \max\{0, x\}$.

**Lipschitz Continuous and Smooth Functions.** A function $f : \mathcal{X} \subseteq \mathbb{R}^d \to \mathbb{R}$ is $L_1$-*Lipschitz continuous* ($L_1$-LC) if $|f(\mathbf{x}) - f(\mathbf{x}')| \leqslant L_1 \|\mathbf{x} - \mathbf{x}'\|_2$ for every $\mathbf{x}, \mathbf{x}' \in \mathcal{X}$. Similarly, $f$ is $L_2$-*Lipschitz smooth* ($L_2$-LS) if it is continuously differentiable and its gradient $\nabla_\mathbf{x} f$ is $L_2$-LC, i.e., $\|\nabla_\mathbf{x} f(\mathbf{x}) - \nabla_\mathbf{x} f(\mathbf{x}')\|_2 \leqslant L_2 \|\mathbf{x} - \mathbf{x}'\|_2$ for every $\mathbf{x}, \mathbf{x}' \in \mathcal{X}$.

**Markov Decision Processes.** The environment is modeled as a Markov Decision Process (MDP, Puterman, 1990). An MDP is represented by $\mathcal{M} := (\mathcal{S}, \mathcal{A}, p, r, \rho_0, \gamma)$, where $\mathcal{S} \subseteq \mathbb{R}^{d_\mathcal{S}}$ and $\mathcal{A} \subseteq \mathbb{R}^{d_\mathcal{A}}$ are the measurable state and action spaces; $p : \mathcal{S} \times \mathcal{A} \to \Delta(\mathcal{S})$ is the transition model, where $p(\mathbf{s}'|\mathbf{s}, \mathbf{a})$ specifies the probability density of landing in state $\mathbf{s}' \in \mathcal{S}$ by playing action $\mathbf{a} \in \mathcal{A}$ in state $\mathbf{s} \in \mathcal{S}$; $r : \mathcal{S} \times \mathcal{A} \to [-R_{\max}, R_{\max}]$ is the reward function, where $r(\mathbf{s}, \mathbf{a})$ specifies the reward the agent gets when playing action $\mathbf{a}$ in state $\mathbf{s}$; $\rho_0 \in \Delta(\mathcal{S})$ is the initial-state distribution; $\gamma \in [0, 1]$ is the discount factor. A trajectory $\tau = (\mathbf{s}_{\tau,0}, \mathbf{a}_{\tau,0}, \ldots, \mathbf{s}_{\tau,T-1}, \mathbf{a}_{\tau,T-1})$ of length $T \in \mathbb{N} \cup \{+\infty\}$ is a sequence of $T$ state-action pairs. The *discounted return* of a trajectory $\tau$ is given by:

$$R(\tau) := \sum_{t=0}^{T-1} \gamma^t r(\mathbf{s}_{\tau,t}, \mathbf{a}_{\tau,t}).$$

We admit $\gamma = 1$ only when $T < +\infty$.

**Deterministic Parametric Policies.** We consider a *parametric deterministic policy* $\boldsymbol{\mu}_{\boldsymbol{\theta}} : \mathcal{S} \to \mathcal{A}$, where $\boldsymbol{\theta} \in \Theta \subseteq \mathbb{R}^{d_\Theta}$ is the parameter vector belonging to the parameter space $\Theta$. The performance of $\boldsymbol{\mu}_{\boldsymbol{\theta}}$ is assessed via the *expected return* $J_\mathrm{D} : \Theta \to \mathbb{R}$, defined as:

$$J_\mathrm{D}(\boldsymbol{\theta}) := \mathop{\mathbb{E}}_{\tau \sim p_\mathrm{D}(\cdot|\boldsymbol{\theta})} [R(\tau)],$$

where $p_\mathrm{D}(\tau; \boldsymbol{\theta}) := \rho_0(\mathbf{s}_{\tau,0}) \prod_{t=0}^{T-1} p(\mathbf{s}_{\tau,t+1}|\mathbf{s}_{\tau,t}, \boldsymbol{\mu}_{\boldsymbol{\theta}}(\mathbf{s}_{\tau,t}))$ is the density of trajectory $\tau$ induced by policy $\boldsymbol{\mu}_{\boldsymbol{\theta}}$. The agent's goal consists of finding an optimal parameter $\boldsymbol{\theta}_\mathrm{D}^* \in \arg\max_{\boldsymbol{\theta} \in \Theta} J_\mathrm{D}(\boldsymbol{\theta})$ and we denote $J_\mathrm{D}^* := J_\mathrm{D}(\boldsymbol{\theta}_\mathrm{D}^*)$.

**White Noise Exploration.** Similarly to Montenegro et al. (2024), we introduce the notion of white noise that we will employ to define white-noise-based exploration.

**Definition 2.1** (White Noise). *Let $d \in \mathbb{N}$. A family of probability distributions $\{\Phi_{d,\sigma}\}_{\sigma \in \mathbb{R}_{>0}} \subset \Delta(\mathbb{R}^d)$ is a white noise if for every $\sigma \in \mathbb{R}_{>0}$:*

$$\mathop{\mathbb{E}}_{\boldsymbol{\epsilon} \sim \Phi_{d,\sigma}} [\boldsymbol{\epsilon}] = \mathbf{0}_d \quad and \quad \mathop{\mathbb{E}}_{\boldsymbol{\epsilon} \sim \Phi_{d,\sigma}} [\|\boldsymbol{\epsilon}\|_2^2] \leqslant d\sigma^2.$$

Unlike Montenegro et al. (2024), which assumes static

stochasticity, we consider *families* of distributions parameterized by $\sigma$, referred to as the *stochasticity amount*. Definition 2.1 enables the analysis of algorithms that dynamically vary $\sigma$ during learning. This includes, for example, zero-mean Gaussian distributions $\epsilon \sim \mathcal{N}(\mathbf{0}_d, \sigma\Lambda)$ with $\lambda_{\max}(\Lambda) \leqslant 1$, where $\mathbb{E}[\|\epsilon\|_2^2] = \sigma^2 \operatorname{tr}(\Lambda) \leqslant d\sigma^2$. This noise is assumed to be *white* across exploration steps.

**Action-Based (AB) Exploration.** In AB exploration, we consider a *parametric stochastic policy* $\pi_{\theta,\sigma} : \mathcal{S} \to \Delta(\mathcal{A})$ built upon an underlying deterministic policy $\mu_{\theta}$, by perturbing each action suggested by $\mu_{\theta}$ with a white-noise random vector. The policy is used to sample actions $\mathbf{a}_t \sim \pi_{\theta,\sigma}(\cdot|\mathbf{s}_t)$ played in state $\mathbf{s}_t$ for *every step* $t$ of interaction. Formally, we consider the following definition of *white noise policies*.

**Definition 2.2** (White Noise Policies). *Let $\theta \in \Theta$, $\sigma \in \mathbb{R}_{>0}$, and $\mu_{\theta} : \mathcal{S} \to \mathcal{A}$ be a parametric deterministic policy and let $\Phi_{d_{\mathcal{A}},\sigma}$ be a white noise distribution (Def. 2.1). A white noise policy $\pi_{\theta,\sigma} : \mathcal{S} \to \Delta(\mathcal{A})$ is such that, for every state $s \in \mathcal{S}$, the action $a \sim \pi_{\theta,\sigma}(\cdot|s)$ satisfies $a = \mu_{\theta}(s) + \epsilon$, where $\epsilon \sim \Phi_{d_{\mathcal{A}},\sigma}$ which is sampled independently at every step (i.e., whenever an action is sampled).*

This definition justifies the name for AB exploration since the exploration is carried out at the action level. From now on, we refer to this kind of stochastic policies, calling $\sigma$ the *stochasticity amount* of $\pi_{\theta,\sigma}$.

The performance of $\pi_{\theta,\sigma}$ is assessed via the *expected return* $J_{\mathrm{A}} : \Theta \times \mathbb{R}_{>0} \to \mathbb{R}$, defined as:

$$J_{\mathrm{A}}(\theta, \sigma) := \mathop{\mathbb{E}}_{\tau \sim p_{\mathrm{A}}(\cdot|\theta)} [R(\tau)],$$

being $p_{\mathrm{A}}(\tau; \theta)$ the density of trajectory $\tau$ induced by $\pi_{\theta,\sigma}$.

In AB exploration, whenever $\sigma > 0$, we aim to learn $\theta_{\mathrm{A}}^*(\sigma) \in \arg\max_{\theta \in \Theta} J_{\mathrm{A}}(\theta, \sigma)$. Given a $J_{\mathrm{A}}(\theta, \sigma)$ differentiable w.r.t. $\theta$, PG methods (Peters & Schaal, 2008) update the parameter $\theta$ via gradient ascent: $\theta_{t+1} \leftarrow \theta_t + \zeta_t \widehat{\nabla}_{\theta} J_{\mathrm{A}}(\theta_t, \sigma)$, where $\zeta_t > 0$ is the *step size* and $\widehat{\nabla}_{\theta} J_{\mathrm{A}}(\theta, \sigma)$ is an estimator of $\nabla_{\theta} J_{\mathrm{A}}(\theta, \sigma)$. In particular, we consider the GPOMDP *estimator* (Baxter & Bartlett, 2001) which employs $N$ independent trajectories $\{\tau_i\}_{i=1}^N$ collected with policy $\pi_{\theta,\sigma}$ (i.e., $\tau_i \sim p_{\mathrm{A}}(\cdot; \theta)$), where $N$ is called *batch size*. In this paper, we just consider GPOMDP, since REINFORCE, which is conceptually similar, suffers from larger variance (Williams, 1992; Baxter & Bartlett, 2001). More details on AB exploration are presented in Appendix B.

**Parameter-Based (PB) Exploration.** In PB exploration, we use a *parametric stochastic hyperpolicy* $\nu_{\theta,\sigma} \subseteq \Delta(\Theta)$ built upon an underlying deterministic policy $\mu_{\theta}$, by perturbing the parameter vector $\theta$ with a white-noise random vector. The hyperpolicy is used to sample parameters $\theta' \sim \nu_{\theta,\sigma}$ to be plugged in the deterministic policy $\mu_{\theta'}$ at the beginning

of *every trajectory*. Formally, we consider the following definition of *white noise hyperpolicies*.

**Definition 2.3** (White Noise Hyperpolicies). *Let $\theta \in \Theta$, $\sigma \in \mathbb{R}_{>0}$, and $\mu_{\theta} : \mathcal{S} \to \mathcal{A}$ be a parametric deterministic policy and let $\Phi_{d_{\Theta},\sigma}$ be a white-noise distribution (Def. 2.1). A white noise hyperpolicy $\nu_{\theta,\sigma} \in \Delta(\Theta)$ is such that, for every parameterization $\theta \in \Theta$, the parameterization $\theta' \sim \nu_{\theta,\sigma}$ satisfies $\theta' = \theta + \epsilon$, where $\epsilon \sim \Phi_{d_{\Theta},\sigma}$ which is sampled independently for every trajectory.*

This definition justifies the name for PB exploration, since the exploration is carried out at the parameter level. Note that, differently from AB exploration, in PB one, before starting to collect each trajectory, the current parametrization for the deterministic policy $\theta$ is perturbed with a noise vector $\epsilon \sim \Phi_{d_{\Theta},\sigma}$, then the deterministic policy $\mu_{\theta+\epsilon}$ is used for the entire trajectory. From now on, we focus on this kind of stochastic hyperpolicies, calling $\sigma$ the *stochasticity amount* for $\nu_{\theta,\sigma}$.

The performance index of $\nu_{\theta,\sigma}$ is $J_{\mathrm{P}} : \Theta \times \mathbb{R}_{>0} \to \mathbb{R}$, that is the expectation over $\theta'$ of $J_{\mathrm{D}}(\theta')$ defined as:

$$J_{\mathrm{P}}(\theta, \sigma) := \mathop{\mathbb{E}}_{\theta' \sim \nu_{\theta,\sigma}} [J_{\mathrm{D}}(\theta')].$$

Whenever $\sigma > 0$, PB exploration aims at learning $\theta_{\mathrm{P}}^*(\sigma) \in \arg\max_{\theta \in \Theta} J_{\mathrm{P}}(\theta, \sigma)$. If $J_{\mathrm{P}}(\theta, \sigma)$ is differentiable w.r.t. $\theta$, PGPE (Sehnke et al., 2010) updates the hyperparameter $\theta$ by gradient ascent: $\theta_{t+1} \leftarrow \theta_t + \zeta_t \widehat{\nabla}_{\theta} J_{\mathrm{P}}(\theta_t, \sigma)$. PGPE uses an estimator of $\nabla_{\theta} J_{\mathrm{P}}(\theta, \sigma)$ which employs $N$ independent parameter-trajectory pairs $\{(\theta_i, \tau_i)\}_{i=1}^N$, collected with hyperpolicy $\nu_{\theta,\sigma}$, that is, $\theta_i \sim \nu_{\theta,\sigma}$ and $\tau_i \sim p_{\mathrm{D}}(\cdot; \theta_i)$. Also in this case, $N$ is called *batch size*. More details on PB exploration are presented in Appendix B.

**Exploration-Agnostic Framework.** In this work, we consider both AB and PB explorations and present the results using a unified notation. We define $J_{\dagger}$ with $\dagger \in \{\mathrm{P}, \mathrm{A}\}$ as the exploration-agnostic objective and $\theta^*(\sigma) \in \arg\max_{\theta \in \Theta} J_{\dagger}(\theta, \sigma)$ as the corresponding optimal parameterization for a given stochasticity amount $\sigma$. This notation is also applied to other quantities, mapped to their respective exploration (either AB or PB) in Appendix C.

## 3. PES: Algorithm Description

In this section, we present Phased Exploration Schedule (PES) a phase-based algorithm with PG subroutines which aims to output a deterministic policy. The algorithm, whose pseudo-code is presented in Algorithm 1, in each phase $p \in [\![0, P-1]\!]$ runs a **PB** or an **AB** policy gradient method (i.e., a PG subroutine) with a fixed stochasticity amount $\sigma_p$ to learn in $K_p$ iterations a parameterization $\theta_p$ starting from the output of the previous phase, $\theta_{p-1}$.

When starting a new phase, PES updates the stochasticity

**Algorithm 1** PES.

---

**Input** : Number of phases $P$, Iterations per phase $(K_i)_{i=1}^{P}$,
Initial parameter $\bar{\boldsymbol{\theta}}$, Stochasticity schedule $(\sigma_i)_{i=1}^{P}$,
Learning rate schedule $(\zeta_i)_{i=1}^{P}$, Batch size $N$

Initialize $\boldsymbol{\theta}_0 \leftarrow \bar{\boldsymbol{\theta}}$
**for** $p \in [\![P]\!]$ **do**
$\quad$ $\boldsymbol{\theta}_p \leftarrow$ Run for $K_p$ iterations a **PB** or **AB** PG from $\boldsymbol{\theta}_{p-1}$, with
$\quad$ fixed stochasticity $\sigma_p$, learning rate $\zeta_p$, batch size $N$
**end**
Return $\boldsymbol{\theta}_P$

---

according to a *decreasing deterministic schedule*. This allows the algorithm to begin with high stochasticity $\sigma_{\max}$, aiding in the discovery of a good parameterization $\boldsymbol{\theta}_0$, and then gradually reduces $\sigma$. As we will show, this process ensures that the final parameterization $\boldsymbol{\theta}_{P-1}$ is nearly optimal for the final stochasticity $\sigma_{P-1}$, which can be set to the desired suboptimality $\epsilon$ for deterministic policy deployment. This approach avoids the issues of using a fixed stochasticity of $\epsilon$, as discussed in Section 1.

In particular, PES employs the following deterministic schedule to update the stochasticity:

$$\sigma_p = \sigma_{\max}(p+1)^{-y},$$

with $\sigma_{\max}, y \in \mathbb{R}_{>0}$, and $p \in [\![0, P-1]\!]$. As we shall discuss later, this choice, besides fulfilling the previously mentioned desiderata, enables last-iterate convergence guarantees to the optimal deterministic policy.

**Theoretical Motivation.** From a theoretical perspective, since each PG subroutine is run with a fixed exploration amount $\sigma_p$, under appropriate assumptions, following (Montenegro et al., 2024), it is possible to assess the sample complexity $NK_p$ (i.e., the number of trajectories needed for phase $p$) to ensure a global last-iterate convergence of $J_\dagger(\cdot, \sigma_p)$ to a desired accuracy $\epsilon$:

$$J_\dagger(\boldsymbol{\theta}^*(\sigma_p), \sigma_p) - \mathbb{E}[J_\dagger(\boldsymbol{\theta}_p, \sigma_p)] \leqslant \epsilon,$$

recalling that the optimal (hyper)policy parameterization $\boldsymbol{\theta}^*(\sigma)$ depends on the stochasticity itself. As we will show, by $(i)$ using the PES deterministic schedule for stochasticity $\sigma_p$ and $(ii)$ satisfying the conditions for last-iterate convergence of each PG subroutine, it suffices to determine the number of phases $P$ to ensure convergence to a deterministic policy. Before presenting PES' last-iterate convergence guarantees to the optimal deterministic policy (Sec. 6), we first outline the assumptions required for state-of-the-art last-iterate convergence of PG subroutines (Sec. 4) and then examine the impact of reducing stochasticity on $J_\dagger$ (Sec. 5).

## 4. PG Subroutines Convergence

In this section, we recall the assumptions needed to guarantee state-of-the-art last-iterate convergence results for each PG subroutine of PES. A more detailed discussion is presented in Appendix D.

**Assumptions for PG Subroutines Convergence.** Montenegro et al. (2024) show that to ensure last-iterate global convergence of a PG subroutine with a fixed stochasticity $\sigma_p$, the objective $J_\dagger(\cdot, \sigma_p)$ has to fulfill three properties: $(i)$ smoothness of $J_\dagger(\cdot, \sigma_p)$ w.r.t. $\boldsymbol{\theta}$, $(ii)$ *weak gradient domination* (WGD) on $J_\dagger(\cdot, \sigma_p)$ w.r.t. $\boldsymbol{\theta}$, and $(iii)$ bounded variance of the employed estimator. This set of assumptions is standard in the PG convergence literature (Papini et al., 2018; Agarwal et al., 2021; Yuan et al., 2022; Fatkhullin et al., 2023; Bhandari & Russo, 2024). In our dynamic exploration scenario, we need such conditions to hold for every $\sigma_p$. Condition $(i)$ holds for both $J_P$ and $J_A$ for every exploration amount $\sigma_p$ under the following assumption.

**Assumption 4.1** ($J_D$ is $L_2$-LS w.r.t. $\boldsymbol{\theta}$). *There exists $L_2 \in \mathbb{R}_{\geqslant 0}$ such that for every $\boldsymbol{\theta} \in \mathbb{R}^{d_\Theta}$, the following holds:*

$$\|\nabla_{\boldsymbol{\theta}}^2 J_D(\boldsymbol{\theta})\|_2 \leqslant L_2.$$

Indeed, the smoothness of $J_D$ w.r.t. the parameter $\boldsymbol{\theta}$ is inherited by the stochastic objectives.

**Lemma 4.1** ($J_\dagger$ Inherited Smoothness). *Under Assumption 4.1, for every $\boldsymbol{\theta} \in \Theta$ and $\sigma \in \mathbb{R}_{\geqslant 0}$, it holds:*

$$\|\nabla_{\boldsymbol{\theta}}^2 J_\dagger(\boldsymbol{\theta}, \sigma)\|_2 \leqslant L_2.$$

Condition $(ii)$ can be inherited by both $J_P$ and $J_A$ under the assumption that WGD holds for $J_D$.

**Assumption 4.2** (WGD on $J_D$). *There exist $\alpha_D > 0$ and $\beta_D \geqslant 0$ such that, for every $\boldsymbol{\theta} \in \Theta$, the following holds:*

$$J_D^* - J_D(\boldsymbol{\theta}) \leqslant \alpha_D \|\nabla_{\boldsymbol{\theta}} J_D(\boldsymbol{\theta})\|_2 + \beta_D.$$

We highlight that, if $\beta_D = 0$, then we are in the presence of the so-called *strong gradient domination*, meaning that the objective $J_D$ has no local optima. When $\beta_D > 0$, WGD holds and $J_D$ may admit local optima. Before showing the WGD inheritance, we need to introduce a further assumption on the MDP and $\boldsymbol{\mu}_{\boldsymbol{\theta}}$ regularity, needed for the **AB** case.

**Assumption 4.3** (MDP and $\boldsymbol{\mu}_{\boldsymbol{\theta}}$ Regularity). *Let $\boldsymbol{s}, \boldsymbol{s}' \in \mathcal{S}$. The log transition model $\log p(\boldsymbol{s}'|\boldsymbol{s}, \cdot)$ is $L_p$-LC and $L_{2,p}$-LS, and the reward function $r(\boldsymbol{s}, \cdot)$ is $L_r$-LC and $L_{2,r}$-LS w.r.t. the action. Moreover, the deterministic policy $\boldsymbol{\mu}_{\boldsymbol{\theta}}(\boldsymbol{s})$ is $L_\mu$-LC and $L_{2,\mu}$-LS w.r.t. its parameters.*

Further details for this assumption are provided in Appendix D. Next, we show the inheritance of WGD properties by $J_A$ and $J_P$.

**Theorem 4.2** (Inherited WGD for $J_\dagger$, Montenegro et al. 2024). *Considering a (hyper)policy complying with Definitions 2.2 or 2.3, under Assumptions 4.2 and 4.3, for any $\sigma \in \mathbb{R}_{>0}$ and $\boldsymbol{\theta} \in \Theta$, the following holds:*

$$J_\dagger(\boldsymbol{\theta}^*(\sigma), \sigma) - J_\dagger(\boldsymbol{\theta}, \sigma) \leqslant \alpha_D \|\nabla_{\boldsymbol{\theta}} J_\dagger(\boldsymbol{\theta}, \sigma)\|_2 + \beta_\dagger(\sigma),$$

*with $\beta_\dagger(\sigma) := \beta_D + \mathcal{W}_\dagger \sigma$, for some $\mathcal{W}_\dagger \geqslant 0$.*

Notice that the multiplier $\alpha_D$ of the norm of the gradient is

that of Assumption 4.2, whereas the term $\beta_\dagger(\sigma)$ acquires a dependence on the exploration amount $\sigma$.

**Remark 4.1** (When does WGD hold?). *We remark that the WGD property depends on both the policy parameterization class and the environment. For instance, Bhandari & Russo (2024) show that, for AB exploration in tabular environments and using natural policy parameterizations, the Polyak-Łojasiewicz (PL) condition holds. This is a stronger condition than WGD, requiring $J_A(\boldsymbol{\theta}^*(\sigma), \sigma) - J_A(\boldsymbol{\theta}, \sigma) \leqslant \alpha \|\nabla_{\boldsymbol{\theta}} J_A(\boldsymbol{\theta}, \sigma)\|_2^2$ for some $\alpha \in \mathbb{R}_{>0}$. Moreover, Mei et al. (2020) show that, for AB exploration in tabular environments and using a softmax policy parameterization, i.e., $\pi_{\boldsymbol{\theta}, \sigma}(\mathbf{a} \mid \mathbf{s}) \propto \exp(\boldsymbol{\theta}(\mathbf{s}, \mathbf{a}))$, the WGD property holds for $J_A$ with $\beta_A = 0$. Finally, Ding et al. (2022) prove a more general result, still in the context of AB exploration, showing that WGD holds whenever: $(i)$ the Fisher information matrix induced by the policy $\pi_{\boldsymbol{\theta}, \sigma}$ is non-degenerate for every $\boldsymbol{\theta} \in \Theta$, i.e., $\boldsymbol{F}(\boldsymbol{\theta}, \sigma) = \mathbb{E}_{\pi_{\boldsymbol{\theta}, \sigma}}[\nabla_{\boldsymbol{\theta}} \log \pi_{\boldsymbol{\theta}, \sigma}(\mathbf{a} \mid \mathbf{s}) \nabla_{\boldsymbol{\theta}} \log \pi_{\boldsymbol{\theta}, \sigma}(\mathbf{a} \mid \mathbf{s})^\top] \geq \mu_F \boldsymbol{I}$ for some $\mu_F > 0$; and $(ii)$ a "compatible function approximation bias" is upper bounded by $\epsilon_{bias}$. Under these conditions, WGD holds on $J_A$ with constants $\alpha_A = G \mu_F^{-1}$ and $\beta_A = (1 - \gamma^{-1}) \sqrt{\epsilon_{bias}}$, where $G$ is such that $\|\nabla_{\boldsymbol{\theta}} \log \pi_{\boldsymbol{\theta}}(\mathbf{a} \mid \mathbf{s})\|_2 \leqslant G$. Finding a hyperpolicy class that induces WGD on $J_P$, in the sense of (Ding et al., 2022), remains an open problem.*

Finally, condition $(iii)$, requiring the variance of the estimator $\widehat{\nabla}_{\boldsymbol{\theta}} J_\dagger(\boldsymbol{\theta}, \sigma)$ to be bounded, is granted by the following assumption on the scores of the stochastic (hyper)policy.

**Assumption 4.4** (Bounded Scores). *Let $\Phi_{d, \sigma} \in \Delta(\mathbb{R}^d)$ be a white-noise distribution complying with Definition 2.1 with $\sigma \in \mathbb{R}_{>0}$ and density $\phi_{d, \sigma}$. $\phi_{d, \sigma}$ is differentiable in its argument and there exists a constant $c > 0$ such that:*

$$(i) \quad \mathbb{E}_{\boldsymbol{\epsilon} \sim \Phi_{s, \sigma}} [\|\nabla_{\boldsymbol{\epsilon}} \log \phi_{d, \sigma}(\boldsymbol{\epsilon})\|_2^2] \leqslant cd\sigma^{-2},$$

$$(ii) \quad \mathbb{E}_{\boldsymbol{\epsilon} \sim \Phi_{s, \sigma}} [\|\nabla_{\boldsymbol{\epsilon}}^2 \log \phi_{d, \sigma}(\boldsymbol{\epsilon})\|_2] \leqslant c\sigma^{-2}.$$

As an example, this assumption is fulfilled by zero-mean Gaussian noise $\boldsymbol{\epsilon} \sim \mathcal{N}(\mathbf{0}_d, \sigma\boldsymbol{\Lambda})$. Under such an assumption, the variance of the estimator $\widehat{\nabla}_{\boldsymbol{\theta}} J_\dagger(\boldsymbol{\theta}, \sigma)$ is bounded.

**Lemma 4.3** (Bounded Estimator Variance). *Considering a (hyper)policy complying with Definitions 2.2 (PB) or 2.3 (AB), under Assumptions 4.3 and 4.4 the following holds:*

$$\mathbb{V}\mathrm{ar}[\widehat{\nabla}_{\boldsymbol{\theta}} J_\dagger(\boldsymbol{\theta}, \sigma)] \leqslant \frac{\mathcal{V}_{\dagger, \boldsymbol{\theta}}}{N\sigma^2}, \quad \text{for some } \mathcal{V}_{\dagger, \boldsymbol{\theta}} \geqslant 0.$$

**PG Subroutines Convergence.** Under the assumptions discussed so far, a PG subroutine with a fixed stochasticity $\sigma_p$ enjoys the following convergence guarantee.

**Theorem 4.4** (PG Global Last-Iterate Convergence, Montenegro et al. 2024). *Consider running for $K_p$ iterations a PG algorithm with a (hyper)policy satisfying Definitions 2.2 or 2.3 with an exploration amount $\sigma_p \in \mathbb{R}_{>0}$. Under As-sumptions 4.1, 4.2, 4.3, and 4.4, by selecting an appropriate constant step size $\zeta_p$ and a sample complexity $NK_p$ satisfying:*

$$NK_p \geqslant \widetilde{\mathcal{O}}\left(\frac{\alpha_D^4 L_2 \mathcal{V}_{\dagger, \boldsymbol{\theta}}}{\epsilon^3 \sigma_p^2}\right),$$

*then $J_\dagger(\boldsymbol{\theta}^*(\sigma_p), \sigma_p) - \mathbb{E}[J_\dagger(\boldsymbol{\theta}_p, \sigma_p)] \leqslant \epsilon + \beta_\dagger(\sigma_p)$, where the notation $\widetilde{\mathcal{O}}(\cdot)$ hides logarithmic terms.*

In Section 6, we will use the assumptions and the results presented in this section to ensure the convergence of each PG subroutine, and study the convergence of the whole phased process of PES to the optimal deterministic policy.

## 5. The Effects of Dynamic Stochasticity

In this section, we discuss the effects on $J_\dagger$ when varying the stochasticity $\sigma$. Specifically, we seek to understand under which assumptions a small variation of the stochasticity $\sigma$ attains a small variation of the expected return $J_\dagger(\cdot, \sigma)$. Intuitively, this is a minimal requirement for the design of algorithms like PES aiming to *dynamically* change the stochasticity during the learning process.

As highlighted in Section 2, we recall that the optimal (hyper)policy parameterization $\boldsymbol{\theta}^*(\sigma)$ depends on the stochasticity itself, i.e., $\boldsymbol{\theta}^*(\sigma) \in \arg\max_{\boldsymbol{\theta} \in \Theta} J_\dagger(\boldsymbol{\theta}, \sigma)$.

**Assumptions.** Before presenting the results, we introduce regularity assumptions on the deterministic objective $J_D$. The following assumption, needed in the analysis of the PB case, requires the $J_D$ to be LC in the policy parameters.

**Assumption 5.1** ($J_D$ is $L_J$-Lipschitz w.r.t. $\boldsymbol{\theta}$). *$J_D$ is $L_J$-Lipschitz w.r.t. parameterization $\boldsymbol{\theta}$, i.e., for every $\boldsymbol{\theta}, \boldsymbol{\theta}' \in \Theta$:*

$$|J_D(\boldsymbol{\theta}) - J_D(\boldsymbol{\theta}')| \leqslant L_J \|\boldsymbol{\theta} - \boldsymbol{\theta}'\|_2.$$

For analyzing the AB case, we need a similar assumption on the *non-stationary* (NS) deterministic objective. Let $\underline{\boldsymbol{\epsilon}} = (\boldsymbol{\epsilon}_t)_{t=0}^{T-1} \sim \Phi_{d_{\mathcal{A}}, \sigma}^T$ be a sequence of independently sampled noise vectors. Let $\underline{\boldsymbol{\mu}} = (\boldsymbol{\mu})_{t=0}^{T-1}$ be a NS deterministic policy, where, at time step $t$ the deterministic policy $\boldsymbol{\mu}_t : \mathcal{S} \to \mathcal{A}$ is played, with $\boldsymbol{\mu}_t = \boldsymbol{\mu}_{\boldsymbol{\theta}} + \boldsymbol{\epsilon}_t$. The objective for this kind of policy is $J_D(\underline{\boldsymbol{\mu}}) = \mathbb{E}_{\tau \sim p_D(\cdot | \underline{\boldsymbol{\mu}})}[R(\tau)]$, where $p_D(\tau | \underline{\boldsymbol{\mu}})$ is the density of a trajectory $\tau$ induced by the NS policy.

**Assumption 5.2** ($J_D$ is $L_{1, \underline{\boldsymbol{\mu}}}$-Lipschitz w.r.t. $\underline{\boldsymbol{\mu}}$). *The performance $J_D$ of the NS deterministic policy $\underline{\boldsymbol{\mu}}$ is $(L_t)_{t=0}^{T-1}$-Lipschitz w.r.t. the non stationary policy, i.e., for every $\underline{\boldsymbol{\mu}}, \underline{\boldsymbol{\mu}}'$:*

$$|J_D(\underline{\boldsymbol{\mu}}) - J_D(\underline{\boldsymbol{\mu}}')| \leqslant \sum_{t=0}^{T} L_t \sup_{\boldsymbol{s} \in \mathcal{S}} \|\boldsymbol{\mu}_t(\boldsymbol{s}) - \boldsymbol{\mu}'_t(\boldsymbol{s})\|_2.$$

*Moreover, we denote $L_{1, \underline{\boldsymbol{\mu}}} = \sum_{t=0}^{T-1} L_t$.*

In the following, we consider (hyper)policies complying with Definitions 2.2 (PB) or 2.3 (AB). Unfortunately, this,

combined with the assumptions presented above, is not enough to ensure the desired regularity, as shown in Example 5.1.

**Example 5.1.** *Consider a one-state MDP with $\mathcal{A} = [-1, 1]$, $r(a) = \max\{0, a\}$, $T = 1$, and constant deterministic policy, $\mu = 0$, fulfilling Assumptions 5.1 and 5.2. We consider the following white noise, fulfilling Definition 2.1:*

$$\Phi_{1,\sigma} = \begin{cases} \frac{1}{2}\delta_{-\sigma} + \frac{1}{2}\delta_\sigma & \text{if } \sigma < \overline{\sigma} \\ \text{Uniform}\left([-\sqrt{3}\sigma, \sqrt{3}\sigma]\right) & \text{otherwise} \end{cases},$$

*where $\overline{\sigma} > 0$. We compute the expected return, which is just the expected reward, in our case:*

$$J(\sigma) = \begin{cases} \frac{\sigma}{2} & \text{if } \sigma < \overline{\sigma} \\ \frac{\sqrt{3}}{4}\sigma & \text{otherwise} \end{cases}.$$

*We immediately observe that $J(\sigma)$ is discontinuous in the point $\sigma = \overline{\sigma}$.*

Thus, we introduce the following assumption that ensures a form of regularity in the way the noise is generated.

**Assumption 5.3** (Scale-invariant Noise). *Let $\{\Phi_{d,\sigma}\}_{\sigma>0}$ be a family of white noise distributions (Def. 2.1). For every $\sigma \in \mathbb{R}_{>0}$, let $\epsilon \sim \Phi_{d,\sigma}$ and $\overline{\epsilon} \sim \Phi_{d,1}$. It holds that: $\epsilon \overset{D}{=} \sigma\overline{\epsilon}$, where $\overset{D}{=}$ denotes equality in distribution.*

Assumption 5.3 establishes that we can generate the noise random variable $\epsilon$ associated with a certain stochasticity $\sigma$ by generating a noise random variable $\overline{\epsilon}$ associated with a conventional stochasticity (e.g., 1, but other choices are possible) and multiplying the variable by $\sigma$. This assumption is fulfilled, for instance, by zero-mean Gaussian distributions $\mathcal{N}(\mathbf{0}_d, \sigma^2\Lambda)$. In particular, under Assumption 5.3, for any stochasticity $\sigma$ and function $g$, it holds that $\mathbb{E}_{\epsilon\sim\Phi_{d,\sigma}}[g(\epsilon)] = \mathbb{E}_{\overline{\epsilon}\sim\Phi_{d,1}}[g(\sigma\overline{\epsilon})]$. This represents the key observation to prove that the expected return $J_\dagger(\cdot, \sigma)$ is LC w.r.t. the stochasticity $\sigma$, and subsequent results on the optimal performance, as illustrated in the following theorem.

**Theorem 5.1.** *If the (hyper)policy complies with Definitions 2.2 (**PB**) or 2.3 (**AB**), under Assumptions 5.1 (**PB**) or 5.2 (**AB**), and Assumption 5.3, for every $\theta \in \Theta$ and $\sigma_1, \sigma_2 \in \mathbb{R}_{\geqslant 0}$, it holds that:*

*(i)* $|J_\dagger(\theta, \sigma_1) - J_\dagger(\theta, \sigma_2)| \leqslant D_\dagger|\sigma_1 - \sigma_2|$

*(ii)* $J_\dagger^*(\sigma_1) - J_\dagger(\theta^*(\sigma_2), \sigma_1) \leqslant 2D_\dagger|\sigma_1 - \sigma_2|$

*(iii)* $|J_\dagger(\theta^*(\sigma_1), \sigma_1) - J_\dagger(\theta^*(\sigma_2), \sigma_2)| \leqslant D_\dagger|\sigma_1 - \sigma_2|$,

*where $\dagger \in \{P, A\}$, $D_A = L_{1,\underline{\mu}}\sqrt{d_\mathcal{A}}$, and $D_P = L_J\sqrt{d_\Theta}$.*

Theorem 5.1 will be crucial for establishing the convergence guarantees of PES. Indeed, besides bounding the loss when varying the stochasticity $\sigma$ while keeping the same parameterization $\theta$ (point *(i)*), it also quantifies the distance in expected return between optima under different stochastic-ity amounts (points *(ii)-(iii)*).

# 6. `PES`: Convergence

In this section, we demonstrate that PES achieves last-iterate convergence guarantees to the globally optimal deterministic policy. Specifically, we require that each PG subroutine $p \in [\![0, P-1]\!]$, run under a stochasticity $\sigma_p$, exhibits last-iterate convergence to the optimal (hyper)policy $\theta_p$ with a suboptimality $\epsilon$, i.e., $J_\dagger(\theta^*(\sigma_p), \sigma_p) - \mathbb{E}[J_\dagger(\theta_p, \sigma_p)] \leqslant \epsilon$.

Given the decreasing deterministic schedule employed by PES, if these guarantees are satisfied, the algorithm ensures that the next phase is initialized with an $\epsilon$-optimal parameterization from the previous phase. This allows us to leverage the results of Theorem 5.1 to control the performance loss when switching from phase $p-1$ to phase $p$, i.e., $J_\dagger(\theta^*(\sigma_p), \sigma_p) - J_\dagger(\theta_{p-1}, \sigma_p)$.

Given the convergence of PG subroutines at each phase (Theorem 4.4) and the regularity of the objective $J_\dagger$ with respect to stochasticity (Theorem 5.1), we just need to carefully select the number of phases $P$ to establish the last-iterate convergence of PES to the optimal deterministic policy.

**Theorem 6.1** (PES Convergence). *Employing a (hyper)policy complying with Definitions 2.2 (**PB**) or 2.3 (**AB**), under Assumptions 5.1 (**PB**) or 5.2 (**AB**), 5.3, and under the assumptions for the convergence of PG subroutines (Sec. 4), if PES is run for $P = (\epsilon/\sigma_{\max})^{-1/y}$ phases, the output parameterization $\theta_{P-1}$ is such that:*

$$J_D^* - \mathbb{E}[J_D(\theta_{P-1})] \leqslant (1 + 2D_\dagger + \mathcal{W}_\dagger)\epsilon + \beta_D,$$

*with a sample complexity:*

$$NK \geqslant \widetilde{\mathcal{O}}\left(\frac{L_2\alpha_D^2\mathcal{V}_{\dagger,\theta}(2D_\dagger + \mathcal{W}_\dagger)}{\epsilon^5}\right),$$

*where $K = \sum_{p=0}^{P-1} K_p$.*

Notice that, by setting the number of phases to $P = (\epsilon/\sigma_{\max})^{-1/y}$, we set the final stochasticity $\sigma_{P-1} = \epsilon$. In Theorem 6.1, we recover the $\widetilde{\mathcal{O}}(\epsilon^{-5})$ rate of Montenegro et al. (2024) for last-iterate convergence to the optimal deterministic policy, with a fixed stochasticity amount $\sigma = \epsilon$. As highlighted in Section 1, keeping a static stochasticity $\sigma$ as small as $\epsilon$ has practical limitations. Instead, PES takes advantage of scaling the exploration amount $\sigma_p$ at each phase and starting from a good parameterization for the previous exploration amount $\sigma_{p-1}$. These benefits will be further discussed in Section 9.

Moreover, the convergence results of PES highlight a trade-off between the parameters $\sigma_{\max}$ and $y$. Specifically, $\sigma_{\max}$ controls the stochasticity of the first phase and the number of phases required to output the optimal deterministic policy. A large value of $\sigma_{\max}$ makes PES to learn starting from a highly stochastic objective, requiring more phases to con-

verge, while a small $\sigma_{\max}$ needs less phases to converge, but limits the range of the stochasticity. Furthermore, $y$ determines the number of phases and the smoothness of the schedule. A small value of $y$ results in a large number of phases $P$, making the reduction of stochasticity smoother.

Finally, by plugging the constants for PB († = P) or AB († = A) exploration into the results of Theorem 6.1, we recover the standard trade-offs between the two exploration strategies (see Metelli et al., 2018). Specifically, PB exploration suffers from high-dimensional parameterizations (i.e., large $d_\Theta$), while AB struggles with long-lasting interactions with the environment and with high-dimensional action spaces (i.e., large $T$ and $d_\mathcal{A}$).

**Remark 6.1** (On employing Hessian-aided PG subroutines). *We remark that the PG subroutines used within* PES *are not limited to vanilla PG methods, as considered in this paper. For instance, one could employ Hessian-aided algorithms such as HARPG (Fatkhullin et al., 2023), which enjoy faster last-iterate global convergence guarantees of order $\widetilde{\mathcal{O}}(\epsilon^{-2})$. In our setting, this translates into a rate of $\widetilde{\mathcal{O}}(\epsilon^{-2}\sigma^{-2})$ when using a fixed $\sigma$, due to the bound on the policy scores (Assumption 4.4), which scales as $\mathcal{O}(\sigma^{-2})$. As a consequence,* PES *with HARPG converges to the optimal deterministic policy with a sample complexity of $\widetilde{\mathcal{O}}(\epsilon^{-4})$, improving upon our current result at the cost of relying on PG subroutines that incorporate Hessian information.*

## 7. SL-PG: Convergence Analysis

In this section, we discuss the convergence of PG methods jointly learning parameters $\boldsymbol{\theta}$ and stochasticity $\sigma$ (via a suitable parameterization $\xi$) of (hyper)policies. From now on, we refer to this common practice as SL-PG (Stochasticity-Learning Policy Gradients). As discussed in Section 1, this approach has proven to be very successful in practical scenarios (Schulman et al., 2015; Duan et al., 2016; Likmeta et al., 2020), since it potentially allows PGs to *adapt* the stochasticity level to exit from local optima when needed. However, the literature on PG convergence guarantees only considers a static stochasticity (Yuan et al., 2022; Bhandari & Russo, 2024; Montenegro et al., 2024).

SL-PG learns, in addition to the (hyper)policy parameters $\boldsymbol{\theta}$, a certain scalar parameterization $\xi \in \mathbb{R}$ of $\sigma \in \mathbb{R}_{>0}$, employing a *stochasticity mapping function* $f : \mathbb{R} \to [\sigma_{\min}, \sigma_{\max}]$, with $\sigma_{\min}, \sigma_{\max} \in \mathbb{R}_{>0}$, to map $\xi$ to $\sigma$ as $\sigma = f(\xi)$. To assess the convergence of SL-PG, the stochasticity mapping function $f$ has to fulfill the following condition.

**Assumption 7.1** ($f$ Regularity). *There exist $L_{1,f} \geqslant 0$ and $L_{2,f} \geqslant 0$ such that, for every $\xi_1, \xi_2 \in \mathbb{R}$, we have:*

$$|f(\xi_1) - f(\xi_2)| \leqslant L_{1,f}|\xi_1 - \xi_2|,$$

$$\left|\frac{\partial}{\partial\xi}f(\xi_1) - \frac{\partial}{\partial\xi}f(\xi_2)\right| \leqslant L_{2,f}|\xi_1 - \xi_2|.$$

Considering $\boldsymbol{v} := (\boldsymbol{\theta}^\top, \xi)^\top$ as the joint parameterization, SL-PG aims to optimize the objective $\widetilde{J}_\dagger(\boldsymbol{v}) := J_\dagger(\boldsymbol{\theta}, f(\xi))$ by updating the parameterization $\boldsymbol{v}_k$ via stochastic gradient ascent $\boldsymbol{v}_{k+1} \leftarrow \boldsymbol{v}_k + \delta_k \widehat{\nabla}_{\boldsymbol{v}} \widetilde{J}_\dagger(\boldsymbol{v}_k)$, where $\delta_k > 0$ is a step size and $\widehat{\nabla}_{\boldsymbol{v}} \widetilde{J}_\dagger(\cdot)$ is an unbiased estimator of $\nabla_{\boldsymbol{v}} J_\dagger(\cdot)$ (e.g., GPOMDP for **AB** and PGPE for **PB**) computed from a batch of $N$ trajectories.

**Convergence Conditions.** We anticipate that SL-PG exhibits last-iterate convergence guarantees to the optimal *stochastic* (hyper)policy characterized by a joint parameterization $\boldsymbol{v}^* \in \arg\max_{\boldsymbol{v} \in \Theta \times \mathbb{R}} \widetilde{J}_\dagger(\boldsymbol{v})$. We also denote $\widetilde{J}_\dagger^* := \widetilde{J}_\dagger(\boldsymbol{v}^*)$. To achieve such guarantees, we need similar conditions to the ones required for the convergence of PGs with fixed stochasticity (Section 4). In particular, the objective $\widetilde{J}_\dagger(\boldsymbol{v})$ has to fulfill three properties: $(i)$ smoothness of $\widetilde{J}_\dagger(\cdot)$ w.r.t. $\boldsymbol{v}$, $(ii)$ *weak gradient domination* (WGD) of $\widetilde{J}_\dagger(\cdot)$ w.r.t. $\boldsymbol{v}$, and $(iii)$ bounded variance of the gradient estimator. To ensure these properties for the joint parametrization $\boldsymbol{v}$ if the conditions of Section 4 hold for the parametrization $\boldsymbol{\theta}$ (with fixed $\sigma$), we only need to guarantee analogous conditions for the parameter $\xi$.

Condition $(i)$ requires an additional assumption on the smoothness of the deterministic objective with respect to the NS policy, which is necessary for the **AB** case due to the inherent nature of this type of exploration.

**Assumption 7.2** ($J_\mathrm{D}$ is $L_{2,\underline{\boldsymbol{\mu}}}$-LS w.r.t. $\underline{\boldsymbol{\mu}}$). *The performance $J_D$ of the non-stationary deterministic policy $\underline{\boldsymbol{\mu}}$ is $(L_{2,t})_{t=0}^{T-1}$-LC in the non-stationary policy, i.e., for every $\underline{\boldsymbol{\mu}}, \underline{\boldsymbol{\mu}}'$:*

$$\|\nabla_{\underline{\boldsymbol{\mu}}}J_D(\underline{\boldsymbol{\mu}}) - \nabla_{\underline{\boldsymbol{\mu}}}J_D(\underline{\boldsymbol{\mu}}')\|_2 \leqslant \sum_{t=0}^{T-1} L_{2,t}\sup_{\mathbf{s}\in\mathcal{S}}\left\|\boldsymbol{\mu}_t(\mathbf{s}) - \boldsymbol{\mu}_t'(\mathbf{s})\right\|_2.$$

*Furthermore, we denote $L_{2,\underline{\boldsymbol{\mu}}} := \sum_{t=0}^{T-1} L_{2,t}$.*

Then, condition $(i)$ is inherited from the deterministic objective by both $\widetilde{J}_\mathrm{P}$ and $\widetilde{J}_\mathrm{A}$, establishing, in addition to Lemma 4.1, that $J_\dagger(\boldsymbol{\theta}, f(\xi))$ is $L_{1\dagger,\xi}$-LC and $L_{2\dagger,\xi}$-LS w.r.t. $\xi$ (details in Appendix G).

**Lemma 7.1** ($J_\dagger$ is $L_{1\dagger,\xi}$-LC and $L_{2\dagger,\xi}$-LS w.r.t. $\xi$). *Under Assumptions 4.1, 5.1, 5.2, 7.1, and 7.2 it holds that:*

$$\left|\frac{\partial}{\partial\xi}J_\dagger(\boldsymbol{\theta}, f(\xi))\right| \leqslant L_{1\dagger,\xi} \quad and \quad \left|\frac{\partial^2}{\partial\xi^2}J_\dagger(\boldsymbol{\theta}, f(\xi))\right| \leqslant L_{2\dagger,\xi}.$$

When Lemmas 4.1 and 7.1 hold, then $\|\nabla_{\boldsymbol{v}}^2 \widetilde{J}_\dagger(\boldsymbol{v})\| \leqslant L_{2\dagger,\boldsymbol{v}} := L_{2\dagger,\xi} + L_2$.

To meet condition $(ii)$, we introduce directly WGD on $\widetilde{J}_\dagger$ w.r.t. the optimization variable $\boldsymbol{v}$.

**Assumption 7.3** (WGD on $\widetilde{J}_\dagger$ w.r.t. $\boldsymbol{v}$). *There exist $\alpha_{\boldsymbol{v}} > 0$ and $\beta_{\boldsymbol{v}} \geqslant 0$ such that, for every $\boldsymbol{v} \in \mathbb{R}^{d_\Theta+1}$, it holds:*

$$\widetilde{J}_\dagger^* - \widetilde{J}_\dagger(\boldsymbol{v}) \leqslant \alpha_{\boldsymbol{v}}\left\|\nabla_{\boldsymbol{v}}\widetilde{J}_\dagger(\boldsymbol{v})\right\|_2 + \beta_{\boldsymbol{v}}.$$

Notice that the meaning of this assumption is the same of Assumption 4.2 explained in Section 4. Moreover, in Appendix G, we demonstrate that this can be inherited by other assumptions and results introduced previously.

Finally, condition $(iii)$ holds for both $\widetilde{J}_A$ and $\widetilde{J}_P$ under the same assumptions needed for the boundedness of the variance of $\widehat{\nabla}_{\boldsymbol{\theta}} J_\dagger(\boldsymbol{\theta}, \sigma)$. In particular, we have the following result on the boundedness of the variance of $\widehat{\nabla}_\xi J_\dagger(\boldsymbol{\theta}, f(\xi))$.

**Lemma 7.2** (Bounded $\widehat{\nabla}_\xi J_\dagger(\boldsymbol{\theta}, f(\xi))$)**.** *If the (hyper)policy satisfies Definitions 2.2 or 2.3, under Assumptions 4.4 and 5.3, using an exploration mapping $f(\cdot)$ fulfilling Assumption 7.1, for any parameterization $\xi$, it holds that:*

$$\mathbb{V}\mathrm{ar}[\widehat{\nabla}_\xi J_\dagger(\boldsymbol{\theta}, f(\xi))] \leqslant \frac{\mathcal{V}_{\dagger,\xi}}{Nf(\xi)^2}.$$

Then, the variance of $\widehat{\nabla}_{\boldsymbol{v}}\widetilde{J}_\dagger(\boldsymbol{v})$ is bounded by $(\mathcal{V}_{\dagger,\boldsymbol{\theta}} + \mathcal{V}_{\dagger,\xi})N^{-1}f(\xi)^{-2}$ (further details in Appendix G).

**Convergence Results.** We now study the last-iterate convergence of SL-PG to the optimal stochastic (hyper)policy, characterized by a total parameterization $\boldsymbol{v}^*$.

**Theorem 7.3** (SL-PG Convergence)**.** *If the (hyper)policy satisfies Definitions 2.2 or 2.3, under Assumptions 4.1, 4.3, 4.4, 5.1 (PB) or 5.2 (AB), 5.3, 7.1, 7.2, and 7.3, running SL-PG for $K$ iterations, with a suitable constant choice for the learning rate $\delta_k$, the output parameterization $\boldsymbol{v}_K$ is such that:*

$$\widetilde{J}_\dagger^* - \mathbb{E}[\widetilde{J}_\dagger(\boldsymbol{v}_K)] \leqslant \epsilon + \beta_{\boldsymbol{v}},$$

*with a total sample complexity of:*

$$NK = \widetilde{\mathcal{O}}\left(\frac{16\alpha_{\boldsymbol{v}}^4 L_{2\dagger,\boldsymbol{v}}\mathcal{V}_{\dagger,\boldsymbol{v}}}{\epsilon^3\sigma_{\min}^2}\right),$$

*where $L_{2\dagger,\boldsymbol{v}} := L_2 + L_{2\dagger,\xi}$ and $\mathcal{V}_{\dagger,\boldsymbol{v}} := \mathcal{V}_{\dagger,\boldsymbol{\theta}} + \mathcal{V}_{\dagger,\xi}$.*

It is worth noticing that, under more demanding assumptions of the ones reported in Section 4, we recover the same rate of Theorem 4.4 for what concerns the last-iterate convergence to the optimal stochastic (hyper)policy. Moreover, in Appendix G, we recover the same result without of Assumption 7.3. However, if one wants to recover the convergence guarantees to the optimal deterministic policy by employing the result of Theorem 5.1, one has to assume that all the $\epsilon$-optimal stochastic (hyper)policies are such that they have a stochasticity of at most $\epsilon$, i.e., $\widetilde{J}_\dagger^* - \mathbb{E}[\widetilde{J}_\dagger(\boldsymbol{v}_K)] \leqslant \epsilon + \beta_{\boldsymbol{v}} \Rightarrow \sigma_K = f(\xi_K) \leqslant \mathcal{O}(\epsilon)$. Then, it would be possible to employ a stochasticity mapping $f$ such that $\sigma_{\min} = \epsilon$, recovering the same rate $\widetilde{\mathcal{O}}(\epsilon^{-5})$ of PES for last-iterate deterministic convergence, being sure that the learned stochasticity $f(\xi_K)$ would be at most $\epsilon$.

Also in this case, by plugging into the results of Theorem 7.3 the constants for PB ($\dagger = $ P) or AB ($\dagger = $ A) exploration, we recover the usual trade-offs between the two paradigms.

**PES vs. SL-PG.** SL-PG is similar to PES in its dynamic handling of stochasticity $\sigma$, but instead of following a fixed decreasing schedule, it adapts $\sigma$, potentially aiding in escaping local optima. However, this adaptability requires stronger assumptions. In practice, PES may also get stuck in local optima if parameters $y$ and $K_p$ are not chosen carefully. While PES offers theoretical guarantees on performance loss when deploying a deterministic policy, SL-PG provides no such assurance, as the convergence behavior of $\sigma$ remains an open problem without additional assumptions. These aspects will be further discussed in Section 9.

## 8. Related Work

In this section, we discuss related work on convergence rates. Further related literature is provided in Appendix A.

The convergence of PGs to stationary points at a rate of $\mathcal{O}(\epsilon^{-4})$ has been known at least since the work of Sutton et al. (1999). However, the recent analysis by Yuan et al. (2022) and Montenegro et al. (2024) provide further clarification on key aspects and required assumptions. Faster variants of REINFORCE, leveraging stochastic variance reduction techniques, were introduced much later (Papini et al., 2018; Xu et al., 2019). These methods achieved improved convergence rates, with the $\mathcal{O}(\epsilon^{-3})$ rate of (Xu et al., 2020) now considered optimal, supported by lower bounds from non-convex stochastic optimization (Arjevani et al., 2023). The same optimality applies to second-order methods (Shen et al., 2019; Arjevani et al., 2020). While the convergence properties of PGPE are similar to those of PG, they have received comparatively less attention. Notably, (Xu et al., 2020) proved the $\mathcal{O}(\epsilon^{-3})$ rate for a variance-reduced version of PGPE. More recently, the focus has shifted toward studying the global convergence of PG to optimal policies under additional assumptions. Pioneering works in this direction include Scherrer & Geist (2014), Fazel et al. (2018), and Bhandari & Russo (2024). These works introduced the concept of gradient domination (also known as the Polyak-Łojasiewicz condition), which has a long-standing history in optimization (Lojasiewicz, 1963; Polyak, 1963; Karimi et al., 2016). Several studies have analyzed the iteration complexity of PG with exact gradients for specific policy classes, such as softmax or direct tabular parametrization, where gradient domination is guaranteed (e.g., Agarwal et al., 2021; Mei et al., 2020; Li et al., 2021). An exception is the sample-based *natural* policy gradient, which has been studied for general smooth policies (Agarwal et al., 2021). Regarding vanilla sample-based PG (e.g., GPOMDP), Liu et al. (2020) were the first to investigate its sample complexity for global optimality. They also introduced the concept of Fisher-non-degeneracy (Ding et al., 2022), enabling the use of gradient domination for a broader class of policies. Under weaker assumptions, Yuan et al. (2022) and Mon-

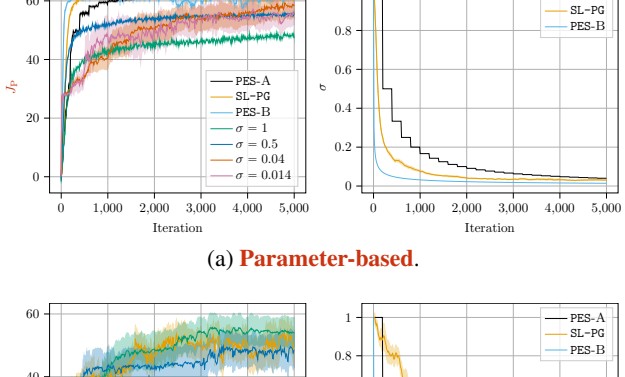

(a) **Parameter-based**.

(b) **Action-based**.

*Figure 1.* $J_\dagger$ and $\sigma$ behavior for **PB** and **AB** in *Swimmer-v5* (5 runs, mean $\pm 95\%$ C.I.).

| Method | PB | $\Delta_\mathrm{P}$ | AB | $\Delta_\mathrm{A}$ |
|---|---|---|---|---|
| PES-A | $61.14 \pm 1.1$ | $-0.12$ | $28.93 \pm 9.89$ | $0.44$ |
| SL-PG | $61.43 \pm 2.73$ | $-0.21$ | $49.83 \pm 13.38$ | $-1.19$ |
| PES-B | $60.40 \pm 2.31$ | $-0.27$ | $29.24 \pm 8.74$ | $0.33$ |

*Table 1.* Deterministic deployment performance (5 runs over 100 trajectories, mean $\pm$ std). $\Delta_\dagger = J_\mathrm{D}(\boldsymbol{\theta}_K) - J_\dagger(\boldsymbol{\theta}_K)$.

tenegro et al. (2024) achieved an improved $\widetilde{\mathcal{O}}(\epsilon^{-3})$ sample complexity. More sophisticated methods, such as variance-reduced approaches, have pushed the sample complexity further. The current state of the art is (Fatkhullin et al., 2023), achieving $\widetilde{\mathcal{O}}(\epsilon^{-2.5})$ for Hessian-free methods and $\widetilde{\mathcal{O}}(\epsilon^{-2})$ for second-order algorithms, with the latter being optimal up to logarithmic terms (Azar et al., 2013). In the case of Gaussian policies, all the aforementioned works implicitly assume that *covariance parameters remain fixed*.

## 9. Experiments

In this section, we numerically validate the presented results. Additional details and experiments are provided in Appendix H.[2] Here, we analyze the behavior of PES and SL-PG in both **AB** and **PB** explorations, comparing them with their static stochasticity counterparts (GPOMDP and PGPE).We conduct the evaluations in the *Swimmer-v5* environment, part of the MuJoCo (Todorov et al., 2012) control suite, using a horizon of $T = 200$. All learning rates are

---

[2]The code is available at https://github.com/MontenegroAlessandro/MagicRL.

managed by the Adam (Kingma & Ba, 2014) optimizer.

For both **PB** and **AB**, we present PES with two different schedules, both starting with $\sigma = 1$. The first (A) schedule consists of $P = 25$ phases, each lasting $K_p = 200$ iterations, with a schedule exponent of $y = 1$. The second (B) schedule includes $P = 5000$ phases, each lasting $K_p = 1$ iteration, with a schedule exponent of $y = 0.5$. SL-PG is executed for $K = 5000$ iterations, using the common exponential parameterization for $\sigma$ (i.e., $\sigma = e^\xi$). The static stochasticity counterparts are also run for $K = 5000$ iterations, employing stochasticity levels $\sigma \in \{1, 0.5, 0.04, 0.014\}$. Here, $\sigma = 1$ represents the maximum stochasticity in the PES schedules, while $\sigma = 0.04$ and $\sigma = 0.014$ correspond to the minima of the first and second PES schedules, respectively.

**Parameter-based.** Figure 1a shows that PES-B outperforms the other methods in convergence to optimal performance, while presenting a similar behavior to SL-PG. PES-A shows a slower convergence due to the longer phases length. For the deterministic deployment PES-A, mimicking what prescribed by theory, scores a higher performance w.r.t. PES-B, showing similar results to SL-PG (Tab. 1).

**Action-based.** This pattern does not hold under **AB**. Figure 1b shows the main drawback of PES, consisting of the possibility of getting stuck in local optima when $K_p$ does not suffice. This is avoided by SL-PG, which has the ability to adapt $\sigma$ to face this situation. This can be noticed also in deterministic deployment scores (Tab. 1).

**Final $\sigma$.** In general, while the final stochasticity of PES is controlled by the imposed schedule, the final stochasticity of SL-PG is unknown until the end of the learning process, as it is learned via stochastic gradient ascent. This has the effect that the loss incurred when switching off the noise is smaller w.r.t. that incurred when using SL-PG (Tab. 1).

## 10. Conclusion

This work studied last-iterate convergence guarantees for PGs with dynamic stochasticity, bridging theory and practice. We introduced PES, a phase-based approach that deterministically scales down stochasticity, enjoying last-iterate convergence to the optimal deterministic policy with a rate of $\widetilde{\mathcal{O}}(\epsilon^{-5})$. Additionally, we analyzed the common practice of jointly learning (hyper)policy parameters and stochasticity, showing its last-iterate convergence to the optimal stochastic (hyper)policy with a rate of $\widetilde{\mathcal{O}}(\epsilon^{-3}\sigma_\mathrm{min}^{-2})$, under stronger assumptions than PES. Future work should further narrow the theory-practice gap by exploring guarantees for single-iteration phases (i.e., $K_p = 1$ for all $p$), relaxing assumptions for SL-PG convergence, and analyzing the learning dynamics of the stochasticity value in SL-PG. Moreover, it would be interesting to determine SL-PG convergence employing a stochasticity mapping $f$ letting $\sigma = 0$.

## Impact Statement

This paper presents work whose goal is to advance the field of Machine Learning. There are many potential societal consequences of our work, none which we feel must be specifically highlighted here.

## Acknowledgments

Funded by the European Union – Next Generation EU within the project NRPP M4C2, Investment 1.3 DD. 341 – 15 March 2022 – FAIR – Future Artificial Intelligence Research – Spoke 4 – PE00000013 – D53C22002380006.

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

# A. Additional Related Work

In this appendix, we discuss additional related work w.r.t. the literature on convergence rates, discussed in Section 8.

**Deterministic Policies.** Value-based RL algorithms, such as Q-learning, naturally deliver deterministic policies as their final solution. In contrast, most PG methods are designed to search within the space of non-degenerate stochastic policies. As noted by Sutton et al. (1999), this characteristic is often seen as an advantage rather than a limitation, as the optimal policy is frequently stochastic in partially observable environments. The ability to deploy deterministic policies is one of the key advantages of PGPE and related evolutionary techniques (Schwefel, 1993; Wierstra et al., 2014), as well as model-based approaches like PILCO (Deisenroth & Rasmussen, 2011). In the domain of action-based policy search, the Deterministic Policy Gradient (DPG) algorithm introduced by Silver et al. (2014) was the first to explicitly focus on searching in the space of deterministic policies. Unlike PGPE, DPG uses stochastic policies during the learning process for exploration, similarly to value-based methods, while largely ignoring distribution mismatch caused by off-policy sampling. Despite this, DPG served as the foundation for popular deep RL algorithms, including (Lillicrap et al., 2016; Fujimoto et al., 2018). In (Xiong et al., 2022), the convergence of *on-policy* DPG (fully deterministic) to a stationary point was established with a sample complexity of order $\mathcal{O}(\epsilon^{-4})$. However, their analysis relies on an explorability assumption (Asm. 4 in their paper), which is standard for stochastic policies but very demanding for deterministic ones. A more practical approach to achieving fully deterministic DPG was proposed by Saleh et al. (2022), who also discuss the benefits of deterministic policies. As expected, fully deterministic learning is only feasible under strong assumptions about the regularity of the environment. Recently, (Montenegro et al., 2024) consider for both parameter-based and action-based PGs the more common scenario of evaluating stochastic policies at training time, only to deploy a good deterministic policy in the end. The authors show that, under weak gradient dominance assumptions, this practice enjoys last-iterate global convergence guarantees to the optimal deterministic policy with a rate of order $\widetilde{\mathcal{O}}(\epsilon^{-5})$, requiring to maintain fixed right from the beginning of the learning process the (hyper)policy variance at a quantity which is $\mathcal{O}(\epsilon)$, limiting the practicality of this approach.

**Policy Variance.** When optimizing Gaussian policies using policy-gradient methods, the scale parameters (such as the variance or, more generally, the covariance matrix of the policy) are often assumed to be fixed in theoretical analyses but are typically optimized via gradient descent in practice. To the best of our knowledge, there is no comprehensive theory addressing the impact of varying policy (or hyperpolicy) variance on the convergence rates of PGs. Ahmed et al. (2019) were among the first to seriously examine how policy stochasticity influences the geometry of the objective function, although their focus was on entropy regularization. Papini et al. (2020) addressed the issue from a different point of view, proposing the use of second-order information to avoid the greediness of gradient updates, which they argue is particularly problematic for scale parameters. Their work emphasizes monotonic improvement rather than convergence guarantees. Bolland et al. (2023) and Mobahi & Fisher (2015) studied PG with Gaussian policies through the framework of *optimization by continuation* (Allgower & Georg, 1990), treating the process as a sequence of smoothed versions of the deterministic policy optimization problem. Unfortunately, the theoretical foundation for optimization by continuation remains sparse, even if this approach presents similarities with our proposal PES. It is important to note that the common practice of *learning* (Wierstra et al., 2014; Schulman et al., 2015; Duan et al., 2016; Likmeta et al., 2020) the exploration parameters alongside the other policy parameters invalidates all known convergence results for GPOMDP. This is because the smoothness of the stochastic objective is inversely proportional to the policy variance (Papini et al., 2022). In contrast, entropy-regularized policy optimization is better analyzed through mirror descent theory rather than traditional stochastic gradient descent theory (Shani et al., 2020).

**Comparing AB and PB Exploration.** A foundational work on this topic is the paper by Zhao et al. (2011), which provides upper bounds on the variance of the REINFORCE and PGPE estimators. Their analysis highlights the better dependence of PGPE on the task horizon compared to REINFORCE. The idea that variance reduction alone does not fully capture the efficiency of policy gradient methods has emerged more recently (Ahmed et al., 2019). Montenegro et al. (2024) revisited the comparison between action-based (AB) and parameter-based (PB) methods through the lens of modern sample complexity theory, arriving at similar conclusions but offering, in our view, a more comprehensive understanding of the subject. Another set of works that extensively compares AB and PB exploration includes (Metelli et al., 2018; 2020; 2021). These studies analyze the trade-off between the task horizon and the number of policy parameters, both theoretically and experimentally, but in the specific context of trust-region methods.

## B. Stochastic Policy Gradients

**Deterministic Parametric Policies.** We consider a *parametric deterministic policy* $\boldsymbol{\mu_\theta} : \mathcal{S} \to \mathcal{A}$, where $\boldsymbol{\theta} \in \Theta \subseteq \mathbb{R}^{d_\Theta}$ is the parameter vector belonging to the parameter space $\Theta$. The performance of $\boldsymbol{\mu_\theta}$ is assessed via the *expected return* $J_\mathrm{D} : \Theta \to \mathbb{R}$, defined as:

$$J_\mathrm{D}(\boldsymbol{\theta}) := \mathop{\mathbb{E}}_{\tau \sim p_\mathrm{D}(\cdot|\boldsymbol{\theta})} [R(\tau)],$$

where

$$p_\mathrm{D}(\tau; \boldsymbol{\theta}) := \rho_0(\mathbf{s}_{\tau,0}) \prod_{t=0}^{T-1} p(\mathbf{s}_{\tau,t+1}|\mathbf{s}_{\tau,t}, \boldsymbol{\mu_\theta}(\mathbf{s}_{\tau,t}))$$

is the density of trajectory $\tau$ induced by policy $\boldsymbol{\mu_\theta}$. The agent's goal consists of finding an optimal parameter $\boldsymbol{\theta}_\mathrm{D}^* \in \arg\max_{\boldsymbol{\theta} \in \Theta} J_\mathrm{D}(\boldsymbol{\theta})$ and we denote $J_\mathrm{D}^* := J_\mathrm{D}(\boldsymbol{\theta}_\mathrm{D}^*)$.

**White Noise Exploration.** As it is standard for papers concerning PG convergence (Papini et al., 2018; Yuan et al., 2022; Fatkhullin et al., 2023), in this paper we consider white-noise-based *undirected* exploration. As discussed in Montenegro et al. (2024), this kind of exploration is just a characterization of stochastic (hyper)policies which is particularly intuitive when one has to deal with selecting the exploration amount. Before formally defining *action-based* and *parameter-based* exploration, we provide a preliminary definition of white noise.

**Definition B.1** (White Noise). *Let $d \in \mathbb{N}$ and $\sigma > 0$. A probability distribution $\Phi_{d,\sigma} \in \Delta(\mathbb{R}^d)$ is a white noise if:*

$$\mathop{\mathbb{E}}_{\boldsymbol{\epsilon} \sim \Phi_{d,\sigma}} [\boldsymbol{\epsilon}] = \mathbf{0}_d \quad and \quad \mathop{\mathbb{E}}_{\boldsymbol{\epsilon} \sim \Phi_{d,\sigma}} [\|\boldsymbol{\epsilon}\|_2^2] \leqslant d\sigma^2. \tag{1}$$

Definition 2.1 includes also zero-mean gaussian distributions $\boldsymbol{\epsilon} \sim \mathcal{N}(\mathbf{0}_d, \Sigma)$, where $\mathbb{E}_{\boldsymbol{\epsilon} \sim \mathcal{N}(\mathbf{0}_d, \Sigma)}[\|\boldsymbol{\epsilon}\|_2^2] = \mathrm{tr}(\Sigma) \leqslant d\sigma^2$. Note that the noise is intended to be white among different time (or iteration) steps, while the noise vector for a particular realization may not be white. For instance, if the distribution at hand is Gaussian, it may be non-isotropic. Moreover, in principle it is possible to employ different white noise distributions throughout the exploration steps.

**Action-Based (AB) Exploration.** In AB exploration, we consider a *parametric stochastic policy* $\pi_{\boldsymbol{\rho}} : \mathcal{S} \to \Delta(\mathcal{A})$, where $\boldsymbol{\rho} \in \mathcal{P} \subseteq \mathbb{R}^{d_\mathcal{P}}$ is the parameter vector. The policy is used to sample actions $\mathbf{a}_t \sim \pi_{\boldsymbol{\rho}}(\cdot|\mathbf{s}_t)$ to be played in state $\mathbf{s}_t$ for *every step $t$* of interaction. Stochastic policies $\pi_{\boldsymbol{\rho}}$ can be considered as built upon an underlying deterministic policy $\boldsymbol{\mu_\theta}$ (over the same parameter space $\Theta$), by perturbing each action suggested by $\boldsymbol{\mu_\theta}$ with a white-noise random vector. Formally, we consider the following definition of *white noise policies*:

**Definition B.2** (White noise policies). *Let $\boldsymbol{\theta} \in \Theta$, $\sigma > 0$, and $\boldsymbol{\mu_\theta} : \mathcal{S} \to \mathcal{A}$ be a parametric deterministic policy and let $\Phi_{d_\mathcal{A}, \sigma}$ be a white noise distribution compliant with Definition 2.1. A white noise policy $\pi_{\boldsymbol{\theta}, \sigma} : \mathcal{S} \to \Delta(\mathcal{A})$ is such that, for every state $\boldsymbol{s} \in \mathcal{S}$, the action $\boldsymbol{a} \sim \pi_{\boldsymbol{\theta}, \sigma}(\cdot|\boldsymbol{s})$ satisfies $\boldsymbol{a} = \boldsymbol{\mu_\theta}(\boldsymbol{s}) + \boldsymbol{\epsilon}$, where $\boldsymbol{\epsilon} \sim \Phi_{d_\mathcal{A}, \sigma}$ which is sampled independently at every step (i.e., whenever an action is sampled).*

Note that $(i)$ a policy complying with Definition 2.2 is a standard stochastic policy and $(ii)$ this definition of white noise policy justifies the name for AB exploration, since the exploration is carried out at the action level. From now on, we refer to this kind of stochastic policies, referring to $\sigma$ as the *exploration amount* for $\pi_{\boldsymbol{\theta}, \sigma}$.

The performance of $\pi_{\boldsymbol{\theta}}$ is assessed via the *expected return* $J_\mathrm{A} : \Theta \times \mathbb{R} \to \mathbb{R}$, defined as:

$$J_\mathrm{A}(\boldsymbol{\theta}, \sigma) := \mathop{\mathbb{E}}_{\tau \sim p_\mathrm{A}(\cdot|\boldsymbol{\theta})} [R(\tau)],$$

where

$$p_\mathrm{A}(\tau; \boldsymbol{\theta}) := \rho_0(\mathbf{s}_{\tau,0}) \prod_{t=0}^{T-1} \pi_{\boldsymbol{\theta}, \sigma}(\mathbf{a}_{\tau,t}|\mathbf{s}_{\tau,t}) p(\mathbf{s}_{\tau,t+1}|\mathbf{s}_{\tau,t}, \mathbf{a}_{\tau,t})$$

is the density of trajectory $\tau$ induced by policy $\pi_{\boldsymbol{\theta}, \sigma}$.

Given that we consider white-noise policies, another definition of $J_\mathrm{A}$ is possible. Let $\underline{\boldsymbol{\mu}} = (\boldsymbol{\mu})_{t=0}^{T-1}$ be a *non-stationary* deterministic policy, where, at time step $t$ the deterministic policy $\boldsymbol{\mu}_t : \mathcal{S} \to \mathcal{A}$ is played. The objective for this kind of policy is $J_\mathrm{D}(\underline{\boldsymbol{\mu}}) = \mathbb{E}_{\tau \sim p_\mathrm{D}(\cdot|\underline{\boldsymbol{\mu}})}[R(\tau)]$, with $p_\mathrm{D}(\tau|\underline{\boldsymbol{\mu}}) = \boldsymbol{\rho}_0(s_{\tau,0}) \prod_{t=0}^{T-1} p(s_{\tau,t+1}|s_{\tau,t}, \boldsymbol{\mu}_t(s_{\tau,t}))$. Now, let $\underline{\boldsymbol{\epsilon}} = (\epsilon_t)_{t=0}^{T-1} \sim \Phi_{d_\mathcal{A}, \sigma}^T$ be a sequence of independently sampled noise vectors. We can denote $\underline{\boldsymbol{\mu_\theta}} + \underline{\boldsymbol{\epsilon}} = (\boldsymbol{\mu_\theta} + \epsilon_t)_{t=0}^{T-1}$. Thus, the AB performance

index can be expressed as:

$$J_{\mathrm{A}}(\boldsymbol{\theta}, \sigma) = \mathop{\mathbb{E}}_{\underline{\boldsymbol{\epsilon}} \sim \Phi_{d_{\mathcal{A}}, \sigma}^{T}} \left[ J_{\mathrm{D}}(\underline{\boldsymbol{\mu_{\theta}}} + \underline{\boldsymbol{\epsilon}}) \right].$$

In AB exploration, whenever $\sigma > 0$, we aim to learn $\boldsymbol{\theta}_{\mathrm{A}}^{*}(\sigma) \in \arg\max_{\boldsymbol{\theta} \in \Theta} J_{\mathrm{A}}(\boldsymbol{\theta}, \sigma)$. If $J_{\mathrm{A}}(\boldsymbol{\theta}, \sigma)$ is differentiable w.r.t. $\boldsymbol{\theta}$, PG methods (Peters & Schaal, 2008) update the parameter $\boldsymbol{\theta}$ via gradient ascent: $\boldsymbol{\theta}_{t+1} \leftarrow \boldsymbol{\theta}_t + \zeta_t \widehat{\nabla}_{\boldsymbol{\theta}} J_{\mathrm{A}}(\boldsymbol{\theta}_t, \sigma)$, where $\zeta_t > 0$ is the *step size* and $\widehat{\nabla}_{\boldsymbol{\theta}} J_{\mathrm{A}}(\boldsymbol{\theta}, \sigma)$ is an estimator of $\nabla_{\boldsymbol{\theta}} J_{\mathrm{A}}(\boldsymbol{\theta}, \sigma)$. In particular, the GPOMDP *estimator* (Baxter & Bartlett, 2001) is:

$$\widehat{\nabla}_{\boldsymbol{\theta}} J_{\mathrm{A}}(\boldsymbol{\theta}, \sigma) \coloneqq \frac{1}{N} \sum_{i=1}^{N} \sum_{t=0}^{T-1} \left( \sum_{k=0}^{t} \nabla_{\boldsymbol{\theta}} \log \pi_{\boldsymbol{\theta}, \sigma}(\mathbf{a}_{\tau_i, k} | \mathbf{s}_{\tau_i, k}) \right) \gamma^t r(\mathbf{s}_{\tau_i, t}, \mathbf{a}_{\tau_i, t}),$$

where $N$ is the number of independent trajectories $\{\tau_i\}_{i=1}^{N}$ collected with policy $\pi_{\boldsymbol{\theta}, \sigma}$ ($\tau_i \sim p_{\mathrm{A}}(\cdot; \boldsymbol{\theta})$), called *batch size*. In this paper, we just consider GPOMDP, and we will not present REINFORCE, given that these two solutions are conceptually similar, and the estimator considered in the latter suffers from larger variance (Williams, 1992; Baxter & Bartlett, 2001).

**Parameter-Based (PB) Exploration.** In PB exploration, we use a *parametric stochastic hyperpolicy* $\nu_{\boldsymbol{\rho}} \subseteq \Delta(\Theta)$, where $\boldsymbol{\rho} \in \mathcal{P} \subseteq \mathbb{R}^{d_{\mathcal{P}}}$ is the parameter vector. The hyperpolicy is used to sample parameters $\boldsymbol{\theta} \sim \nu_{\boldsymbol{\rho}}$ to be plugged in the deterministic policy $\boldsymbol{\mu_{\theta}}$ at the beginning of *every trajectory*. As for the AB case, stochastic hyperpolicies $\nu_{\boldsymbol{\rho}}$ can be considered as built upon an underlying deterministic policy $\mu_{\boldsymbol{\theta}}$ (over the same parameter space $\Theta$), by perturbing the parameter vector $\boldsymbol{\theta}$ with a white-noise random vector. Formally, we consider the following definition of *white noise hyperpolicies*:

**Definition B.3** (White noise hyperpolicies). *Let $\boldsymbol{\theta} \in \Theta$, $\sigma > 0$, and $\mu_{\boldsymbol{\theta}} : \mathcal{S} \to \mathcal{A}$ be a parametric deterministic policy and let $\Phi_{d_{\Theta}, \sigma}$ be a white-noise distribution compliant with Definition 2.1. A white noise hyperpolicy $\nu_{\boldsymbol{\theta}, \sigma} \in \Delta(\Theta)$ is such that, for every parameterization $\boldsymbol{\theta} \in \Theta$, the parameterization $\boldsymbol{\theta}' \sim \nu_{\boldsymbol{\theta}, \sigma}$ satisfies $\boldsymbol{\theta}' = \boldsymbol{\theta} + \boldsymbol{\epsilon}$, where $\boldsymbol{\epsilon} \sim \Phi_{d_{\Theta}, \sigma}$ which is sampled independently at every trajectory.*

Also in this case, $(i)$ a hyperpolicy complying with Definition 2.3 is a standard stochastic hyperpolicy and $(ii)$ this definition of white noise hyperpolicy justifies the name for PB exploration, since the exploration is carried out at the parameter level. It is worth noticing that, differently from what happens in AB exploration, in PB exploration, before starting to collect each trajectory, the current parametrization for the deterministic policy $\boldsymbol{\theta}$ is perturbed with a noise vector $\boldsymbol{\epsilon} \sim \Phi_{d_{\Theta}}(\sigma)$, then the deterministic policy $\mu_{\boldsymbol{\theta} + \boldsymbol{\epsilon}}$ is used for the entire trajectory. From now on, we focus on this kind of stochastic hyperpolicies, referring to $\sigma$ as the *exploration amount* for $\nu_{\boldsymbol{\theta}, \sigma}$.

The performance index of $\nu_{\boldsymbol{\theta}, \sigma}$ is $J_{\mathrm{P}} : \mathbb{R}^{d_{\Theta}} \times \mathbb{R} \to \mathbb{R}$, that is the expectation over $\boldsymbol{\theta}'$ of $J_{\mathrm{D}}(\boldsymbol{\theta}')$ defined as:

$$J_{\mathrm{P}}(\boldsymbol{\theta}, \sigma) \coloneqq \mathop{\mathbb{E}}_{\boldsymbol{\theta}' \sim \nu_{\boldsymbol{\theta}, \sigma}} \left[ J_{\mathrm{D}}(\boldsymbol{\theta}') \right].$$

Leveraging on the definition of white noise hyperpolicy, we can provide the following additional definition for the PB performance index $J_{\mathrm{P}}$:

$$J_{\mathrm{P}}(\boldsymbol{\theta}, \sigma) = \mathop{\mathbb{E}}_{\boldsymbol{\epsilon} \sim \Phi_{d_{\Theta}, \sigma}} \left[ J_{\mathrm{D}}(\boldsymbol{\theta} + \boldsymbol{\epsilon}) \right]$$

Whenever $\sigma > 0$, PB exploration aims at learning $\boldsymbol{\theta}_{\mathrm{P}}^{*}(\sigma) \in \arg\max_{\boldsymbol{\theta} \in \Theta} J_{\mathrm{P}}(\boldsymbol{\theta}, \sigma)$. If $J_{\mathrm{P}}(\boldsymbol{\theta}, \sigma)$ is differentiable w.r.t. $\boldsymbol{\theta}$, PGPE (Sehnke et al., 2010) updates the hyperparameter $\boldsymbol{\theta}$ by gradient ascent: $\boldsymbol{\theta}_{t+1} \leftarrow \boldsymbol{\theta}_t + \zeta_t \widehat{\nabla}_{\boldsymbol{\theta}} J_{\mathrm{P}}(\boldsymbol{\theta}_t, \sigma)$. PGPE uses an estimator of $\nabla_{\boldsymbol{\theta}} J_{\mathrm{P}}(\boldsymbol{\theta}, \sigma)$ defined as:

$$\widehat{\nabla}_{\boldsymbol{\theta}} J_{\mathrm{P}}(\boldsymbol{\theta}, \sigma) = \frac{1}{N} \sum_{i=1}^{N} \nabla_{\boldsymbol{\theta}} \log \nu_{\boldsymbol{\theta}, \sigma}(\boldsymbol{\theta}_i) R(\tau_i),$$

where the *batch size* $N$ is the number of independent parameter-trajectory pairs $\{(\boldsymbol{\theta}_i, \tau_i)\}_{i=1}^{N}$, collected with hyperpolicy $\nu_{\boldsymbol{\theta}, \sigma}$, that is, $\boldsymbol{\theta}_i \sim \nu_{\boldsymbol{\theta}, \sigma}$ and $\tau_i \sim p_{\mathrm{D}}(\cdot; \boldsymbol{\theta}_i)$.

## C. Exploration Mapping Quantities

In this section, we present a mapping table of various quantities, often expressed in their exploration-agnostic form along this work, to the specific expression they have in the **Parameter-based** and **Action-based** exploration paradigms.

| Constant | Parameter-based | Action-based |
|:---:|:---:|:---:|
| $D_\dagger$ | $L_J \sqrt{d_\Theta}$ | $L_{1,\boldsymbol{\mu}} \sqrt{d_\mathcal{A}}$ |
| $\mathcal{W}_\dagger$ | $\alpha_\mathrm{D} L_2 \sqrt{d_\Theta} + L_J \sqrt{d_\Theta}$ | $\alpha_\mathrm{D} \psi \sqrt{d_\mathcal{A}} + L_{1,\boldsymbol{\mu}} \sqrt{d_\mathcal{A}}$ |
| $\mathcal{V}_{\dagger,\boldsymbol{\theta}}$ | $\frac{R_\mathrm{max}^2 c d_\Theta}{N(1-\gamma)^2}$ | $\frac{R_\mathrm{max}^2 c d_\mathcal{A} L_\mu^2}{N(1-\gamma)^3}$ |
| $L_{1\dagger,\xi}$ | $L_J L_{1,f} \sqrt{d_\Theta}$ | $L_{1,\boldsymbol{\mu}} L_{1,f} \sqrt{T d_\mathcal{A}}$ |
| $L_{2\dagger,\xi}$ | $L_2 L_{1,f}^2 d_\Theta + L_J L_{2,f} \sqrt{d_\Theta}$ | $L_{2,\boldsymbol{\mu}} L_{1,f}^2 T d_\mathcal{A} + L_{1,\boldsymbol{\mu}} L_{2,f} \sqrt{T d_\mathcal{A}}$ |
| $L_{3\dagger,\xi}$ | $L_2 L_{1,f} \sqrt{d_\Theta}$ | $L_{2,\boldsymbol{\mu}} L_{1,f} \sqrt{T d_\mathcal{A}}$ |
| $\mathcal{V}_{\dagger,\xi}$ | $\frac{R_\mathrm{max}^2 c d_\Theta^2 L_{1,f}^2}{N(1-\gamma)^2}$ | $\frac{R_\mathrm{max}^2 c d_\mathcal{A}^2 L_{1,f}^2}{N(1-\gamma)^3}$ |

As can be noticed from the table, there is a trade-off between PB and AB explorations, as already highlighted in previous works (Metelli et al., 2018; Montenegro et al., 2024). Indeed, PB exploration may suffer from large parameterizations (i.e., $d_\Theta$ large), while AB exploration may suffer from long-lasting interactions with the environment and from highly-dimensioned actions spaces (i.e., $T$ or $d_\mathcal{A}$ large). In particular, for AB exploration, when we are in an infinite horizon setting (i.e., $\gamma < 1$ and $T = +\infty$) we identify the length of a trajectory with the effective horizon $T \approx \widetilde{\mathcal{O}}(1/(1-\gamma))$. This approximation only affects logarithmic terms in the sample complexity (Yuan et al., 2022).

# D. Details on the Convergence of PG Subroutines

As highlighted in Section 4, in order guarantee last-iterate global convergence of a PG subroutine employing a (hyper)policy complying with Definitions 2.2 and 2.3 with $\sigma > 0$ as exploration amount, three conditions are needed: $(i)$ smoothness of $J_\dagger(\boldsymbol{\theta}, \sigma)$ w.r.t. $\boldsymbol{\theta}$, $(ii)$ WGD on $J_\dagger(\boldsymbol{\theta}, \sigma)$ w.r.t. $\boldsymbol{\theta}$, and $(iii)$ bounded variance of the estimator $\widehat{\nabla}_{\boldsymbol{\theta}} J_\dagger(\boldsymbol{\theta}, \sigma)$.

## D.1. Inherited Smoothness

The smoothness of $J_\mathrm{P}$ and $J_\mathrm{A}$ is simply inherited by the smoothness of $J_\mathrm{D}$.

**Lemma 4.1** ($J_\dagger$ Inherited Smoothness). *Under Assumption 4.1, for every $\boldsymbol{\theta} \in \Theta$ and $\sigma \in \mathbb{R}_{\geqslant 0}$, it holds:*

$$\|\nabla_{\boldsymbol{\theta}}^2 J_\dagger(\boldsymbol{\theta}, \sigma)\|_2 \leqslant L_2.$$

*Proof.* See Lemmas D.3 and D.7 by Montenegro et al. (2024). $\square$

## D.2. Inherited Weak Gradient Domination

WGD on $J_\mathrm{P}$ and $J_\mathrm{A}$ can be inherited by WGD on $J_\mathrm{D}$, with additional characterization of the MDP and the deterministic policy $\mu_{\boldsymbol{\theta}}$. Such an assumption, that has already been stated in the main paper (Asm 4.3), is rewritten here in Assumptions D.1 and D.2 for the sake of rigor and clarity.

**Assumption D.1** (MDP Regularity). *The log transition model $\log p(\boldsymbol{s}'|\boldsymbol{s}, \cdot)$ is $L_p$-LC and $L_{2,p}$-LS, and the reward function $r(\boldsymbol{s}, \cdot)$ is $L_r$-LC and $L_{2,r}$-LS w.r.t. the action for every $\boldsymbol{s}, \boldsymbol{s}' \in \mathcal{S}$, i.e., for every $\boldsymbol{a}, \boldsymbol{a}' \in \mathcal{A}$:*

$$|\log p(\boldsymbol{s}'|\boldsymbol{s}, \boldsymbol{a}) - \log p(\boldsymbol{s}'|\boldsymbol{s}, \boldsymbol{a}')| \leqslant L_p \|\boldsymbol{a} - \boldsymbol{a}'\|_2$$
$$\|\nabla_{\boldsymbol{a}} \log p(\boldsymbol{s}'|\boldsymbol{s}, \boldsymbol{a}) - \nabla_{\boldsymbol{a}} \log p(\boldsymbol{s}'|\boldsymbol{s}, \boldsymbol{a}')\|_2 \leqslant L_{2,p} \|\boldsymbol{a} - \boldsymbol{a}'\|_2$$
$$|r(\boldsymbol{s}, \boldsymbol{a}) - r(\boldsymbol{s}, \boldsymbol{a}')| \leqslant L_r \|\boldsymbol{a} - \boldsymbol{a}'\|_2$$
$$\|r(\boldsymbol{s}, \boldsymbol{a}) - r(\boldsymbol{s}, \boldsymbol{a}')\|_2 \leqslant L_{2,r} \|\boldsymbol{a} - \boldsymbol{a}'\|_2.$$

**Assumption D.2** ($\mu_{\boldsymbol{\theta}}$ Regularity). *The deterministic policy $\mu_{\boldsymbol{\theta}}(\boldsymbol{s})$ is $L_\mu$-LC and $L_{2,\mu}$-LS w.r.t. parameter for every $\boldsymbol{s} \in \mathcal{S}$, i.e., for every $\boldsymbol{\theta}, \boldsymbol{\theta}' \in \Theta$:*

$$\|\mu_{\boldsymbol{\theta}}(\boldsymbol{s}) - \mu_{\boldsymbol{\theta}'}(\boldsymbol{s})\|_2 \leqslant L_\mu \|\boldsymbol{\theta} - \boldsymbol{\theta}'\|_2$$
$$\|\nabla_{\boldsymbol{\theta}} \mu_{\boldsymbol{\theta}}(\boldsymbol{s}) - \nabla_{\boldsymbol{\theta}} \mu_{\boldsymbol{\theta}'}(\boldsymbol{s})\|_2 \leqslant L_{2,\mu} \|\boldsymbol{\theta} - \boldsymbol{\theta}'\|_2.$$

Under Assumptions D.1 and D.2, WGD can be inherited by $J_\mathrm{P}$ and $J_\mathrm{A}$ when also Assumption 4.2 holds (i.e., WGD on $J_\mathrm{D}$ w.r.t. $\boldsymbol{\theta}$). We stress that these two assumptions are needed for the **AB** case. Thus, they are unavoidable when one wants to consider an exploration-agnostic setting.

**Theorem 4.2** (Inherited WGD for $J_\dagger$, Montenegro et al. 2024). *Considering a (hyper)policy complying with Definitions 2.2 or 2.3, under Assumptions 4.2 and 4.3, for any $\sigma \in \mathbb{R}_{>0}$ and $\boldsymbol{\theta} \in \Theta$, the following holds:*

$$J_\dagger(\boldsymbol{\theta}^*(\sigma), \sigma) - J_\dagger(\boldsymbol{\theta}, \sigma) \leqslant \alpha_D \|\nabla_{\boldsymbol{\theta}} J_\dagger(\boldsymbol{\theta}, \sigma)\|_2 + \beta_\dagger(\sigma),$$

*with $\beta_\dagger(\sigma) := \beta_D + \mathcal{W}_\dagger \sigma$, for some $\mathcal{W}_\dagger \geqslant 0$.*

*Proof.* See Theorems 7.1 and 7.2 by Montenegro et al. (2024). $\square$

For what concerns the quantity $\beta_\dagger(\sigma) = \beta_\mathrm{D} + \mathcal{W}_\dagger \sigma$, we have the following mapping to **PB** and **AB** explorations:

- **PB**: $\mathcal{W}_\mathrm{P} = \alpha_\mathrm{D} L_2 \sqrt{d_\Theta} + L_J \sqrt{d_\Theta}$;
- **AB**: $\mathcal{W}_\mathrm{A} = \alpha_\mathrm{D} \psi \sqrt{d_\mathcal{A}} + L \sqrt{d_\mathcal{A}}$, with $\psi = \mathcal{O}((1 - \gamma)^{-4})$. In particular:

$$\psi = L_\mu \left( \frac{L_p^2 R_{\max} \gamma}{(1 - \gamma)^4} + \frac{(L_r L_p + R_{\max} L_{2,p} + L_p L_r \gamma)}{(1 - \gamma)^2} + \frac{L_{2,r}}{1 - \gamma} \right) (1 - \gamma^T),$$

as can be seen in Lemma D.11 by Montenegro et al. (2024).

### D.3. Bounded Variance of the Estimators

**Lemma 4.3** (Bounded Estimator Variance). *Considering a (hyper)policy complying with Definitions 2.2 (**PB**) or 2.3 (**AB**), under Assumptions 4.3 and 4.4 the following holds:*

$$\mathbb{Var}[\widehat{\nabla}_{\boldsymbol{\theta}} J_{\dagger}(\boldsymbol{\theta}, \sigma)] \leqslant \frac{\mathcal{V}_{\dagger,\boldsymbol{\theta}}}{N\sigma^2}, \quad for\ some\ \mathcal{V}_{\dagger,\boldsymbol{\theta}} \geqslant 0.$$

*Proof.* See Lemmas D.2 and D.6 by Montenegro et al. (2024). □

It is worth noticing that for **PB** exploration with a hyperpolicy complying with Definition 2.3, it suffices to be under Assumption 4.4 to obtain the result of Lemma 4.3.

### D.4. PG Global Last-Iterate Convergence

**Theorem 4.4** (PG Global Last-Iterate Convergence, Montenegro et al. 2024). *Consider running for $K_p$ iterations a PG algorithm with a (hyper)policy satisfying Definitions 2.2 or 2.3 with an exploration amount $\sigma_p \in \mathbb{R}_{>0}$. Under Assumptions 4.1, 4.2, 4.3, and 4.4, by selecting an appropriate constant step size $\zeta_p$ and a sample complexity $NK_p$ satisfying:*

$$NK_p \geqslant \widetilde{\mathcal{O}}\left(\frac{\alpha_D^4 L_2 \mathcal{V}_{\dagger,\boldsymbol{\theta}}}{\epsilon^3 \sigma_p^2}\right),$$

*then $J_{\dagger}(\boldsymbol{\theta}^*(\sigma_p), \sigma_p) - \mathbb{E}[J_{\dagger}(\boldsymbol{\theta}_p, \sigma_p)] \leqslant \epsilon + \beta_{\dagger}(\sigma_p)$, where the notation $\widetilde{\mathcal{O}}(\cdot)$ hides logarithmic terms.*

*Proof.* This result follows directly from Theorem 6.1 by Montenegro et al. (2024). However, in the original result, the term $\sigma_p^{-2}$ is hidden inside the variance bound term. here, we explicitly show it because we need to take care of the dependence on the specific exploration amount since we have to consider it to be dynamic. □

# E. Proofs of Section 5

**Theorem 5.1.** *If the (hyper)policy complies with Definitions 2.2 (**PB**) or 2.3 (**AB**), under Assumptions 5.1 (**PB**) or 5.2 (**AB**), and Assumption 5.3, for every $\boldsymbol{\theta} \in \Theta$ and $\sigma_1, \sigma_2 \in \mathbb{R}_{\geqslant 0}$, it holds that:*

$$(i) \ |J_\dagger(\boldsymbol{\theta}, \sigma_1) - J_\dagger(\boldsymbol{\theta}, \sigma_2)| \leqslant D_\dagger |\sigma_1 - \sigma_2|$$
$$(ii) \ J_\dagger^*(\sigma_1) - J_\dagger(\boldsymbol{\theta}^*(\sigma_2), \sigma_1) \leqslant 2D_\dagger |\sigma_1 - \sigma_2|$$
$$(iii) \ |J_\dagger(\boldsymbol{\theta}^*(\sigma_1), \sigma_1) - J_\dagger(\boldsymbol{\theta}^*(\sigma_2), \sigma_2)| \leqslant D_\dagger |\sigma_1 - \sigma_2|,$$

*where $\dagger \in \{P, A\}$, $D_A = L_{1,\boldsymbol{\mu}} \sqrt{d_\mathcal{A}}$, and $D_P = L_J \sqrt{d_\Theta}$.*

*Proof.* To prove result $(i)$, we have to consider the specific exploration strategies to exploit the additional noise characterization provided by Assumption 5.3.

**Result $(i)$: Parameter-based.** The objective $J_P$ has the form:

$$J_P(\boldsymbol{\theta}, \sigma) = \mathop{\mathbb{E}}_{\boldsymbol{\epsilon} \sim \Phi_{d_\Theta, \sigma}} [J_P(\boldsymbol{\theta} + \boldsymbol{\epsilon}, 0)] = \mathop{\mathbb{E}}_{\boldsymbol{\epsilon} \sim \Phi_{d_\Theta, \sigma}} [J_D(\boldsymbol{\theta} + \boldsymbol{\epsilon})], \tag{2}$$

for any parameterization $\boldsymbol{\theta} \in \Theta$ and exploration amount $\sigma \in \mathbb{R}_{>0}$.

Thus, we have the following:

$$|J_P(\boldsymbol{\theta}, \sigma_1) - J_P(\boldsymbol{\theta}, \sigma_2)| \tag{3}$$

$$= \left| \mathop{\mathbb{E}}_{\boldsymbol{\epsilon}_1 \sim \Phi_{d_\Theta, \sigma_1}} [J_D(\boldsymbol{\theta} + \boldsymbol{\epsilon}_1)] - \mathop{\mathbb{E}}_{\boldsymbol{\epsilon}_2 \sim \Phi_{d_\Theta, \sigma_2}} [J_D(\boldsymbol{\theta} + \boldsymbol{\epsilon}_2)] \right|. \tag{4}$$

Since we are under Assumption 5.3, we can write the following:

$$|J_P(\boldsymbol{\theta}, \sigma_1) - J_P(\boldsymbol{\theta}, \sigma_2)| \tag{5}$$

$$= \left| \mathop{\mathbb{E}}_{\bar{\boldsymbol{\epsilon}} \sim \Phi_{d_\Theta, 1}} [J_D(\boldsymbol{\theta} + \sigma_1 \bar{\boldsymbol{\epsilon}}) - J_D(\boldsymbol{\theta} + \sigma_2 \bar{\boldsymbol{\epsilon}})] \right| \tag{6}$$

$$\leqslant \mathop{\mathbb{E}}_{\bar{\boldsymbol{\epsilon}} \sim \Phi_{d_\Theta, 1}} [|J_D(\boldsymbol{\theta} + \sigma_1 \bar{\boldsymbol{\epsilon}}) - J_D(\boldsymbol{\theta} + \sigma_2 \bar{\boldsymbol{\epsilon}})|], \tag{7}$$

where we applied Jensen's inequality.

By applying Assumption 5.1, we obtain:

$$|J_P(\boldsymbol{\theta}, \sigma_1) - J_P(\boldsymbol{\theta}, \sigma_2)| \tag{8}$$

$$\leqslant \mathop{\mathbb{E}}_{\bar{\boldsymbol{\epsilon}} \sim \Phi_{d_\Theta, 1}} [|J_D(\boldsymbol{\theta} + \sigma_1 \bar{\boldsymbol{\epsilon}}) - J_D(\boldsymbol{\theta} + \sigma_2 \bar{\boldsymbol{\epsilon}})|] \tag{9}$$

$$\leqslant L_J \mathop{\mathbb{E}}_{\bar{\boldsymbol{\epsilon}} \sim \Phi_{d_\Theta, 1}} [\|\boldsymbol{\theta} + \sigma_1 \bar{\boldsymbol{\epsilon}} - \boldsymbol{\theta} - \sigma_2 \bar{\boldsymbol{\epsilon}}\|_2] \tag{10}$$

$$= L_J \mathop{\mathbb{E}}_{\bar{\boldsymbol{\epsilon}} \sim \Phi_{d_\Theta, 1}} [\|\sigma_1 \bar{\boldsymbol{\epsilon}} - \sigma_2 \bar{\boldsymbol{\epsilon}}\|_2] \tag{11}$$

$$\leqslant L_J \sqrt{\mathop{\mathbb{E}}_{\bar{\boldsymbol{\epsilon}} \sim \Phi_{d_\Theta, 1}} [\|\sigma_1 \bar{\boldsymbol{\epsilon}} - \sigma_2 \bar{\boldsymbol{\epsilon}}\|_2^2]} \tag{12}$$

$$= L_J \sqrt{\mathop{\mathbb{E}}_{\bar{\boldsymbol{\epsilon}} \sim \Phi_{d_\Theta, 1}} [\|\bar{\boldsymbol{\epsilon}}\|_2^2 (\sigma_2 - \sigma_2)^2]} \tag{13}$$

$$\leqslant L_J \sqrt{d_\Theta} |\sigma_1 - \sigma_2|, \tag{14}$$

where we applied the Cauchy-Schwartz inequality and Definition 2.1.

**Result $(i)$: Action-based.** The proof for this point is similar to the one for result $(i)$ (**PB**). Here, we can express $J_A$ as:

$$J_A(\boldsymbol{\theta}, \sigma) = \mathop{\mathbb{E}}_{\underline{\boldsymbol{\epsilon}} \sim \Phi_{d_\mathcal{A}, \sigma}^T} [J_D(\underline{\boldsymbol{\mu}}_{\boldsymbol{\theta}} + \underline{\boldsymbol{\epsilon}})], \tag{15}$$

for every parameterization $\boldsymbol{\theta} \in \Theta$ and exploration amount $\sigma \in \mathbb{R}^+$.

Since we are under Assumption 5.3, we can write the following (with a little abuse of notation):

$$|J_{\mathrm{A}}(\boldsymbol{\theta}, \sigma_1) - J_{\mathrm{A}}(\boldsymbol{\theta}, \sigma_2)| \tag{16}$$

$$= \left| \underset{\boldsymbol{\epsilon}_1 \sim \Phi_{d_{\mathcal{A}}, \sigma_1}^T}{\mathbb{E}} \left[ J_{\mathrm{D}}(\boldsymbol{\mu}_{\boldsymbol{\theta}} + \boldsymbol{\epsilon}_1) \right] - \underset{\boldsymbol{\epsilon}_2 \sim \Phi_{d_{\mathcal{A}}, \sigma_2}^T}{\mathbb{E}} \left[ J_{\mathrm{D}}(\boldsymbol{\mu}_{\boldsymbol{\theta}} + \boldsymbol{\epsilon}_2) \right] \right|. \tag{17}$$

By applying Assumption 5.1, we obtain:

$$|J_{\mathrm{A}}(\boldsymbol{\theta}, \sigma_1) - J_{\mathrm{A}}(\boldsymbol{\theta}, \sigma_2)| \tag{18}$$

$$= \left| \underset{\bar{\boldsymbol{\epsilon}} \sim \Phi_{d_{\mathcal{A}}, 1}^T}{\mathbb{E}} \left[ J_{\mathrm{D}}(\boldsymbol{\mu}_{\boldsymbol{\theta}} + \sigma_1 \bar{\boldsymbol{\epsilon}}) - J_{\mathrm{D}}(\boldsymbol{\mu}_{\boldsymbol{\theta}} + \sigma_2 \bar{\boldsymbol{\epsilon}}) \right] \right| \tag{19}$$

$$\leqslant \underset{\bar{\boldsymbol{\epsilon}} \sim \Phi_{d_{\mathcal{A}}, 1}^T}{\mathbb{E}} \left[ \left| J_{\mathrm{D}}(\boldsymbol{\mu}_{\boldsymbol{\theta}} + \sigma_1 \bar{\boldsymbol{\epsilon}}) - J_{\mathrm{D}}(\boldsymbol{\mu}_{\boldsymbol{\theta}} + \sigma_2 \bar{\boldsymbol{\epsilon}}) \right| \right], \tag{20}$$

where we employed Jensen's inequality.

Now, by applying Assumption 5.2, we obtain:

$$|J_{\mathrm{A}}(\boldsymbol{\theta}, \sigma_1) - J_{\mathrm{A}}(\boldsymbol{\theta}, \sigma_2)| \tag{21}$$

$$\leqslant \underset{\bar{\boldsymbol{\epsilon}} \sim \Phi_{d_{\mathcal{A}}, 1}^T}{\mathbb{E}} \left[ \left| J_{\mathrm{D}}(\boldsymbol{\mu}_{\boldsymbol{\theta}} + \sigma_1 \bar{\boldsymbol{\epsilon}}) - J_{\mathrm{D}}(\boldsymbol{\mu}_{\boldsymbol{\theta}} + \sigma_2 \bar{\boldsymbol{\epsilon}}) \right| \right] \tag{22}$$

$$\leqslant \underset{\bar{\boldsymbol{\epsilon}} \sim \Phi_{d_{\mathcal{A}}, 1}^T}{\mathbb{E}} \left[ \sum_{t=0}^{T-1} L_t \sup_{s \in \mathcal{S}} \left\| \boldsymbol{\mu}_{\boldsymbol{\theta}}(s) + \sigma_1 \bar{\boldsymbol{\epsilon}} - \boldsymbol{\mu}_{\boldsymbol{\theta}}(s) - \sigma_2 \bar{\boldsymbol{\epsilon}} \right\|_2 \right] \tag{23}$$

$$\leqslant \sum_{t=0}^{T-1} L_t \underset{\bar{\boldsymbol{\epsilon}} \sim \Phi_{d_{\mathcal{A}}, 1}}{\mathbb{E}} \left[ \left\| \sigma_1 \bar{\boldsymbol{\epsilon}} - \sigma_2 \bar{\boldsymbol{\epsilon}} \right\|_2 \right] \tag{24}$$

$$\leqslant \sum_{t=0}^{T-1} L_t \sqrt{\underset{\bar{\boldsymbol{\epsilon}} \sim \Phi_{d_{\mathcal{A}}, 1}}{\mathbb{E}} \left[ \left\| \sigma_1 \bar{\boldsymbol{\epsilon}} - \sigma_2 \bar{\boldsymbol{\epsilon}} \right\|_2^2 \right]} \tag{25}$$

$$\leqslant L_{1, \boldsymbol{\mu}} \sqrt{d_{\mathcal{A}}} |\sigma_1 - \sigma_2|, \tag{26}$$

where we applied the Cauchy-Schwartz inequality and Definition 2.1

**Result** $(ii)$. Here, we exploit the result $(i)$. In particular, we sum and subtract the quantity $J_{\dagger}(\boldsymbol{\theta}^*(\sigma_1), \sigma_2)$, and then we exploit the fact that $J_{\dagger}(\boldsymbol{\theta}^*(\sigma_2), \sigma_2) \geqslant J_{\dagger}(\boldsymbol{\theta}^*(\sigma_1), \sigma_2)$. We start by summing and subtracting the quantity $J_{\dagger}(\boldsymbol{\theta}^*(\sigma_1), \sigma_2)$.

$$J_{\dagger}^*(\sigma_1) - J_{\dagger}(\boldsymbol{\theta}^*(\sigma_2), \sigma_1) \tag{27}$$

$$= J_{\dagger}(\boldsymbol{\theta}^*(\sigma_1), \sigma_1) - J_{\dagger}(\boldsymbol{\theta}^*(\sigma_2), \sigma_1) \tag{28}$$

$$= J_{\dagger}(\boldsymbol{\theta}^*(\sigma_1), \sigma_1) - J_{\dagger}(\boldsymbol{\theta}^*(\sigma_2), \sigma_1) \pm J_{\dagger}(\boldsymbol{\theta}^*(\sigma_1), \sigma_2). \tag{29}$$

Now, we exploit the fact that $J_{\dagger}(\boldsymbol{\theta}^*(\sigma_2), \sigma_2) \geqslant J_{\dagger}(\boldsymbol{\theta}^*(\sigma_1), \sigma_2)$.

$$J_{\dagger}^*(\sigma_1) - J_{\dagger}(\boldsymbol{\theta}^*(\sigma_2), \sigma_1) \tag{30}$$

$$\leqslant J_{\dagger}(\boldsymbol{\theta}^*(\sigma_1), \sigma_1) - J_{\dagger}(\boldsymbol{\theta}^*(\sigma_1), \sigma_2) + J_{\dagger}(\boldsymbol{\theta}^*(\sigma_2), \sigma_2) - J_{\dagger}(\boldsymbol{\theta}^*(\sigma_2), \sigma_1). \tag{31}$$

Now, we apply twice result $(i)$, obtaining:

$$J_{\dagger}^*(\sigma_1) - J_{\dagger}(\boldsymbol{\theta}^*(\sigma_2), \sigma_1) \leqslant 2 D_{\dagger} |\sigma_1 - \sigma_2|. \tag{32}$$

**Result** $(iii)$. To prove this result, we separately consider the cases in which $J_{\dagger}(\boldsymbol{\theta}^*(\sigma_1), \sigma_1) \geqslant J_{\dagger}(\boldsymbol{\theta}^*(\sigma_2), \sigma_2)$ or $J_{\dagger}(\boldsymbol{\theta}^*(\sigma_1), \sigma_1) \leqslant J_{\dagger}(\boldsymbol{\theta}^*(\sigma_2), \sigma_2)$.

In the first case, we consider $J_{\dagger}(\boldsymbol{\theta}^*(\sigma_1), \sigma_1) \geqslant J_{\dagger}(\boldsymbol{\theta}^*(\sigma_2), \sigma_2)$. Thus, we have what follows:

$$J_{\dagger}(\boldsymbol{\theta}^*(\sigma_1), \sigma_1) - J_{\dagger}(\boldsymbol{\theta}^*(\sigma_2), \sigma_2) \tag{33}$$

$$= J_{\dagger}(\boldsymbol{\theta}^*(\sigma_1), \sigma_1) - J_{\dagger}(\boldsymbol{\theta}^*(\sigma_2), \sigma_2) \pm J_{\dagger}(\boldsymbol{\theta}^*(\sigma_1), \sigma_2) \tag{34}$$

$$= J_{\dagger}(\boldsymbol{\theta}^*(\sigma_1), \sigma_1) - J_{\dagger}(\boldsymbol{\theta}^*(\sigma_1), \sigma_2) + \underbrace{J_{\dagger}(\boldsymbol{\theta}^*(\sigma_1), \sigma_2) - J_{\dagger}(\boldsymbol{\theta}^*(\sigma_2), \sigma_2)}_{\leqslant 0} \tag{35}$$

$$\leqslant J_\dagger(\boldsymbol{\theta}^*(\sigma_1), \sigma_1) - J_\dagger(\boldsymbol{\theta}^*(\sigma_1), \sigma_2). \tag{36}$$

Now, we use result $(i)$, obtaining:

$$J_\dagger(\boldsymbol{\theta}^*(\sigma_1), \sigma_1) - J_\dagger(\boldsymbol{\theta}^*(\sigma_2), \sigma_2) \leqslant D_\dagger |\sigma_1 - \sigma_2|. \tag{37}$$

The remaining case is $J_\dagger(\boldsymbol{\theta}^*(\sigma_1), \sigma_1) \leqslant J_\dagger(\boldsymbol{\theta}^*(\sigma_2), \sigma_2)$. With a similar result, we have:

$$J_\dagger(\boldsymbol{\theta}^*(\sigma_2), \sigma_2) - J_\dagger(\boldsymbol{\theta}^*(\sigma_1), \sigma_1) \tag{38}$$

$$= J_\dagger(\boldsymbol{\theta}^*(\sigma_2), \sigma_2) - J_\dagger(\boldsymbol{\theta}^*(\sigma_1), \sigma_1) \pm J_\dagger(\boldsymbol{\theta}^*(\sigma_2), \sigma_1) \tag{39}$$

$$= J_\dagger(\boldsymbol{\theta}^*(\sigma_2), \sigma_2) - J_\dagger(\boldsymbol{\theta}^*(\sigma_2), \sigma_1) + \underbrace{J_\dagger(\boldsymbol{\theta}^*(\sigma_2), \sigma_1) - J_\dagger(\boldsymbol{\theta}^*(\sigma_1), \sigma_1)}_{\leqslant 0} \tag{40}$$

$$\leqslant J_\dagger(\boldsymbol{\theta}^*(\sigma_2), \sigma_2) - J_\dagger(\boldsymbol{\theta}^*(\sigma_2), \sigma_1). \tag{41}$$

As for the previous case, we use result $(i)$, obtaining:

$$J_\dagger(\boldsymbol{\theta}^*(\sigma_2), \sigma_2) - J_\dagger(\boldsymbol{\theta}^*(\sigma_1), \sigma_1) \leqslant D_\dagger |\sigma_1 - \sigma_2|. \tag{42}$$

$$\square$$

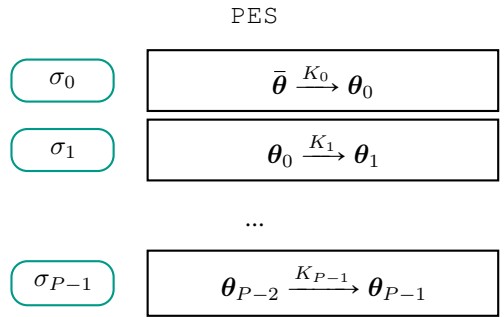

*Figure 2.* Graphical representation of PES.

# F. Proofs of Section 6

**Theorem 6.1** (PES Convergence). *Employing a (hyper)policy complying with Definitions 2.2 (PB) or 2.3 (AB), under Assumptions 5.1 (PB) or 5.2 (AB), 5.3, and under the assumptions for the convergence of PG subroutines (Sec. 4), if PES is run for $P = (\epsilon/\sigma_{\max})^{-1/y}$ phases, the output parameterization $\boldsymbol{\theta}_{P-1}$ is such that:*

$$J_D^* - \mathbb{E}[J_D(\boldsymbol{\theta}_{P-1})] \leqslant (1 + 2D_\dagger + \mathcal{W}_\dagger)\epsilon + \beta_D,$$

*with a sample complexity:*

$$NK \geqslant \tilde{\mathcal{O}}\left(\frac{L_2 \alpha_D^2 \mathcal{V}_{\dagger,\boldsymbol{\theta}}(2D_\dagger + \mathcal{W}_\dagger)}{\epsilon^5}\right),$$

*where $K = \sum_{p=0}^{P-1} K_p$.*

*Proof.* **Outline.** To asses the converge of PES we need to apply:

- Theorem F.1 by Montenegro et al. (2024) to every phase $p \in [\![0, P-1]\!]$ to ensure the last-iterate global convergence to the $p^{\text{th}}$ phase's optimum;
- Theorem 5.1 to quantify the distance between $J_\dagger(\boldsymbol{\theta}^*(\sigma_p), \sigma_p) - \mathbb{E}[J_\dagger(\boldsymbol{\theta}_{p-1}, \sigma_p)]$ each time we initialize a phase $p$ with the output of the preceding one. To use this theorem, we need to be under Assumptions 5.1 (PB) or 5.2 (AB) and Assumption 5.3. Moreover, the employed (hyper)policy has to be compliant with Definitions 2.2 and 2.3.

In the following, we use $\bar{\boldsymbol{\theta}}$ to indicate the initialization for the parameters and we define $\boldsymbol{\theta}_{-1} := \bar{\boldsymbol{\theta}}$.

**Convergence of PG Subroutines.** In order to apply Montenegro et al. (Theorem F.1, 2024) to every phase $p \in [\![0, P-1]\!]$, the following conditions must hold:

- The objective $J_\dagger(\cdot, \sigma_p)$ has to be $L_2$-LS w.r.t. the parameterization $\boldsymbol{\theta}$. This is obtained under Assumption 4.1 (Lemma 4.1);
- WGD holds for $J_\dagger(\boldsymbol{\theta}, \sigma_p)$ w.r.t. $\boldsymbol{\theta}$. Under Assumption 4.2 and other regularity assumptions on the deterministic policy and on the MDP (see Appendix D), this can be obtained (Thr. 4.2) as:

$$J_\dagger(\boldsymbol{\theta}^*(\sigma_p), \sigma_p) - J_\dagger(\boldsymbol{\theta}, \sigma_p) \leqslant \alpha_D \|\nabla_{\boldsymbol{\theta}} J_\dagger(\boldsymbol{\theta}, \sigma_p)\|_2 + \beta_\dagger(\sigma_p),$$

with $\beta_\dagger(\sigma_p) := \beta_D + \mathcal{W}_\dagger \sigma_p$;

- Bounded variance of the estimators, which is obtained under Assumption 4.4 and other regularity assumptions on the deterministic policy and on the MDP (see Appendix D and Lemma 4.3) as:

$$\mathbb{V}\mathrm{ar}[\hat{\nabla}_{\boldsymbol{\theta}} J_\dagger(\boldsymbol{\theta}, \sigma_p)] \leqslant \frac{\mathcal{V}_{\dagger,\boldsymbol{\theta}}}{N\sigma_p^2}.$$

We recall that the constant $\mathcal{V}_{\dagger,\boldsymbol{\theta}}$ does not depend on the quantity $\sigma$, as reported in Lemma 4.3.

Being in a generic phase $p \in [\![0, P-1]\!]$, we can apply Theorem F.1 by Montenegro et al. (2024) to assess the minimum number of iterations $K_p$ needed to ensure last-iterate global convergence of the $p^{\text{th}}$ PG subroutine. In particular, running the PG subroutine with a constant learning rate

$$\zeta_p = \frac{\epsilon^2 N \sigma_p^2}{4\alpha_D^2 L_2 \mathcal{V}_{\dagger,\boldsymbol{\theta}}} \tag{43}$$

for $K_p$ iterations such that

$$K_p \geqslant \frac{16\alpha_{\mathrm{D}}^4 L_2 \mathcal{V}_{\dagger,\boldsymbol{\theta}}}{N\epsilon^3 \sigma_p^2} \log \frac{(J_{\dagger}(\boldsymbol{\theta}^*(\sigma_p), \sigma_p) - \mathbb{E}[J_{\dagger}(\boldsymbol{\theta}_{p-1}, \sigma_p)] - \beta_{\dagger}(\sigma_p))^+}{\epsilon}, \tag{44}$$

then the output parameterization $\boldsymbol{\theta}_p$ is such that

$$J_{\dagger}(\boldsymbol{\theta}^*(\sigma_p), \sigma_p) - \mathbb{E}[J_{\dagger}(\boldsymbol{\theta}_p, \sigma_p)] \leqslant \epsilon + \beta_{\dagger}(\sigma_p). \tag{45}$$

**Total Iteration Complexity.** The total amount of iterations don by PES is given by:

$$K = \sum_{i=0}^{P-1} K_i. \tag{46}$$

By requiring each step size $\zeta_i$ to satisfy Equation (43) and $K_i$ to satisfy Equation (44), then each PG subroutine enjoys last-iterate convergence guarantees as Equation (45). Thus, we can require the following condition on $K$:

$$K = \sum_{i=0}^{P-1} K_i \tag{47}$$

$$= K_0 + \sum_{i=1}^{P-1} K_i \tag{48}$$

$$\geqslant \frac{16\alpha_{\mathrm{D}}^4 L_2 \mathcal{V}_{\dagger,\boldsymbol{\theta}}}{N\epsilon^3 \sigma_{\max}^2} \log \frac{(J_{\dagger}(\boldsymbol{\theta}^*(\sigma_{\max}), \sigma_{\max}) - J_{\dagger}(\bar{\boldsymbol{\theta}}, \sigma_{\max}) - \beta_{\dagger}(\sigma_{\max}))^+}{\epsilon} \tag{49}$$

$$+ \sum_{i=1}^{P-1} \frac{16\alpha_{\mathrm{D}}^4 L_2 \mathcal{V}_{\dagger,\boldsymbol{\theta}}}{N\epsilon^3 \sigma_i^2} \log \left( \frac{(J_{\dagger}(\boldsymbol{\theta}^*(\sigma_i), \sigma_i) - \mathbb{E}[J_{\dagger}(\boldsymbol{\theta}_{i-1}, \sigma_i)] - \beta_{\dagger}(\sigma_i))^+}{\epsilon} \right). \tag{50}$$

By construction of PES, and by the fact that each PG subroutine enjoys last-iterate convergence guarantees, we can quantify the term $J_{\dagger}(\boldsymbol{\theta}^*(\sigma_i), \sigma_i) - \mathbb{E}[J_{\dagger}(\boldsymbol{\theta}_{i-1}, \sigma_i)] - \beta_{\dagger}(\sigma_i)$. Indeed, except for the first phase $i = 0$, we have:

$$J_{\dagger}(\boldsymbol{\theta}^*(\sigma_i), \sigma_i) - \mathbb{E}[J_{\dagger}(\boldsymbol{\theta}_{i-1}, \sigma_i)] - \beta_{\dagger}(\sigma_i) \tag{51}$$

$$\leqslant J_{\dagger}(\boldsymbol{\theta}^*(\sigma_i), \sigma_i) - \mathbb{E}[J_{\dagger}(\boldsymbol{\theta}_{i-1}, \sigma_{i-1})] + D_{\dagger}|\sigma_i - \sigma_{i-1}| - \beta_{\dagger}(\sigma_i) \tag{52}$$

$$\leqslant J_{\dagger}(\boldsymbol{\theta}^*(\sigma_i), \sigma_i) - J_{\dagger}(\boldsymbol{\theta}^*(\sigma_{i-1}), \sigma_{i-1}) + D_{\dagger}|\sigma_i - \sigma_{i-1}| + \epsilon + \beta_{\dagger}(\sigma_{i-1}) - \beta_{\dagger}(\sigma_i) \tag{53}$$

$$\leqslant 2D_{\dagger}|\sigma_i - \sigma_{i-1}| + \epsilon + \beta_{\dagger}(\sigma_{i-1}) - \beta_{\dagger}(\sigma_i) \tag{54}$$

$$= 2D_{\dagger}(\sigma_{i-1} - \sigma_i) + \epsilon + \beta_{\dagger}(\sigma_{i-1}) - \beta_{\dagger}(\sigma_i), \tag{55}$$

where we employed Theorem 5.1, we considered that PES employs a non-increasing $\sigma_i$ schedule, and we exploited the fact that each final parameterization of each phase satisfies Equation (45).

Moreover, since $\beta_{\dagger}(\sigma_i) = \beta_{\mathrm{D}} + \mathcal{W}_{\dagger}\sigma_i$, we have the following:

$$J_{\dagger}(\boldsymbol{\theta}^*(\sigma_i), \sigma_i) - \mathbb{E}[J_{\dagger}(\boldsymbol{\theta}_{i-1}, \sigma_i)] - \beta_{\dagger}(\sigma_i) \leqslant (2D_{\dagger} + \mathcal{W}_{\dagger})(\sigma_{i-1} - \sigma_i) + \epsilon. \tag{56}$$

Exploiting this last result, we can enforce a stronger but simpler condition to hold for $K$:

$$K \geqslant \frac{16\alpha_{\mathrm{D}}^4 L_2 \mathcal{V}_{\dagger,\boldsymbol{\theta}}}{N\epsilon^3 \sigma_{\max}^2} \log \frac{(J_{\dagger}(\boldsymbol{\theta}^*(\sigma_{\max}), \sigma_{\max}) - J_{\dagger}(\bar{\boldsymbol{\theta}}, \sigma_{\max}) - \beta_{\dagger}(\sigma_{\max}))^+}{\epsilon} \tag{57}$$

$$+ \sum_{i=1}^{P-1} \frac{16\alpha_{\mathrm{D}}^4 L_2 \mathcal{V}_{\dagger,\boldsymbol{\theta}}}{N\epsilon^3 \sigma_i^2} \log \left( 1 + \frac{(2D_{\dagger} + \mathcal{W}_{\dagger})(\sigma_{i-1} - \sigma_i)}{\epsilon} \right). \tag{58}$$

Now, considering the specific $\sigma_i$ schedule employed by PES, we have that:

$$\sigma_{i-1} - \sigma_i = \sigma_{\max} \left( i^{-y} - (i+1)^{-y} \right), \tag{59}$$

thus recovering the following condition on the iteration complexity:

$$K \geqslant \frac{16\alpha_{\mathrm{D}}^4 L_2 \mathcal{V}_{\dagger,\boldsymbol{\theta}}}{N\epsilon^3 \sigma_{\max}^2} \log \frac{(J_{\dagger}(\boldsymbol{\theta}^*(\sigma_{\max}), \sigma_{\max}) - J_{\dagger}(\bar{\boldsymbol{\theta}}, \sigma_{\max}) - \beta_{\dagger}(\sigma_{\max}))^+}{\epsilon} \tag{60}$$

$$+ \sum_{i=1}^{P-1} \frac{16\alpha_{\mathrm{D}}^4 L_2 \mathcal{V}_{\dagger,\boldsymbol{\theta}}}{N\epsilon^3 \sigma_i^2} \log\left(1 + \frac{\sigma_{\max}(2D_\dagger + \mathcal{W}_\dagger)\left(i^{-y} - (i+1)^{-y}\right)}{\epsilon}\right). \tag{61}$$

Before going on with the proof, we notice that $x \geqslant \log(1+x)$ for any $x > -1$. We shall apply this with $x = \frac{\sigma_0(2D_\dagger + \mathcal{W}_\dagger)(\sigma_{i-1} - \sigma_i)}{\epsilon}$, which is positive. In light of this, we enforce an even stronger condition to hold for $K$:

$$K \geqslant \frac{16\alpha_{\mathrm{D}}^4 L_2 \mathcal{V}_{\dagger,\boldsymbol{\theta}}}{N\epsilon^3 \sigma_{\max}^2} \log \frac{(J_\dagger(\boldsymbol{\theta}^*(\sigma_{\max}), \sigma_{\max}) - J_\dagger(\bar{\boldsymbol{\theta}}, \sigma_{\max}) - \beta_\dagger(\sigma_{\max}))^+}{\epsilon} \tag{62}$$

$$+ \frac{16\alpha_{\mathrm{D}}^4 L_2 \mathcal{V}_{\dagger,\boldsymbol{\theta}}(2D_\dagger + \mathcal{W}_\dagger)}{N\epsilon^4 \sigma_{\max}} \sum_{i=1}^{P-1} \frac{i^{-y} - (i+1)^{-y}}{(i+1)^{-2y}}, \tag{63}$$

where we also substituted the value $\sigma_i^{-2}$ with $\sigma_{\max}^{-2}(i+1)^{2y}$, given the $\sigma$ schedule employed by PES.

Now, we calculate the convergence value of the summation:

$$\sum_{i=1}^{P-1} \frac{(i^{-y} - (i+1)^{-y})}{(i+1)^{-2y}} \tag{64}$$

$$= \sum_{i=1}^{P-1} ((i+1)^y - i^y)\left(\frac{i+1}{i}\right)^y \tag{65}$$

$$\leqslant 2^y \sum_{i=1}^{P-1} ((i+1)^y - i^y) \tag{66}$$

$$= 2^y(P^y - 1) \leqslant 2^y P^y. \tag{67}$$

This result suggests a particular choice for the number of phases $P$ and exponent $y$. Indeed, we select these two quantities such that $P^y = \sigma_{\max}/\epsilon$ since this choice leads to a $\sigma_{P-1} = \epsilon$. Having made such a choice, we can write:

$$\sum_{i=1}^{P-1} \frac{(i^{-y} - (i+1)^{-y})}{(i+1)^{-2y}} \leqslant 2^y \frac{\sigma_{\max}}{\epsilon}. \tag{68}$$

Considering $P^y = \sigma_{\max}/\epsilon$, we can require the total iteration complexity to satisfy the even stronger condition:

$$K \geqslant \frac{16\alpha_{\mathrm{D}}^4 L_2 \mathcal{V}_{\dagger,\boldsymbol{\theta}}}{N\epsilon^3 \sigma_{\max}^2} \log \frac{(J_\dagger(\boldsymbol{\theta}^*(\sigma_{\max}), \sigma_{\max}) - J_\dagger(\bar{\boldsymbol{\theta}}, \sigma_{\max}) - \beta_\dagger(\sigma_{\max}))^+}{\epsilon} + \frac{16\alpha_{\mathrm{D}}^4 L_2 \mathcal{V}_{\dagger,\boldsymbol{\theta}}(2D_\dagger + \mathcal{W}_\dagger)}{N\epsilon^5}. \tag{69}$$

The latter is the number of iterations ensuring, together with the choices of learning rates $(\zeta_i)_{i=0}^{P-1}$ (Eq. 43), the $(\sigma_i)_{i=0}^{P-1}$ schedule, and the number of phases $P = (\epsilon/\sigma_{\max})^{-1/y}$, that the final parameterization satisfies Equation (45):

$$J_\dagger(\boldsymbol{\theta}^*(\sigma_{P-1}), \sigma_{P-1}) - \mathbb{E}[J_\dagger(\boldsymbol{\theta}_{P-1}, \sigma_{P-1})] \leqslant \epsilon + \beta_\dagger(\sigma_{P-1}). \tag{70}$$

In particular, since $\sigma_{P-1} = \epsilon$, we have the following:

$$J_\dagger(\boldsymbol{\theta}^*(\sigma_{P-1}), \sigma_{P-1}) - \mathbb{E}[J_\dagger(\boldsymbol{\theta}_{P-1}, \sigma_{P-1})] \tag{71}$$
$$= J_\dagger(\boldsymbol{\theta}^*(\epsilon), \epsilon) - \mathbb{E}[J_\dagger(\boldsymbol{\theta}_{P-1}, \epsilon)] \leqslant \epsilon + \beta_\dagger(\epsilon). \tag{72}$$

Finally, to prove the convergence to deterministic global optimum, we use the results of Theorem 5.1:

$$J_\dagger(\boldsymbol{\theta}^*(\epsilon), \epsilon) - \mathbb{E}[J_\dagger(\boldsymbol{\theta}_{P-1}, \epsilon)] \tag{73}$$
$$\geqslant J_\dagger(\boldsymbol{\theta}^*(\epsilon), \epsilon) - \mathbb{E}[J_{\mathrm{D}}(\boldsymbol{\theta}_{P-1})] - D_\dagger\epsilon \tag{74}$$
$$= J_{\mathrm{D}}(\boldsymbol{\theta}^*(0)) - \mathbb{E}[J_{\mathrm{D}}(\boldsymbol{\theta}_{P-1})] + J_\dagger(\boldsymbol{\theta}^*(\epsilon), \epsilon) - J_{\mathrm{D}}(\boldsymbol{\theta}^*(0)) - D_\dagger\epsilon \tag{75}$$
$$\geqslant J_{\mathrm{D}}(\boldsymbol{\theta}^*(0)) - \mathbb{E}[J_{\mathrm{D}}(\boldsymbol{\theta}_{P-1})] - 2D_\dagger\epsilon. \tag{76}$$

All in all, we have:

$$J_{\mathrm{D}}^* - \mathbb{E}[J_{\mathrm{D}}(\boldsymbol{\theta}_{P-1})] \leqslant (1 + 2D_\dagger)\epsilon + \beta_\dagger(\epsilon) = (1 + 2D_\dagger + \mathcal{W}_\dagger)\epsilon + \beta_{\mathrm{D}}, \tag{77}$$

with a total sample complexity $NK = \mathcal{O}(\epsilon^{-5})$. $\qquad\square$

# G. Proofs of Section 7

## G.1. `SL-PG` Stochasticity Estimators

Here, we present the estimators to update $\xi$ in `SL-PG`. It is worth noticing that they have a similar form to the ones presented in Section 2, but the scores of the (hyper)policies has to be computed w.r.t. $\xi$. In the following, we consider $\sigma = f(\xi)$.



**Parameter-based**


$$\widehat{\nabla}_\xi J_{\mathrm{P}}(\boldsymbol{\theta}, \sigma) = \frac{1}{N} \sum_{i=1}^{N} \frac{\partial}{\partial \xi} \log \nu_{\boldsymbol{\theta},\sigma}(\boldsymbol{\theta}_i) R(\tau_i)$$



**Action-based**


$$\widehat{\nabla}_\xi J_{\mathrm{A}}(\boldsymbol{\theta}, \sigma) := \frac{1}{N} \sum_{i=1}^{N} \sum_{t=0}^{T-1} \left( \sum_{k=0}^{t} \frac{\partial}{\partial \xi} \log \pi_{\boldsymbol{\theta},\sigma}(\mathbf{a}_{\tau_i,k} | \mathbf{s}_{\tau_i,k}) \right) \gamma^t r(\mathbf{s}_{\tau_i,t}, \mathbf{a}_{\tau_i,t})$$

## G.2. Inherited Smoothness of $J_\dagger$ w.r.t. the Stochasticity Parameterization

**Lemma G.1.** *($J_P$ Inherited Smoothness w.r.t. $\sigma$) If the hyperpolicy satisfies Definition 2.3, under Assumptions 4.1 and 5.3, for every $\boldsymbol{\theta} \in \mathbb{R}^{d_\Theta}$ and $\sigma \in \mathbb{R}^+$, the second derivative of $J_P$ w.r.t. $\sigma$ is bounded as follows:*

$$\left| \frac{\partial^2}{\partial \sigma^2} J_P(\boldsymbol{\theta}, \sigma) \right| \leqslant d_\Theta L_2.$$

*Proof.* Given the fact that the hyperpolicy $\nu_{\boldsymbol{\theta},\sigma}$ satisfies Definition 2.3 and that Assumption 5.3 holds, we can write the following:

$$\frac{\partial^2}{\partial \sigma^2} J_{\mathrm{P}}(\boldsymbol{\theta}, \sigma) = \frac{\partial^2}{\partial \sigma^2} \mathbb{E}_{\boldsymbol{\epsilon} \sim \Phi_{d_\Theta,1}} \left[ J_{\mathrm{D}}(\boldsymbol{\theta} + \sigma \boldsymbol{\epsilon}) \right] \tag{78}$$

$$= \mathbb{E}_{\boldsymbol{\epsilon} \sim \Phi_{d_\Theta,1}} \left[ \frac{\partial^2}{\partial \sigma^2} J_{\mathrm{D}}(\boldsymbol{\theta} + \sigma \boldsymbol{\epsilon}) \right]. \tag{79}$$

We now need to apply the chain rule to the term $\frac{\partial^2}{\partial \sigma^2} J_{\mathrm{D}}(\boldsymbol{\theta} + \sigma \boldsymbol{\epsilon})$:

$$\frac{\partial^2}{\partial \sigma^2} J_{\mathrm{D}}(\boldsymbol{\theta} + \sigma \boldsymbol{\epsilon}) = \frac{\partial}{\partial \sigma} \left( \frac{\partial}{\partial \sigma} J_{\mathrm{D}}(\boldsymbol{\theta} + \sigma \boldsymbol{\epsilon}) \right) \tag{80}$$

$$= \frac{\partial}{\partial \sigma} \left( \nabla_{\boldsymbol{\eta}} \left. J_{\mathrm{D}}(\boldsymbol{\eta}) \right|_{\boldsymbol{\eta} = \boldsymbol{\theta} + \sigma \boldsymbol{\epsilon}} \frac{\partial (\boldsymbol{\theta} + \sigma \boldsymbol{\epsilon})}{\partial \sigma} \right) \tag{81}$$

$$= \frac{\partial}{\partial \sigma} \left( \nabla_{\boldsymbol{\eta}} \left. J_{\mathrm{D}}(\boldsymbol{\eta}) \right|_{\boldsymbol{\eta} = \boldsymbol{\theta} + \sigma \boldsymbol{\epsilon}} \boldsymbol{\epsilon} \right) \tag{82}$$

$$= \boldsymbol{\epsilon}^\top \nabla_{\boldsymbol{\eta}}^2 \left. J_{\mathrm{D}}(\boldsymbol{\eta}) \right|_{\boldsymbol{\eta} = \boldsymbol{\theta} + \sigma \boldsymbol{\epsilon}} \boldsymbol{\epsilon}. \tag{83}$$

Thus, applying the absolute value to the quantity $\frac{\partial^2}{\partial \sigma^2} J_{\mathrm{P}}(\boldsymbol{\theta}, \sigma)$, and recalling that Assumption 4.1 holds, we have what follows:

$$\left| \frac{\partial^2}{\partial \sigma^2} J_{\mathrm{P}}(\boldsymbol{\theta}, \sigma) \right| \leqslant \mathbb{E}_{\boldsymbol{\epsilon} \sim \Phi_{d_\Theta,1}} \left[ \left| \frac{\partial^2}{\partial \sigma^2} J_{\mathrm{D}}(\boldsymbol{\theta} + \sigma \boldsymbol{\epsilon}) \right| \right] \tag{84}$$

$$= \mathbb{E}_{\boldsymbol{\epsilon} \sim \Phi_{d_\Theta,1}} \left[ \left| \boldsymbol{\epsilon}^\top \nabla_{\boldsymbol{\eta}}^2 \left. J_{\mathrm{D}}(\boldsymbol{\eta}) \right|_{\boldsymbol{\eta} = \boldsymbol{\theta} + \sigma \boldsymbol{\epsilon}} \boldsymbol{\epsilon} \right| \right] \tag{85}$$

$$\leqslant L_2 \mathbb{E}_{\boldsymbol{\epsilon} \sim \Phi_{d_\Theta,1}} \left[ \|\boldsymbol{\epsilon}\|_2^2 \right] \tag{86}$$

$$\leqslant L_2 d_\Theta. \tag{87}$$

$$\square$$

**Lemma G.2.** *($J_P$ Inherited Smoothness w.r.t. $\xi$) If the hyperpolicy satisfies Definition 2.3, under Assumptions 4.1, 7.1, 5.1,*

and 5.3, for every $\boldsymbol{\theta} \in \mathbb{R}^{d_\Theta}$ and $\sigma \in \mathbb{R}^+$, the second derivative of $J_P$ w.r.t. $\xi$ is bounded as follows:

$$\left| \frac{\partial^2}{\partial \xi^2} J_P(\boldsymbol{\theta}, f(\xi)) \right| \leqslant L_{2P,\xi}.$$

where $L_{2P,\xi} = L_2 L_{1,f}^2 d_\Theta + L_J L_{2,f} \sqrt{d_\Theta}$.

*Proof.* The proof is similar to the one of Lemma G.1.

We start by exploiting Definition 2.3 and Assumption 5.3:

$$\frac{\partial^2}{\partial \xi^2} J_P(\boldsymbol{\theta}, f(\xi)) = \frac{\partial^2}{\partial \xi^2} \mathop{\mathbb{E}}_{\boldsymbol{\epsilon} \sim \Phi_{d_\Theta, 1}} [J_D(\boldsymbol{\theta} + f(\xi)\boldsymbol{\epsilon})] \tag{88}$$

$$= \mathop{\mathbb{E}}_{\boldsymbol{\epsilon} \sim \Phi_{d_\Theta, 1}} \left[ \frac{\partial^2}{\partial \xi^2} J_D(\boldsymbol{\theta} + f(\xi)\boldsymbol{\epsilon}) \right]. \tag{89}$$

We now need to apply the chain rule to the term $\frac{\partial^2}{\partial \xi^2} J_D(\boldsymbol{\theta} + f(\xi)\boldsymbol{\epsilon})$:

$$\frac{\partial^2}{\partial \xi^2} J_D(\boldsymbol{\theta} + f(\xi)\boldsymbol{\epsilon}) = \frac{\partial}{\partial \xi} \left( \frac{\partial}{\partial \xi} J_D(\boldsymbol{\theta} + f(\xi)\boldsymbol{\epsilon}) \right) \tag{90}$$

$$= \frac{\partial}{\partial \xi} \left( \nabla_{\boldsymbol{\eta}} J_D(\boldsymbol{\eta})|_{\boldsymbol{\eta} = \boldsymbol{\theta} + f(\xi)\boldsymbol{\epsilon}} \frac{\partial(\boldsymbol{\theta} + f(\xi)\boldsymbol{\epsilon})}{\partial \xi} \right) \tag{91}$$

$$= \frac{\partial}{\partial \xi} \left( \nabla_{\boldsymbol{\eta}} J_D(\boldsymbol{\eta})|_{\boldsymbol{\eta} = \boldsymbol{\theta} + f(\xi)\boldsymbol{\epsilon}} \boldsymbol{\epsilon} \frac{\partial}{\partial \xi} f(\xi) \right) \tag{92}$$

$$= \frac{\partial}{\partial \xi} \left( \nabla_{\boldsymbol{\eta}} J_D(\boldsymbol{\eta})|_{\boldsymbol{\eta} = \boldsymbol{\theta} + f(\xi)\boldsymbol{\epsilon}} \right) \boldsymbol{\epsilon} \frac{\partial}{\partial \xi} f(\xi) + \nabla_{\boldsymbol{\eta}} J_D(\boldsymbol{\eta})|_{\boldsymbol{\eta} = \boldsymbol{\theta} + f(\xi)\boldsymbol{\epsilon}} \boldsymbol{\epsilon} \frac{\partial^2}{\partial \xi^2} f(\xi) \tag{93}$$

$$= \boldsymbol{\epsilon}^\top \nabla_{\boldsymbol{\eta}}^2 J_D(\boldsymbol{\eta})|_{\boldsymbol{\eta} = \boldsymbol{\theta} + f(\xi)\boldsymbol{\epsilon}} \boldsymbol{\epsilon} \frac{\partial}{\partial \xi} f(\xi)^2 + \nabla_{\boldsymbol{\eta}} J_D(\boldsymbol{\eta})|_{\boldsymbol{\eta} = \boldsymbol{\theta} + f(\xi)\boldsymbol{\epsilon}} \boldsymbol{\epsilon} \frac{\partial^2}{\partial \xi^2} f(\xi). \tag{94}$$

Thus, applying the absolute value to the quantity $\frac{\partial^2}{\partial \xi^2} J_P(\boldsymbol{\theta}, f(\xi))$, and considering Assumptions 4.1 and 7.1, we have what follows:

$$\left| \frac{\partial^2}{\partial \xi^2} J_P(\boldsymbol{\theta}, f(\xi)) \right| \tag{95}$$

$$\leqslant \mathop{\mathbb{E}}_{\boldsymbol{\epsilon} \sim \Phi_{d_\Theta, 1}} \left[ \left| \frac{\partial^2}{\partial \xi^2} J_D(\boldsymbol{\theta} + f(\xi)\boldsymbol{\epsilon}) \right| \right] \tag{96}$$

$$\leqslant \mathop{\mathbb{E}}_{\boldsymbol{\epsilon} \sim \Phi_{d_\Theta, 1}} \left[ \left| \boldsymbol{\epsilon}^\top \nabla_{\boldsymbol{\eta}}^2 J_D(\boldsymbol{\eta})|_{\boldsymbol{\eta} = \boldsymbol{\theta} + f(\xi)\boldsymbol{\epsilon}} \boldsymbol{\epsilon} \frac{\partial}{\partial \xi} f(\xi)^2 \right| \right] + \mathop{\mathbb{E}}_{\boldsymbol{\epsilon} \sim \Phi_{d_\Theta, 1}} \left[ \left| \nabla_{\boldsymbol{\eta}} J_D(\boldsymbol{\eta})|_{\boldsymbol{\eta} = \boldsymbol{\theta} + f(\xi)\boldsymbol{\epsilon}} \boldsymbol{\epsilon} \frac{\partial^2}{\partial \xi^2} f(\xi) \right| \right] \tag{97}$$

$$\leqslant L_2 L_{1,f}^2 d_\Theta + L_J L_{2,f} \sqrt{d_\Theta}. \tag{98}$$

$\square$

**Lemma G.3.** *($J_P$ Inherited LC w.r.t. $\xi$) If the hyperpolicy satisfies Definition 2.3, under Assumptions 5.1, 7.1, and 5.3, for every $\boldsymbol{\theta} \in \mathbb{R}^{d_\Theta}$ and $\sigma \in \mathbb{R}^+$, the derivative of $J_P$ w.r.t. $\xi$ is bounded as follows:*

$$\left| \frac{\partial}{\partial \xi} J_P(\boldsymbol{\theta}, f(\xi)) \right| \leqslant L_{1P,\xi}.$$

where $L_{1P,\xi} = L_J L_{1,f} \sqrt{d_\Theta}$.

*Proof.* The proof is similar to the one of Lemma G.2.

We start by exploiting Definition 2.3 and Assumption 5.3:

$$\frac{\partial}{\partial \xi} J_P(\boldsymbol{\theta}, f(\xi)) = \frac{\partial}{\partial \xi} \mathop{\mathbb{E}}_{\boldsymbol{\epsilon} \sim \Phi_{d_\Theta, 1}} [J_D(\boldsymbol{\theta} + f(\xi)\boldsymbol{\epsilon})] \tag{99}$$

$$= \mathop{\mathbb{E}}_{\boldsymbol{\epsilon} \sim \Phi_{d_\Theta, 1}} \left[ \frac{\partial}{\partial \xi} J_D(\boldsymbol{\theta} + f(\xi)\boldsymbol{\epsilon}) \right]. \tag{100}$$

We now need to apply the chain rule to the term $\frac{\partial}{\partial \xi} J_{\mathrm{D}}(\boldsymbol{\theta} + f(\xi)\boldsymbol{\epsilon})$:

$$\frac{\partial}{\partial \xi} J_{\mathrm{D}}(\boldsymbol{\theta} + f(\xi)\boldsymbol{\epsilon}) = \frac{\partial}{\partial \xi} J_{\mathrm{D}}(\boldsymbol{\theta} + f(\xi)\boldsymbol{\epsilon}) \tag{101}$$

$$= \nabla_{\boldsymbol{\eta}} \, J_{\mathrm{D}}(\boldsymbol{\eta})|_{\boldsymbol{\eta}=\boldsymbol{\theta}+f(\xi)\boldsymbol{\epsilon}} \frac{\partial(\boldsymbol{\theta} + f(\xi)\boldsymbol{\epsilon}))}{\partial \xi} \tag{102}$$

$$= \nabla_{\boldsymbol{\eta}} \, J_{\mathrm{D}}(\boldsymbol{\eta})|_{\boldsymbol{\eta}=\boldsymbol{\theta}+f(\xi)\boldsymbol{\epsilon}} \, \boldsymbol{\epsilon}\frac{\partial}{\partial \xi} f(\xi). \tag{103}$$

Thus, applying the absolute value to the quantity $\frac{\partial}{\partial \xi} J_{\mathrm{P}}(\boldsymbol{\theta}, f(\xi))$, and considering Assumptions 5.1 and 7.1, we have what follows:

$$\left| \frac{\partial}{\partial \xi} J_{\mathrm{P}}(\boldsymbol{\theta}, f(\xi)) \right| \tag{104}$$

$$\leqslant \mathop{\mathbb{E}}_{\boldsymbol{\epsilon}\sim\Phi_{d_{\Theta},1}} \left[ \left| \frac{\partial}{\partial \xi} J_{\mathrm{D}}(\boldsymbol{\theta} + f(\xi)\boldsymbol{\epsilon}) \right| \right] \tag{105}$$

$$\leqslant \mathop{\mathbb{E}}_{\boldsymbol{\epsilon}\sim\Phi_{d_{\Theta},1}} \left[ \left| \nabla_{\boldsymbol{\eta}} \, J_{\mathrm{D}}(\boldsymbol{\eta})|_{\boldsymbol{\eta}=\boldsymbol{\theta}+f(\xi)\boldsymbol{\epsilon}} \, \boldsymbol{\epsilon}\frac{\partial}{\partial \xi} f(\xi) \right| \right] \tag{106}$$

$$\leqslant L_J L_{1,f} \sqrt{d_{\Theta}}. \tag{107}$$

$\square$

**Lemma G.4** ($J_{\mathrm{P}}$ Inherited Lipschitz Gradient w.r.t. $\xi$). *If the hyperpolicy satisfies Definition 2.3, under Assumptions 4.1 and 7.1, for every $\boldsymbol{\theta}_1, \boldsymbol{\theta}_2 \in \Theta$ and $\xi \in \mathbb{R}$, it holds:*

$$\left| \frac{\partial}{\partial \xi} J_P(\boldsymbol{\theta}_1, f(\xi)) - \frac{\partial}{\partial \xi} J_P(\boldsymbol{\theta}_2, f(\xi)) \right| \leqslant L_{1,f} L_2 \sqrt{d_{\Theta}} \|\boldsymbol{\theta}_1 - \boldsymbol{\theta}_2\|_2. \tag{108}$$

*Proof.* Under Assumption 4.1, it holds that:

$$\|\nabla_{\boldsymbol{\theta}} J_{\mathrm{D}}(\boldsymbol{\theta}_1) - \nabla_{\boldsymbol{\theta}} J_{\mathrm{D}}(\boldsymbol{\theta}_2)\|_2 \leqslant L_2 \|\boldsymbol{\theta}_1 - \boldsymbol{\theta}_2\|_2. \tag{109}$$

Under Assumption 5.3, by definition of $J_{\mathrm{P}}$ using a hyperpolicy satisfying Definition 2.3, it holds:

$$J_{\mathrm{P}}(\boldsymbol{\theta}, f(\xi)) = \mathop{\mathbb{E}}_{\boldsymbol{\epsilon}\sim\Phi_{d_{\Theta},1}} \left[ J_{\mathrm{D}}(\boldsymbol{\theta} + f(\xi)\boldsymbol{\epsilon}) \right]. \tag{110}$$

Moreover, by applying the chain rule, we have:

$$\frac{\partial}{\partial \xi} J_{\mathrm{P}}(\boldsymbol{\theta}, f(\xi)) = \nabla_{\boldsymbol{\eta}} J_{\mathrm{D}}(\boldsymbol{\eta})|_{\boldsymbol{\eta}=\boldsymbol{\theta}+f(\xi)\boldsymbol{\epsilon}} \frac{\partial}{\partial \xi} f(\xi)\boldsymbol{\epsilon}. \tag{111}$$

Thus, the following derivation holds:

$$\left| \frac{\partial}{\partial \xi} J_{\mathrm{P}}(\boldsymbol{\theta}_1, f(\xi)) - \frac{\partial}{\partial \xi} J_{\mathrm{P}}(\boldsymbol{\theta}_2, f(\xi)) \right| \tag{112}$$

$$= \left| \frac{\partial}{\partial \xi} \mathop{\mathbb{E}}_{\boldsymbol{\epsilon}\sim\Phi_{d_{\Theta},1}} \left[ J_{\mathrm{D}}(\boldsymbol{\theta}_1 + f(\xi)\boldsymbol{\epsilon}) \right] - \frac{\partial}{\partial \xi} \mathop{\mathbb{E}}_{\boldsymbol{\epsilon}\sim\Phi_{d_{\Theta},1}} \left[ J_{\mathrm{D}}(\boldsymbol{\theta}_2 + f(\xi)\boldsymbol{\epsilon}) \right] \right| \tag{113}$$

$$= \left| \mathop{\mathbb{E}}_{\boldsymbol{\epsilon}\sim\Phi_{d_{\Theta},1}} \left[ \frac{\partial}{\partial \xi} J_{\mathrm{D}}(\boldsymbol{\theta}_1 + f(\xi)\boldsymbol{\epsilon}) \right] - \mathop{\mathbb{E}}_{\boldsymbol{\epsilon}\sim\Phi_{d_{\Theta},1}} \left[ \frac{\partial}{\partial \xi} J_{\mathrm{D}}(\boldsymbol{\theta}_2 + f(\xi)\boldsymbol{\epsilon}) \right] \right| \tag{114}$$

$$= \left| \mathop{\mathbb{E}}_{\boldsymbol{\epsilon}\sim\Phi_{d_{\Theta},1}} \left[ \frac{\partial}{\partial \xi} J_{\mathrm{D}}(\boldsymbol{\theta}_1 + f(\xi)\boldsymbol{\epsilon}) - \frac{\partial}{\partial \xi} J_{\mathrm{D}}(\boldsymbol{\theta}_2 + f(\xi)\boldsymbol{\epsilon}) \right] \right| \tag{115}$$

$$\leqslant \mathop{\mathbb{E}}_{\boldsymbol{\epsilon}\sim\Phi_{d_{\Theta},1}} \left[ \left| \frac{\partial}{\partial \xi} J_{\mathrm{D}}(\boldsymbol{\theta}_1 + f(\xi)\boldsymbol{\epsilon}) - \frac{\partial}{\partial \xi} J_{\mathrm{D}}(\boldsymbol{\theta}_2 + f(\xi)\boldsymbol{\epsilon}) \right| \right] \tag{116}$$

$$= \mathop{\mathbb{E}}_{\boldsymbol{\epsilon}\sim\Phi_{d_{\Theta},1}} \left[ \left| \nabla_{\boldsymbol{\eta}} J_{\mathrm{D}}(\boldsymbol{\eta}_1)|_{\boldsymbol{\eta}_1=\boldsymbol{\theta}_1+f(\xi)\boldsymbol{\epsilon}} \frac{\partial}{\partial \xi} f(\xi)\boldsymbol{\epsilon} - \nabla_{\boldsymbol{\eta}} J_{\mathrm{D}}(\boldsymbol{\eta}_2)|_{\boldsymbol{\eta}_2=\boldsymbol{\theta}_2+f(\xi)\boldsymbol{\epsilon}} \frac{\partial}{\partial \xi} f(\xi)\boldsymbol{\epsilon} \right| \right] \tag{117}$$

$$\leqslant \mathop{\mathbb{E}}_{\epsilon \sim \Phi_{d_\Theta},1} \left[ \left| \frac{\partial}{\partial \xi} f(\xi) \right| \|\epsilon\|_2 \left\| \nabla_{\boldsymbol{\eta}} J_{\mathrm{D}}(\boldsymbol{\eta}_1)|_{\boldsymbol{\eta}_1 = \boldsymbol{\theta}_1 + f(\xi)\epsilon} - \nabla_{\boldsymbol{\eta}} J_{\mathrm{D}}(\boldsymbol{\eta}_2)|_{\boldsymbol{\eta}_2 = \boldsymbol{\theta}_2 + f(\xi)\epsilon} \right\|_2 \right] \tag{118}$$

$$\leqslant L_{1,f} L_2 \sqrt{d_\Theta} \|\boldsymbol{\theta}_1 - \boldsymbol{\theta}_2\|_2, \tag{119}$$

where, in the first inequality, we employed the Jensen's inequality, and, for the last passage, we exploited Assumption 4.1. $\square$

**Lemma G.5.** ($J_A$ *Inherited Smoothness w.r.t.* $\sigma$) *If the policy satisfies Definition 2.2, under Assumption 7.2 and 5.3, for every* $\boldsymbol{\theta} \in \mathbb{R}^{d_\Theta}$ *and* $\sigma \in \mathbb{R}^+$, *the second derivative of* $J_A$ *w.r.t.* $\sigma$ *is bounded as follows:*

$$\left| \frac{\partial^2}{\partial \sigma^2} J_A(\boldsymbol{\theta}, \sigma) \right| \leqslant T d_{\mathcal{A}} L_{2,\underline{\boldsymbol{\mu}}}$$

*Proof.* The proof of this lemma follows the same reasoning of the one of Lemma G.1. The only difference is in the fact that the noise vector $\underline{\epsilon}$ has dimension $T d_{\mathcal{A}}$.

Indeed, given the fact that the policy $\pi_{\boldsymbol{\theta},\sigma}$ satisfies Definition 2.2 and that Assumption 5.3 holds, we can write the following:

$$\frac{\partial^2}{\partial \sigma^2} J_A(\boldsymbol{\theta}, \sigma) = \frac{\partial^2}{\partial \sigma^2} \mathop{\mathbb{E}}_{\underline{\epsilon} \sim \Phi_{d_{\mathcal{A}},1}^T} \left[ J_{\mathrm{D}}(\underline{\boldsymbol{\mu}}_{\boldsymbol{\theta}} + \sigma \underline{\epsilon}) \right] \tag{120}$$

$$= \mathop{\mathbb{E}}_{\underline{\epsilon} \sim \Phi_{d_{\mathcal{A}},1}^T} \left[ \frac{\partial^2}{\partial \sigma^2} J_{\mathrm{D}}(\underline{\boldsymbol{\mu}}_{\boldsymbol{\theta}} + \sigma \underline{\epsilon}) \right]. \tag{121}$$

We now need to apply the chain rule to the term $\frac{\partial^2}{\partial \sigma^2} J_{\mathrm{D}}(\underline{\boldsymbol{\mu}}_{\boldsymbol{\theta}} + \sigma \underline{\epsilon})$:

$$\frac{\partial^2}{\partial \sigma^2} J_{\mathrm{D}}(\underline{\boldsymbol{\mu}}_{\boldsymbol{\theta}} + \sigma \underline{\epsilon}) = \frac{\partial}{\partial \sigma} \left( \frac{\partial}{\partial \sigma} J_{\mathrm{D}}(\underline{\boldsymbol{\mu}}_{\boldsymbol{\theta}} + \sigma \underline{\epsilon}) \right) \tag{122}$$

$$= \frac{\partial}{\partial \sigma} \left( \nabla_{\boldsymbol{\eta}} J_{\mathrm{D}}(\boldsymbol{\eta})|_{\boldsymbol{\eta} = \underline{\boldsymbol{\mu}}_{\boldsymbol{\theta}} + \sigma \underline{\epsilon}} \frac{\partial(\underline{\boldsymbol{\mu}}_{\boldsymbol{\theta}} + \sigma \underline{\epsilon})}{\partial \sigma} \right) \tag{123}$$

$$= \frac{\partial}{\partial \sigma} \left( \nabla_{\boldsymbol{\eta}} J_{\mathrm{D}}(\boldsymbol{\eta})|_{\boldsymbol{\eta} = \underline{\boldsymbol{\mu}}_{\boldsymbol{\theta}} + \sigma \underline{\epsilon}} \underline{\epsilon} \right) \tag{124}$$

$$= \underline{\epsilon}^\top \nabla_{\boldsymbol{\eta}}^2 J_{\mathrm{D}}(\boldsymbol{\eta})|_{\boldsymbol{\eta} = \underline{\boldsymbol{\mu}}_{\boldsymbol{\theta}} + \sigma \underline{\epsilon}} \underline{\epsilon}. \tag{125}$$

Thus, applying the absolute value to the quantity $\frac{\partial^2}{\partial \sigma^2} J_A(\boldsymbol{\theta}, \sigma)$, and recalling that Assumption 7.2 holds, we have what follows:

$$\left| \frac{\partial^2}{\partial \sigma^2} J_A(\boldsymbol{\theta}, \sigma) \right| \leqslant \mathop{\mathbb{E}}_{\underline{\epsilon} \sim \Phi_{d_{\mathcal{A}},1}^T} \left[ \left| \frac{\partial^2}{\partial \sigma^2} J_{\mathrm{D}}(\underline{\boldsymbol{\mu}}_{\boldsymbol{\theta}} + \sigma \underline{\epsilon}) \right| \right] \tag{126}$$

$$= \mathop{\mathbb{E}}_{\underline{\epsilon} \sim \Phi_{d_{\mathcal{A}},1}^T} \left[ \left| \underline{\epsilon}^\top \nabla_{\boldsymbol{\eta}}^2 J_{\mathrm{D}}(\boldsymbol{\eta})|_{\boldsymbol{\eta} = \underline{\boldsymbol{\mu}}_{\boldsymbol{\theta}} + \sigma \underline{\epsilon}} \underline{\epsilon} \right| \right] \tag{127}$$

$$\leqslant L_{2,\underline{\boldsymbol{\mu}}} \mathop{\mathbb{E}}_{\underline{\epsilon} \sim \Phi_{d_{\mathcal{A}},1}^T} \left[ \|\underline{\epsilon}\|_2^2 \right] \tag{128}$$

$$\leqslant L_{2,\underline{\boldsymbol{\mu}}} T d_{\mathcal{A}}. \tag{129}$$

$\square$

**Lemma G.6.** ($J_A$ *Inherited Smoothness w.r.t.* $\xi$) *If the policy satisfies Definition 2.2, under Assumption 7.2, 7.1, 5.2, and 5.3, for every* $\boldsymbol{\theta} \in \mathbb{R}^{d_\Theta}$ *and* $\sigma \in \mathbb{R}^+$, *the second derivative of* $J_A$ *w.r.t.* $\xi$ *is bounded as follows:*

$$\left| \frac{\partial^2}{\partial \xi^2} J_A(\boldsymbol{\theta}, f(\xi)) \right| \leqslant L_{2A,\xi},$$

*where* $L_{2A,\xi} = L_{2,\underline{\boldsymbol{\mu}}} L_{1,f}^2 T d_{\mathcal{A}} + L_{1,\underline{\boldsymbol{\mu}}} L_{2,f} \sqrt{T d_{\mathcal{A}}}$.

*Proof.* The proof of this lemma follows the same reasoning of the one of Lemma G.2. The only difference is in the fact that the noise vector $\underline{\epsilon}$ has dimension $T d_{\mathcal{A}}$, as shown also in the proof of Lemma G.5.

Indeed, given that $\sigma = f(\xi)$ and that $f$ satisfies Assumption 7.1, and given the fact that the policy $\pi_{\boldsymbol{\theta},\sigma}$ satisfies Definition 2.2 and that Assumption 5.3 holds, we can write the following:

$$\frac{\partial^2}{\partial \xi^2} J_{\mathrm{A}}(\boldsymbol{\theta}, f(\xi)) = \frac{\partial^2}{\partial \xi^2} \mathop{\mathbb{E}}_{\underline{\boldsymbol{\epsilon}} \sim \Phi^T_{d_{\mathcal{A}},1}} \left[ J_{\mathrm{D}}(\underline{\boldsymbol{\mu}}_{\boldsymbol{\theta}} + f(\xi)\underline{\boldsymbol{\epsilon}}) \right] \tag{130}$$

$$= \mathop{\mathbb{E}}_{\underline{\boldsymbol{\epsilon}} \sim \Phi^T_{d_{\mathcal{A}},1}} \left[ \frac{\partial^2}{\partial \xi^2} J_{\mathrm{D}}(\underline{\boldsymbol{\mu}}_{\boldsymbol{\theta}} + f(\xi)\underline{\boldsymbol{\epsilon}}) \right]. \tag{131}$$

We now need to apply the chain rule to the term $\frac{\partial^2}{\partial \xi^2} J_{\mathrm{D}}(\underline{\boldsymbol{\mu}}_{\boldsymbol{\theta}} + f(\xi)\underline{\boldsymbol{\epsilon}})$:

$$\frac{\partial^2}{\partial \xi^2} J_{\mathrm{D}}(\underline{\boldsymbol{\mu}}_{\boldsymbol{\theta}} + f(\xi)\underline{\boldsymbol{\epsilon}}) = \frac{\partial}{\partial \xi} \left( \frac{\partial}{\partial \xi} J_{\mathrm{D}}(\underline{\boldsymbol{\mu}}_{\boldsymbol{\theta}} + f(\xi)\underline{\boldsymbol{\epsilon}}) \right) \tag{132}$$

$$= \frac{\partial}{\partial \xi} \left( \nabla_{\boldsymbol{\eta}} J_{\mathrm{D}}(\boldsymbol{\eta})|_{\boldsymbol{\eta}=\underline{\boldsymbol{\mu}}_{\boldsymbol{\theta}}+f(\xi)\underline{\boldsymbol{\epsilon}}} \frac{\partial(\underline{\boldsymbol{\mu}}_{\boldsymbol{\theta}} + f(\xi)\underline{\boldsymbol{\epsilon}})}{\partial \xi} \right) \tag{133}$$

$$= \frac{\partial}{\partial \xi} \left( \nabla_{\boldsymbol{\eta}} J_{\mathrm{D}}(\boldsymbol{\eta})|_{\boldsymbol{\eta}=\underline{\boldsymbol{\mu}}_{\boldsymbol{\theta}}+f(\xi)\underline{\boldsymbol{\epsilon}}} \frac{\partial}{\partial \xi} f(\xi)\underline{\boldsymbol{\epsilon}} \right) \tag{134}$$

$$= \underline{\boldsymbol{\epsilon}}^{\top} \nabla^2_{\boldsymbol{\eta}} J_{\mathrm{D}}(\boldsymbol{\eta})|_{\boldsymbol{\eta}=\underline{\boldsymbol{\mu}}_{\boldsymbol{\theta}}+f(\xi)\underline{\boldsymbol{\epsilon}}} \underline{\boldsymbol{\epsilon}} \frac{\partial}{\partial \xi} f(\xi)^2 + \nabla_{\boldsymbol{\eta}} J_{\mathrm{D}}(\boldsymbol{\eta})|_{\boldsymbol{\eta}=\underline{\boldsymbol{\mu}}_{\boldsymbol{\theta}}+f(\xi)\underline{\boldsymbol{\epsilon}}} \underline{\boldsymbol{\epsilon}} \frac{\partial^2}{\partial \xi^2} f(\xi). \tag{135}$$

Thus, applying the absolute value to the quantity $\frac{\partial^2}{\partial \xi^2} J_{\mathrm{A}}(\boldsymbol{\theta}, f(\xi))$, and recalling that Assumptions 7.2 and 5.2 hold, we have what follows:

$$\left| \frac{\partial^2}{\partial \xi^2} J_{\mathrm{A}}(\boldsymbol{\theta}, f(\xi)) \right| \tag{136}$$

$$\leqslant \mathop{\mathbb{E}}_{\underline{\boldsymbol{\epsilon}} \sim \Phi^T_{d_{\mathcal{A}},1}} \left[ \left| \frac{\partial^2}{\partial \xi^2} J_{\mathrm{D}}(\underline{\boldsymbol{\mu}}_{\boldsymbol{\theta}} + f(\xi)\underline{\boldsymbol{\epsilon}}) \right| \right] \tag{137}$$

$$= \mathop{\mathbb{E}}_{\underline{\boldsymbol{\epsilon}} \sim \Phi^T_{d_{\mathcal{A}},1}} \left[ \left| \underline{\boldsymbol{\epsilon}}^{\top} \nabla^2_{\boldsymbol{\eta}} J_{\mathrm{D}}(\boldsymbol{\eta})|_{\boldsymbol{\eta}=\underline{\boldsymbol{\mu}}_{\boldsymbol{\theta}}+f(\xi)\underline{\boldsymbol{\epsilon}}} \underline{\boldsymbol{\epsilon}} \frac{\partial}{\partial \xi} f(\xi)^2 \right| \right] + \mathop{\mathbb{E}}_{\underline{\boldsymbol{\epsilon}} \sim \Phi^T_{d_{\mathcal{A}},1}} \left[ \left| \nabla_{\boldsymbol{\eta}} J_{\mathrm{D}}(\boldsymbol{\eta})|_{\boldsymbol{\eta}=\underline{\boldsymbol{\mu}}_{\boldsymbol{\theta}}+f(\xi)\underline{\boldsymbol{\epsilon}}} \underline{\boldsymbol{\epsilon}} \frac{\partial^2}{\partial \xi^2} f(\xi) \right| \right] \tag{138}$$

$$\leqslant L_{2,\boldsymbol{\mu}} L^2_{1,f} T d_{\mathcal{A}} + L_{1,\boldsymbol{\mu}} L_{2,f} \sqrt{T d_{\mathcal{A}}}. \tag{139}$$

$\square$

**Lemma G.7.** *($J_A$ Inherited LC w.r.t. $\xi$) If the policy satisfies Definition 2.2, under Assumption 5.2, 7.1, and 5.3, for every $\boldsymbol{\theta} \in \mathbb{R}^{d_{\ominus}}$ and $\sigma \in \mathbb{R}^{+}$, the derivative of $J_A$ w.r.t. $\xi$ is bounded as follows:*

$$\left| \frac{\partial}{\partial \xi} J_A(\boldsymbol{\theta}, f(\xi)) \right| \leqslant L_{1A,\xi},$$

*where $L_{1A,\xi} = L_{1,\boldsymbol{\mu}} L_{1,f} \sqrt{T d_{\mathcal{A}}}$.*

*Proof.* The proof of this lemma follows the same reasoning of the one of Lemma G.3. The only difference is in the fact that the noise vector $\underline{\boldsymbol{\epsilon}}$ has dimension $T d_{\mathcal{A}}$, as shown also in the proof of Lemma G.5.

Indeed, given that $\sigma = f(\xi)$ and that $f$ satisfies Assumption 7.1, and given the fact that the policy $\pi_{\boldsymbol{\theta},\sigma}$ satisfies Definition 2.2 and that Assumption 5.3 holds, we can write the following:

$$\frac{\partial}{\partial \xi} J_{\mathrm{A}}(\boldsymbol{\theta}, f(\xi)) = \frac{\partial}{\partial \xi} \mathop{\mathbb{E}}_{\underline{\boldsymbol{\epsilon}} \sim \Phi^T_{d_{\mathcal{A}},1}} \left[ J_{\mathrm{D}}(\underline{\boldsymbol{\mu}}_{\boldsymbol{\theta}} + f(\xi)\underline{\boldsymbol{\epsilon}}) \right] \tag{140}$$

$$= \mathop{\mathbb{E}}_{\underline{\boldsymbol{\epsilon}} \sim \Phi^T_{d_{\mathcal{A}},1}} \left[ \frac{\partial}{\partial \xi} J_{\mathrm{D}}(\underline{\boldsymbol{\mu}}_{\boldsymbol{\theta}} + f(\xi)\underline{\boldsymbol{\epsilon}}) \right]. \tag{141}$$

We now need to apply the chain rule to the term $\frac{\partial}{\partial \xi} J_{\mathrm{D}}(\underline{\boldsymbol{\mu}}_{\boldsymbol{\theta}} + f(\xi)\underline{\boldsymbol{\epsilon}})$:

$$\frac{\partial}{\partial \xi} J_{\mathrm{D}}(\underline{\boldsymbol{\mu}}_{\boldsymbol{\theta}} + f(\xi)\underline{\boldsymbol{\epsilon}}) = \frac{\partial}{\partial \xi} J_{\mathrm{D}}(\underline{\boldsymbol{\mu}}_{\boldsymbol{\theta}} + f(\xi)\underline{\boldsymbol{\epsilon}}) \tag{142}$$

$$= \nabla_{\boldsymbol{\eta}} \left. J_{\mathrm{D}}(\boldsymbol{\eta}) \right|_{\boldsymbol{\eta} = \underline{\boldsymbol{\mu}}_{\boldsymbol{\theta}} + f(\xi)\underline{\boldsymbol{\epsilon}}} \frac{\partial (\underline{\boldsymbol{\mu}}_{\boldsymbol{\theta}} + f(\xi)\underline{\boldsymbol{\epsilon}})}{\partial \xi} \tag{143}$$

$$= \nabla_{\boldsymbol{\eta}} \left. J_{\mathrm{D}}(\boldsymbol{\eta}) \right|_{\boldsymbol{\eta} = \underline{\boldsymbol{\mu}}_{\boldsymbol{\theta}} + f(\xi)\underline{\boldsymbol{\epsilon}}} \frac{\partial}{\partial \xi} f(\xi)\underline{\boldsymbol{\epsilon}}. \tag{144}$$

Thus, applying the absolute value to the quantity $\frac{\partial}{\partial \xi} J_{\mathrm{A}}(\boldsymbol{\theta}, f(\xi))$, and recalling that Assumption 5.2 holds, we have what follows:

$$\left| \frac{\partial}{\partial \xi} J_{\mathrm{A}}(\boldsymbol{\theta}, f(\xi)) \right| \tag{145}$$

$$\leqslant \underset{\underline{\boldsymbol{\epsilon}} \sim \Phi_{d_{\mathcal{A}},1}^T}{\mathbb{E}} \left[ \left| \frac{\partial}{\partial \xi} J_{\mathrm{D}}(\underline{\boldsymbol{\mu}}_{\boldsymbol{\theta}} + f(\xi)\underline{\boldsymbol{\epsilon}}) \right| \right] \tag{146}$$

$$= \underset{\underline{\boldsymbol{\epsilon}} \sim \Phi_{d_{\mathcal{A}},1}^T}{\mathbb{E}} \left[ \left| \nabla_{\boldsymbol{\eta}} \left. J_{\mathrm{D}}(\boldsymbol{\eta}) \right|_{\boldsymbol{\eta} = \underline{\boldsymbol{\mu}}_{\boldsymbol{\theta}} + f(\xi)\underline{\boldsymbol{\epsilon}}} \frac{\partial}{\partial \xi} f(\xi)\underline{\boldsymbol{\epsilon}} \right| \right] \tag{147}$$

$$\leqslant L_{1,\underline{\boldsymbol{\mu}}} L_{1,f} \sqrt{T d_{\mathcal{A}}}. \tag{148}$$

$\square$

**Lemma G.8** ($J_{\mathrm{A}}$ Inherited Lipschitz Gradient w.r.t. $\xi$)**.** *If the policy satisfies Definition 2.2, under Assumptions 7.2 and 7.1, for every $\boldsymbol{\theta}_1, \boldsymbol{\theta}_2 \in \Theta$ and $\xi \in \mathbb{R}$, it holds:*

$$\left| \frac{\partial}{\partial \xi} J_A(\boldsymbol{\theta}_1, f(\xi)) - \frac{\partial}{\partial \xi} J_A(\boldsymbol{\theta}_2, f(\xi)) \right| \leqslant L_{1,f} L_{2,\underline{\boldsymbol{\mu}}} \sqrt{T d_{\mathcal{A}}} \|\boldsymbol{\theta}_1 - \boldsymbol{\theta}_2\|_2. \tag{149}$$

*Proof.* The proof of this lemma follows the same reasoning of the one of Lemma G.8. The only difference is in the fact that the noise vector $\underline{\boldsymbol{\epsilon}}$ has dimension $T d_{\mathcal{A}}$, as shown also in the proof of Lemma G.5.

Under Assumption 5.3, by definition of $J_{\mathrm{A}}$ using a hyperpolicy satisfying Definition 2.2, it holds:

$$J_{\mathrm{A}}(\boldsymbol{\theta}, f(\xi)) = \underset{\underline{\boldsymbol{\epsilon}} \sim \Phi_{d_{\mathcal{A}},1}^T}{\mathbb{E}} \left[ J_{\mathrm{D}}(\underline{\boldsymbol{\mu}}_{\boldsymbol{\theta}} + f(\xi)\underline{\boldsymbol{\epsilon}}) \right]. \tag{150}$$

Moreover, by applying the chain rule, we have:

$$\frac{\partial}{\partial \xi} J_{\mathrm{A}}(\boldsymbol{\theta}, f(\xi)) = \nabla_{\boldsymbol{\eta}} \left. J_{\mathrm{D}}(\boldsymbol{\eta}) \right|_{\boldsymbol{\eta} = \underline{\boldsymbol{\mu}}_{\boldsymbol{\theta}} + f(\xi)\underline{\boldsymbol{\epsilon}}} \frac{\partial}{\partial \xi} f(\xi)\underline{\boldsymbol{\epsilon}}. \tag{151}$$

Thus, the following derivation holds:

$$\left| \frac{\partial}{\partial \xi} J_{\mathrm{A}}(\boldsymbol{\theta}_1, f(\xi)) - \frac{\partial}{\partial \xi} J_{\mathrm{A}}(\boldsymbol{\theta}_2, f(\xi)) \right| \tag{152}$$

$$= \left| \frac{\partial}{\partial \xi} \underset{\underline{\boldsymbol{\epsilon}} \sim \Phi_{d_{\mathcal{A}},1}^T}{\mathbb{E}} \left[ J_{\mathrm{D}}(\underline{\boldsymbol{\mu}}_{\boldsymbol{\theta}_1} + f(\xi)\underline{\boldsymbol{\epsilon}}) \right] - \frac{\partial}{\partial \xi} \underset{\underline{\boldsymbol{\epsilon}} \sim \Phi_{d_{\mathcal{A}},1}^T}{\mathbb{E}} \left[ J_{\mathrm{D}}(\underline{\boldsymbol{\mu}}_{\boldsymbol{\theta}_2} + f(\xi)\underline{\boldsymbol{\epsilon}}) \right] \right| \tag{153}$$

$$\leqslant \underset{\underline{\boldsymbol{\epsilon}} \sim \Phi_{d_{\mathcal{A}},1}^T}{\mathbb{E}} \left[ \left| \frac{\partial}{\partial \xi} J_{\mathrm{D}}(\underline{\boldsymbol{\mu}}_{\boldsymbol{\theta}_1} + f(\xi)\underline{\boldsymbol{\epsilon}}) - \frac{\partial}{\partial \xi} J_{\mathrm{D}}(\underline{\boldsymbol{\mu}}_{\boldsymbol{\theta}_2} + f(\xi)\underline{\boldsymbol{\epsilon}}) \right| \right] \tag{154}$$

$$= \underset{\underline{\boldsymbol{\epsilon}} \sim \Phi_{d_{\mathcal{A}},1}^T}{\mathbb{E}} \left[ \left| \nabla_{\boldsymbol{\eta}} \left. J_{\mathrm{D}}(\boldsymbol{\eta}_1) \right|_{\boldsymbol{\eta}_1 = \underline{\boldsymbol{\mu}}_{\boldsymbol{\theta}_1} + f(\xi)\underline{\boldsymbol{\epsilon}}} \frac{\partial}{\partial \xi} f(\xi)\underline{\boldsymbol{\epsilon}} - \nabla_{\boldsymbol{\eta}} \left. J_{\mathrm{D}}(\boldsymbol{\eta}_2) \right|_{\boldsymbol{\eta}_2 = \underline{\boldsymbol{\mu}}_{\boldsymbol{\theta}_2} + f(\xi)\underline{\boldsymbol{\epsilon}}} \frac{\partial}{\partial \xi} f(\xi)\underline{\boldsymbol{\epsilon}} \right| \right] \tag{155}$$

$$\leqslant \underset{\underline{\boldsymbol{\epsilon}} \sim \Phi_{d_{\mathcal{A}},1}^T}{\mathbb{E}} \left[ \left| \frac{\partial}{\partial \xi} f(\xi) \right| \|\underline{\boldsymbol{\epsilon}}\|_2 \left\| \nabla_{\boldsymbol{\eta}} \left. J_{\mathrm{D}}(\boldsymbol{\eta}_1) \right|_{\boldsymbol{\eta}_1 = \underline{\boldsymbol{\mu}}_{\boldsymbol{\theta}_1} + f(\xi)\underline{\boldsymbol{\epsilon}}} - \nabla_{\boldsymbol{\eta}} \left. J_{\mathrm{D}}(\boldsymbol{\eta}_2) \right|_{\boldsymbol{\eta}_2 = \underline{\boldsymbol{\mu}}_{\boldsymbol{\theta}_2} + f(\xi)\underline{\boldsymbol{\epsilon}}} \right\|_2 \right] \tag{156}$$

$$\leqslant L_{1,f} L_{2,\underline{\boldsymbol{\mu}}} \sqrt{T d_{\mathcal{A}}} \|\boldsymbol{\theta}_1 - \boldsymbol{\theta}_2\|_2, \tag{157}$$

where, in the first inequality, we employed the Jensen's inequality, and, for the last passage, we exploited Assumption 7.2. $\square$

**Lemma 7.1** ($J_{\dagger}$ is $L_{1\dagger,\xi}$-LC and $L_{2\dagger,\xi}$-LS w.r.t. $\xi$)**.** *Under Assumptions 4.1, 5.1, 5.2, 7.1, and 7.2 it holds that:*

$$\left| \frac{\partial}{\partial \xi} J_{\dagger}(\boldsymbol{\theta}, f(\xi)) \right| \leqslant L_{1\dagger,\xi} \quad \text{and} \quad \left| \frac{\partial^2}{\partial \xi^2} J_{\dagger}(\boldsymbol{\theta}, f(\xi)) \right| \leqslant L_{2\dagger,\xi}.$$

*Proof.* This lemma is the ensemble of Lemmas G.6, G.7, G.2, and G.3. $\qquad\square$

**Lemma G.9** ($J_\dagger$ Inherited Lipschitz Gradient w.r.t. $\xi$)*. If the (hyper)policy satisfies Definition 2.2 (AB) or 2.3 (PB), under Assumptions 4.1 and 7.1, for every $\boldsymbol{\theta}_1, \boldsymbol{\theta}_2 \in \Theta$ and $\xi \in \mathbb{R}$, it holds that:*

$$\left| \frac{\partial}{\partial \xi} J_\dagger(\boldsymbol{\theta}_1, f(\xi)) - \frac{\partial}{\partial \xi} J_\dagger(\boldsymbol{\theta}_2, f(\xi)) \right| \leqslant L_{3\dagger,\xi} \|\boldsymbol{\theta}_1 - \boldsymbol{\theta}_2\|_2.$$

*Proof.* This lemma is the ensemble of Lemmas G.4 and G.8. $\qquad\square$

## G.3. Bounded Variance of the Estimators $\widehat{\nabla}_\xi J_\dagger(\boldsymbol{\theta}, f(\xi))$

**Lemma G.10** (Bounded Hyperpolicy Scores)*. If the hyperpolicy $\nu_{\boldsymbol{\theta},\sigma}$ satisfies Definition 2.3, under Assumption 5.3 and 4.4 and using an exploration mapping function $f(\cdot)$ such that Assumption 7.1 is fulfilled, then the following holds:*

$$\mathop{\mathbb{E}}_{\boldsymbol{\theta}' \sim \nu_{\boldsymbol{\theta},\sigma}} \left[ \left| \frac{\partial}{\partial \xi} \log \nu_{\boldsymbol{\theta},\sigma}(\boldsymbol{\theta}') \right|^2 \right] \leqslant c d_\Theta^2 L_{1,f}^2 \sigma^{-2},$$

*where $\sigma = f(\xi)$.*

*Proof.* From Definition 2.3 and from Assumption 5.3, we know that if $\boldsymbol{\theta}' \sim \nu_{\boldsymbol{\theta},\sigma}$, then

$$\boldsymbol{\theta}' = \boldsymbol{\theta} + \sigma\bar{\boldsymbol{\epsilon}} = \boldsymbol{\theta} + f(\xi)\bar{\boldsymbol{\epsilon}}, \tag{158}$$

where $\bar{\boldsymbol{\epsilon}} \sim \Phi_{d_\Theta, 1}$.

Moreover, we know that

$$\nu_{\boldsymbol{\theta},\sigma}(\boldsymbol{\theta}') = \phi_{d_\Theta,\sigma}(\boldsymbol{\theta}' - \boldsymbol{\theta}) = \phi_{d_\Theta,\sigma}(f(\xi)\bar{\boldsymbol{\epsilon}}). \tag{159}$$

Thus, we can write what follows:

$$\frac{\partial}{\partial \xi} \log \nu_{\boldsymbol{\theta},\sigma}(\boldsymbol{\theta}') = \frac{\partial}{\partial \xi} \log \phi_{d_\Theta,\sigma}(f(\xi)\bar{\boldsymbol{\epsilon}}) = \nabla_{\boldsymbol{\epsilon}} \log \phi_{d_\Theta,\sigma}(\boldsymbol{\epsilon})|_{\boldsymbol{\epsilon}=f(\xi)\bar{\boldsymbol{\epsilon}}} \frac{\partial}{\partial \xi} f(\xi)\bar{\boldsymbol{\epsilon}}. \tag{160}$$

By applying the expectation w.r.t. the sampled parameter $\boldsymbol{\theta}'$ to the square of $\frac{\partial}{\partial \xi} \log \nu_{\boldsymbol{\theta},\sigma}(\boldsymbol{\theta}')$, we obtain:

$$\mathop{\mathbb{E}}_{\boldsymbol{\theta}' \sim \nu_{\boldsymbol{\theta},\sigma}} \left[ \left| \frac{\partial}{\partial \xi} \log \nu_{\boldsymbol{\theta},\sigma}(\boldsymbol{\theta}') \right|^2 \right] = \mathop{\mathbb{E}}_{\boldsymbol{\theta}' \sim \nu_{\boldsymbol{\theta},\sigma}} \left[ \left| \nabla_{\boldsymbol{\epsilon}} \log \phi_{d_\Theta,\sigma}(\boldsymbol{\epsilon})|_{\boldsymbol{\epsilon}=f(\xi)\bar{\boldsymbol{\epsilon}}} \frac{\partial}{\partial \xi} f(\xi)\bar{\boldsymbol{\epsilon}} \right|^2 \right] \tag{161}$$

$$\leqslant L_{1,f}^2 c d_\Theta^2 f(\xi)^{-2}, \tag{162}$$

where we exploited Assumptions 4.4 and 7.1. $\qquad\square$

**Lemma G.11** ($\widehat{\nabla}_\xi J_P$ Bounded Variance)*. If the hyperpolicy $\nu_{\boldsymbol{\theta},\sigma}$ satisfies Definition 2.3, under Assumption 5.3 and 4.4 and using an exploration mapping function $f(\cdot)$ such that Assumption 7.1 is fulfilled and a parameterization $\xi$ for the exploration amount $\sigma$, then the following holds:*

$$\mathop{\mathbb{V}\mathrm{ar}}_{\boldsymbol{\theta}' \sim \nu_{\boldsymbol{\theta},\sigma}} \left[ \widehat{\nabla}_\xi J_P(\boldsymbol{\theta}', \sigma) \right] \leqslant \frac{R_{\max}^2 c d_\Theta^2 L_{1,f}^2}{N(1-\gamma)^2 \sigma^2},$$

*where $\sigma = f(\xi)$.*

*Proof.* We recall that the estimator at hand is:

$$\widehat{\nabla}_\xi J_P(\boldsymbol{\theta}, \sigma) = \frac{1}{N} \sum_{i=1}^{N} \frac{\partial}{\partial \xi} \log \nu_{\boldsymbol{\theta},\sigma}(\boldsymbol{\theta}_i) R(\tau_i),$$

where $N$ is the number of parameter configuration tested (on one trajectory) at each iteration and $\sigma = f(\xi)$. Thus, we can compute the variance of such an estimator as:

$$\mathop{\mathbb{V}\mathrm{ar}}_{\boldsymbol{\theta}' \sim \nu_{\boldsymbol{\theta},\sigma}} \left[ \widehat{\nabla}_\xi J_P(\boldsymbol{\theta}', \sigma) \right] = \frac{1}{N} \mathop{\mathbb{V}\mathrm{ar}}_{\boldsymbol{\theta}' \sim \nu_{\boldsymbol{\theta},\sigma}} \left[ \frac{\partial}{\partial \xi} \log \nu_{\boldsymbol{\theta},\sigma}(\boldsymbol{\theta}') R(\tau_1) \right] \tag{163}$$

$$= \frac{1}{N} \mathop{\mathbb{E}}_{\boldsymbol{\theta}' \sim \nu_{\boldsymbol{\theta},\sigma}} \left[ \left| \left| \frac{\partial}{\partial \xi} \log \nu_{\boldsymbol{\theta},\sigma}(\boldsymbol{\theta}') \right| \right|^2 R(\tau_1)^2 \right]. \tag{164}$$

We can conclude the proof by applying Lemma G.10 and by considering the fact that, given a trajectory $\tau$, $R(\tau)$ is defined as:

$$R(\tau) = \sum_{t=0}^{T-1} \gamma^t r(\mathbf{s}_{\tau,t}, \mathbf{a}_{\tau,t}),$$

with $r(\mathbf{s}, \mathbf{a}) \in [-R_{\max}, R_{\max}]$ for every $\mathbf{s} \in \mathcal{S}$ and $\mathbf{a} \in \mathcal{A}$. $\qquad \square$

**Lemma G.12** (Bounded Policy Scores). *If the policy $\pi_{\boldsymbol{\theta},\sigma}$ satisfies Definition 2.2, under Assumption 5.3 and 4.4 and using an exploration mapping function $f(\cdot)$ such that Assumption 7.1 is fulfilled, then the following holds for every state $\mathbf{s} \in \mathcal{S}$:*

$$\mathop{\mathbb{E}}_{\mathbf{a} \sim \pi_{\boldsymbol{\theta},\sigma}(\cdot|\mathbf{s})} \left[ \left| \left| \frac{\partial}{\partial \xi} \log \pi_{\boldsymbol{\theta},\sigma}(\mathbf{a}|\mathbf{s}) \right| \right|^2 \right] \leqslant c d_{\mathcal{A}}^2 L_{1,f}^2 \sigma^{-2},$$

*where $\sigma = f(\xi)$.*

*Proof.* From Definition 2.2 and from Assumption 5.3, we know that if $\mathbf{a} \sim \pi_{\boldsymbol{\theta},\sigma}(\cdot|\mathbf{s})$, then

$$\mathbf{a} = \mu_{\boldsymbol{\theta}}(\mathbf{s}) + \sigma \bar{\boldsymbol{\epsilon}} = \mu_{\boldsymbol{\theta}}(\mathbf{s}) + f(\xi)\bar{\boldsymbol{\epsilon}}, \tag{165}$$

where $\bar{\boldsymbol{\epsilon}} \sim \Phi_{d_{\mathcal{A}},1}$.

Moreover, we know that

$$\pi_{\boldsymbol{\theta},\sigma}(\mathbf{a}|\mathbf{s}) = \phi_{d_{\mathcal{A}},\sigma}(\mathbf{a} - \mu_{\boldsymbol{\theta}}(\mathbf{s})) = \phi_{d_{\mathcal{A}},\sigma}(f(\xi)\bar{\boldsymbol{\epsilon}}). \tag{166}$$

Thus, we can write what follows:

$$\frac{\partial}{\partial \xi} \log \pi_{\boldsymbol{\theta},\sigma}(\mathbf{a}|\mathbf{s}) = \frac{\partial}{\partial \xi} \log \phi_{d_{\mathcal{A}},\sigma}(f(\xi)\bar{\boldsymbol{\epsilon}}) = \nabla_{\boldsymbol{\epsilon}} \log \phi_{d_{\mathcal{A}},\sigma}(\boldsymbol{\epsilon})|_{\boldsymbol{\epsilon}=f(\xi)\bar{\boldsymbol{\epsilon}}} \frac{\partial}{\partial \xi} f(\xi)\bar{\boldsymbol{\epsilon}}. \tag{167}$$

By applying the expectation w.r.t. the sampled action $\mathbf{a}$ to the 2-norm squared of $\nabla_{\xi} \log \pi_{\boldsymbol{\theta},\sigma}(\mathbf{a}|\mathbf{s})$, we obtain:

$$\mathop{\mathbb{E}}_{\mathbf{a} \sim \pi_{\boldsymbol{\theta},\sigma}(\cdot|\mathbf{s})} \left[ \left| \left| \frac{\partial}{\partial \xi} \log \pi_{\boldsymbol{\theta},\sigma}(\mathbf{a}|\mathbf{s}) \right| \right|^2 \right] = \mathop{\mathbb{E}}_{\boldsymbol{\theta}' \sim \pi_{\boldsymbol{\theta},\sigma}} \left[ \left| \left| \nabla_{\boldsymbol{\epsilon}} \log \phi_{d_{\mathcal{A}},\sigma}(\boldsymbol{\epsilon})|_{\boldsymbol{\epsilon}=f(\xi)\bar{\boldsymbol{\epsilon}}} \frac{\partial}{\partial \xi} f(\xi)\bar{\boldsymbol{\epsilon}} \right| \right|^2 \right] \tag{168}$$

$$\leqslant L_{1,f}^2 c d_{\mathcal{A}}^2 f(\xi)^{-2}, \tag{169}$$

where we exploited Assumptions 4.4 and 7.1. $\qquad \square$

**Lemma G.13** ($\widehat{\nabla}_{\xi} J_{\mathrm{A}}$ Bounded Variance). *If the policy $\pi_{\boldsymbol{\theta},\sigma}$ satisfies Definition 2.2, under Assumption 5.3 and 4.4 and using an exploration mapping function $f(\cdot)$ such that Assumption 7.1 is fulfilled and a parameterization $\xi$ for the exploration amount $\sigma$, then the following holds:*

$$\mathop{\mathbb{V}\mathrm{ar}}_{\tau \sim p_A(\cdot|(\boldsymbol{\theta},\sigma))} \left[ \widehat{\nabla}_{\xi} J_A(\boldsymbol{\theta}, \sigma) \right] \leqslant \frac{R_{\max}^2 c d_{\mathcal{A}}^2 L_{1,f}^2}{N(1-\gamma)^3 \sigma^2},$$

*where $\sigma = f(\xi)$.*

*Proof.* We recall that the estimator at hand is:

$$\widehat{\nabla}_{\xi} J_A(\boldsymbol{\theta}, \sigma) := \frac{1}{N} \sum_{i=1}^{N} \sum_{t=0}^{T-1} \left( \sum_{k=0}^{t} \frac{\partial}{\partial \xi} \log \pi_{\boldsymbol{\theta},\sigma}(\mathbf{a}_{\tau_i,k}|\mathbf{s}_{\tau_i,k}) \right) \gamma^t r(\mathbf{s}_{\tau_i,t}, \mathbf{a}_{\tau_i,t}),$$

where $N$ is the number of trajectories tested at each iteration and $\sigma = f(\xi)$. Thus, we can compute the variance of such an estimator as:

$$\mathop{\mathbb{V}\mathrm{ar}}_{\tau \sim p_A(\cdot|(\boldsymbol{\theta},\sigma))} \left[ \widehat{\nabla}_{\xi} J_A(\boldsymbol{\theta}, \sigma) \right] \tag{170}$$

$$= \frac{1}{N} \mathop{\mathbb{V}\mathrm{ar}}_{\tau \sim p_A(\cdot|(\boldsymbol{\theta},\sigma))} \left[ \sum_{t=0}^{T-1} \left( \sum_{k=0}^{t} \frac{\partial}{\partial \xi} \log \pi_{\boldsymbol{\theta},\sigma}(\mathbf{a}_{\tau_1,k}|\mathbf{s}_{\tau_1,k}) \right) \gamma^t r(\mathbf{s}_{\tau_1,t}, \mathbf{a}_{\tau_1,t}) \right] \tag{171}$$

$$\leqslant \frac{1}{N} \mathop{\mathbb{E}}_{\tau \sim p_{\mathrm{A}}(\cdot|(\boldsymbol{\theta},\sigma))} \left[ \left( \sum_{t=0}^{T-1} \left( \sum_{k=0}^{t} \frac{\partial}{\partial \xi} \log \pi_{\boldsymbol{\theta},\sigma}(\mathbf{a}_{\tau_1,k}|\mathbf{s}_{\tau_1,k}) \right) \gamma^t r(\mathbf{s}_{\tau_1,t}, \mathbf{a}_{\tau_1,t}) \right)^2 \right] \tag{172}$$

$$\leqslant \frac{1}{N} \mathop{\mathbb{E}}_{\tau \sim p_{\mathrm{A}}(\cdot|(\boldsymbol{\theta},\sigma))} \left[ \left( \sum_{t=0}^{T-1} \gamma^t r(\mathbf{s}_{\tau_1,t}, \mathbf{a}_{\tau_1,t})^2 \right) \left( \sum_{t=0}^{T-1} \gamma^t \left( \sum_{k=0}^{t} \frac{\partial}{\partial \xi} \log \pi_{\boldsymbol{\theta},\sigma}(\mathbf{a}_{\tau_1,k}|\mathbf{s}_{\tau_1,k}) \right)^2 \right) \right] \tag{173}$$

$$\leqslant \frac{R_{\max}^2}{N(1-\gamma)} \mathop{\mathbb{E}}_{\tau \sim p_{\mathrm{A}}(\cdot|(\boldsymbol{\theta},\sigma))} \left[ \sum_{t=0}^{T-1} \gamma^t \left( \sum_{k=0}^{t} \frac{\partial}{\partial \xi} \log \pi_{\boldsymbol{\theta},\sigma}(\mathbf{a}_{\tau_1,k}|\mathbf{s}_{\tau_1,k}) \right)^2 \right] \tag{174}$$

$$\leqslant \frac{R_{\max}^2 c d_{\mathcal{A}}^2 L_{1,f}^2}{N(1-\gamma)^3 \sigma^2}, \tag{175}$$

where the second inequality is Cauchy-Schwarz, and the last one is by Lemma G.12. $\qquad \square$

**Lemma 7.2** (Bounded $\widehat{\nabla}_\xi J_\dagger(\boldsymbol{\theta}, f(\xi))$)**.** *If the (hyper)policy satisfies Definitions 2.2 or 2.3, under Assumptions 4.4 and 5.3, using an exploration mapping $f(\cdot)$ fulfilling Assumption 7.1, for any parameterization $\xi$, it holds that:*

$$\mathbb{Var}[\widehat{\nabla}_\xi J_\dagger(\boldsymbol{\theta}, f(\xi))] \leqslant \frac{\mathcal{V}_{\dagger,\xi}}{N f(\xi)^2}.$$

*Proof.* This lemma is the ensemble of Lemmas G.11 and G.13. $\qquad \square$

### G.4. Last-Iterate Convergence Guarantees of `SL-PG` to Optimal Stochastic Policies

#### G.4.1. ADDITIONAL NOTATION

Before delving into the details of `SL-PG` last-iterate convergence guarantees, we introduce additional notation that will be used in the subsequent proofs.

Being in the context of Section 7, we consider the total parameterization $\boldsymbol{v} \in \mathbb{R}^{d_\Theta+1}$:

$$\boldsymbol{v} := (\boldsymbol{\theta}, \xi)^\top, \tag{176}$$

where $\boldsymbol{\theta}$ is the vector of dimension $d_\Theta$ representing the parameterization of the (hyper)policy.

`SL-PG` aims to maximize the objective:

$$\widetilde{J}_\dagger(\boldsymbol{v}) := J_\dagger(\boldsymbol{\theta}, f(\xi)), \tag{177}$$

where $f : \mathbb{R} \to [\sigma_{\min}, \sigma_{\max}]$ is a stochasticity mapping satisfying Assumption 7.1.

The `SL-PG` algorithm updates $\boldsymbol{v}$ as

$$\boldsymbol{v}_{k+1} \leftarrow \boldsymbol{v}_k + \delta \widehat{\nabla}_{\boldsymbol{v}} \widetilde{J}_\dagger(\boldsymbol{v}_k), \tag{178}$$

where $\delta > 0$ is the step size and $\widehat{\nabla}_{\boldsymbol{v}} \widetilde{J}_\dagger(\boldsymbol{v}_k)$ is an unbiased estimator of $\nabla_{\boldsymbol{v}} \widetilde{J}_\dagger(\boldsymbol{v}_k)$ (e.g., GPOMDP (**AB**) or PGPE (**PB**)).

In this context, we intend $\nabla_{\boldsymbol{v}} \widetilde{J}_\dagger(\boldsymbol{v})$ to be a $(d_\Theta + 1)$-dimensional vector:

$$\nabla_{\boldsymbol{v}} \widetilde{J}_\dagger(\boldsymbol{v}) := \left( \frac{\partial \widetilde{J}_\dagger(\boldsymbol{v})}{\partial \theta_1}, ..., \frac{\partial \widetilde{J}_\dagger(\boldsymbol{v})}{\partial \theta_{d_\Theta}}, \frac{\partial \widetilde{J}_\dagger(\boldsymbol{v})}{\partial \xi} \right)^\top. \tag{179}$$

Moreover, we intend $\nabla_{\boldsymbol{\theta}} \widetilde{J}_\dagger(\boldsymbol{v})$ and $\nabla_\xi \widetilde{J}_\dagger(\boldsymbol{v})$ as

$$\nabla_{\boldsymbol{\theta}} \widetilde{J}_\dagger(\boldsymbol{v}) := \left( \frac{\partial \widetilde{J}_\dagger(\boldsymbol{v})}{\partial \theta_1}, ..., \frac{\partial \widetilde{J}_\dagger(\boldsymbol{v})}{\partial \theta_{d_\Theta}}, 0 \right)^\top \quad \text{and} \quad \nabla_\xi \widetilde{J}_\dagger(\boldsymbol{v}) := \left( \mathbf{0}_{d_\Theta}, \frac{\partial \widetilde{J}_\dagger(\boldsymbol{v})}{\partial \xi} \right)^\top, \tag{180}$$

noticing that both of them have the same dimension of $\nabla_{\boldsymbol{v}} \widetilde{J}_\dagger(\boldsymbol{v})$

### G.4.2. CONVERGENCE

**Theorem 7.3** (SL-PG Convergence)**.** *If the (hyper)policy satisfies Definitions 2.2 or 2.3, under Assumptions 4.1, 4.3, 4.4, 5.1 (PB) or 5.2 (AB), 5.3, 7.1, 7.2, and 7.3, running SL-PG for $K$ iterations, with a suitable constant choice for the learning rate $\delta_k$, the output parameterization $\boldsymbol{v}_K$ is such that:*

$$\tilde{J}_\dagger^* - \mathbb{E}[\tilde{J}_\dagger(\boldsymbol{v}_K)] \leqslant \epsilon + \beta_{\boldsymbol{v}},$$

*with a total sample complexity of:*

$$NK = \widetilde{\mathcal{O}}\left( \frac{16\alpha_{\boldsymbol{v}}^4 L_{2\dagger,\boldsymbol{v}} \mathcal{V}_{\dagger,\boldsymbol{v}}}{\epsilon^3 \sigma_{\min}^2} \right),$$

*where $L_{2\dagger,\boldsymbol{v}} := L_2 + L_{2\dagger,\xi}$ and $\mathcal{V}_{\dagger,\boldsymbol{v}} := \mathcal{V}_{\dagger,\boldsymbol{\theta}} + \mathcal{V}_{\dagger,\xi}$.*

*Proof.* Here, the goal is to find the minimum amount of trajectories $NK$ needed, together with an appropriate choice of the learning rate $\delta$, to guarantee that the parameterization that SL-PG outputs in the last iterate $\boldsymbol{v}_K$ is such that the following quantity is bounded:

$$\tilde{J}_\dagger(\boldsymbol{v}^*) - \mathbb{E}\left[\tilde{J}_\dagger(\boldsymbol{v}_K)\right]. \tag{181}$$

The first step is to bound the difference between $\tilde{J}_\dagger(\boldsymbol{v}_{k+1})$ and $\tilde{J}_\dagger(\boldsymbol{v}_k)$.

$$\tilde{J}_\dagger(\boldsymbol{v}_{k+1}) - \tilde{J}_\dagger(\boldsymbol{v}_k) \tag{182}$$

$$\geqslant \left\langle \boldsymbol{v}_{k+1} - \boldsymbol{v}_k, \nabla_{\boldsymbol{v}}\tilde{J}_\dagger(\boldsymbol{v}_k) \right\rangle - \frac{L_{2\dagger,\boldsymbol{v}}}{2} \|\boldsymbol{v}_{k+1} - \boldsymbol{v}_k\|_2^2 \tag{183}$$

$$= \delta \left\langle \widehat{\nabla}_{\boldsymbol{v}}\tilde{J}_\dagger(\boldsymbol{v}_k), \nabla_{\boldsymbol{v}}\tilde{J}_\dagger(\boldsymbol{v}_k) \right\rangle - \frac{L_{2\dagger,\boldsymbol{v}}}{2}\delta^2 \left\|\widehat{\nabla}_{\boldsymbol{v}}\tilde{J}_\dagger(\boldsymbol{v}_k)\right\|_2^2 \tag{184}$$

$$= \delta \left\langle \widehat{\nabla}_{\boldsymbol{\theta}}\tilde{J}_\dagger(\boldsymbol{v}_k) + \widehat{\nabla}_\xi\tilde{J}_\dagger(\boldsymbol{v}_k), \nabla_{\boldsymbol{v}}\tilde{J}_\dagger(\boldsymbol{v}_k) \right\rangle - \frac{L_{2\dagger,\boldsymbol{v}}}{2}\delta^2 \left\|\widehat{\nabla}_{\boldsymbol{\theta}}\tilde{J}_\dagger(\boldsymbol{v}_k) + \widehat{\nabla}_\xi\tilde{J}_\dagger(\boldsymbol{v}_k)\right\|_2^2 \tag{185}$$

$$= \delta \left\langle \widehat{\nabla}_{\boldsymbol{\theta}}\tilde{J}_\dagger(\boldsymbol{v}_k), \nabla_{\boldsymbol{v}}\tilde{J}_\dagger(\boldsymbol{v}_k) \right\rangle + \delta \left\langle \widehat{\nabla}_\xi\tilde{J}_\dagger(\boldsymbol{v}_k), \nabla_{\boldsymbol{v}}\tilde{J}_\dagger(\boldsymbol{v}_k) \right\rangle - \frac{L_{2\dagger,\boldsymbol{v}}}{2}\delta^2 \left\|\widehat{\nabla}_{\boldsymbol{\theta}}\tilde{J}_\dagger(\boldsymbol{v}_k)\right\|_2^2 - \frac{L_{2\dagger,\boldsymbol{v}}}{2}\delta^2 \left\|\widehat{\nabla}_\xi\tilde{J}_\dagger(\boldsymbol{v}_k)\right\|_2^2, \tag{186}$$

where in the first inequality we used the quadratic bound and then we just used the particular form of the vectors $\nabla_{\boldsymbol{v}}\tilde{J}_\dagger(\cdot)$, $\nabla_{\boldsymbol{\theta}}\tilde{J}_\dagger(\cdot)$, and $\nabla_\xi\tilde{J}_\dagger(\cdot)$. Before carrying on with the proof, we recall that the holding of Lemmas 4.3 and 7.2 implies $\left\|\nabla_{\boldsymbol{v}}^2\tilde{J}_\dagger(\boldsymbol{v})\right\| \leqslant L_{2\dagger,\boldsymbol{v}} = L_2 + L_{2\dagger,\xi}$ as explained in Section 7.

Let $\mathcal{F}_k$ denote a $\sigma$-algebra representing all information available at the beginning of the $k^{\text{th}}$ iteration, and let $\mathbb{E}_k[\cdot]$ be short for $\mathbb{E}[\cdot|\mathcal{F}_k]$. Now consider the filtration $\mathcal{F} = (\mathcal{F}_k)_{k=0}^{K-1}$ and let $(\boldsymbol{v}_k)_{k=0}^{K-1}$ be an $\mathcal{F}$-adapted process.

We now apply the conditional expectation $\mathbb{E}_k[\cdot]$:

$$\mathbb{E}_k\left[\tilde{J}_\dagger(\boldsymbol{v}_{k+1}) - \tilde{J}_\dagger(\boldsymbol{v}_k)\right] \tag{187}$$

$$\mathbb{E}_k\left[\tilde{J}_\dagger(\boldsymbol{v}_{k+1})\right] - \tilde{J}_\dagger(\boldsymbol{v}_k) \tag{188}$$

$$\geqslant \mathbb{E}_k\left[\delta \left\langle \widehat{\nabla}_{\boldsymbol{\theta}}\tilde{J}_\dagger(\boldsymbol{v}_k), \nabla_{\boldsymbol{v}}\tilde{J}_\dagger(\boldsymbol{v}_k) \right\rangle + \delta \left\langle \widehat{\nabla}_\xi\tilde{J}_\dagger(\boldsymbol{v}_k), \nabla_{\boldsymbol{v}}\tilde{J}_\dagger(\boldsymbol{v}_k) \right\rangle - \frac{L_{2\dagger,\boldsymbol{v}}}{2}\delta^2 \left\|\widehat{\nabla}_{\boldsymbol{\theta}}\tilde{J}_\dagger(\boldsymbol{v}_k)\right\|_2^2 - \frac{L_{2\dagger,\boldsymbol{v}}}{2}\delta^2 \left\|\widehat{\nabla}_\xi\tilde{J}_\dagger(\boldsymbol{v}_k)\right\|_2^2\right] \tag{189}$$

$$= \delta\left(1 - \frac{L_{2\dagger,\boldsymbol{v}}\delta}{2}\right)\left\|\nabla_{\boldsymbol{\theta}}\tilde{J}_\dagger(\boldsymbol{v}_k)\right\|_2^2 + \delta\left(1 - \frac{L_{2\dagger,\boldsymbol{v}}\delta}{2}\right)\left\|\nabla_\xi\tilde{J}_\dagger(\boldsymbol{v}_k)\right\|_2^2 - \frac{L_{2\dagger,\boldsymbol{v}}\delta^2\mathcal{V}_{\dagger,\boldsymbol{\theta}}}{2Nf(\xi_k)^2} - \frac{L_{2\dagger,\boldsymbol{v}}\delta^2\mathcal{V}_{\dagger,\xi}}{2Nf(\xi_k)^2}, \tag{190}$$

where we employed the unbiasedness of the estimator and the variances bounds (Lemmas 4.3 and 7.2). Now we define $\mathcal{V}_{\dagger,\boldsymbol{v}} := \mathcal{V}_{\dagger,\boldsymbol{\theta}} + \mathcal{V}_{\dagger,\xi}$. Thus, we rewrite the previous result as:

$$\mathbb{E}_k\left[\tilde{J}_\dagger(\boldsymbol{v}_{k+1})\right] - \tilde{J}_\dagger(\boldsymbol{v}_k) \geqslant \delta\left(1 - \frac{L_{2\dagger,\boldsymbol{v}}\delta}{2}\right)\left\|\nabla_{\boldsymbol{v}}\tilde{J}_\dagger(\boldsymbol{v}_k)\right\|_2^2 - \frac{L_{2\dagger,\boldsymbol{v}}\mathcal{V}_{\dagger,\boldsymbol{v}}\delta^2}{2Nf(\xi_k)^2}. \tag{191}$$

Now that we bounded the improvement $\mathbb{E}_k\left[\tilde{J}_\dagger(\boldsymbol{v}_{k+1}) - \tilde{J}_\dagger(\boldsymbol{v}_k)\right]$, we can start handling the desired quantity:

$$\tilde{J}_\dagger(\boldsymbol{v}^*) - \mathbb{E}\left[\tilde{J}_\dagger(\boldsymbol{v}_{k+1})\right] \tag{192}$$

$$= \tilde{J}_\dagger(\boldsymbol{v}^*) - \mathbb{E}\left[\tilde{J}_\dagger(\boldsymbol{v}_k)\right] - \left(\mathbb{E}\left[\tilde{J}_\dagger(\boldsymbol{v}_{k+1}) - \tilde{J}_\dagger(\boldsymbol{v}_k)\right]\right) \tag{193}$$

$$\leqslant \tilde{J}_\dagger(\boldsymbol{v}^*) - \mathbb{E}\left[\tilde{J}_\dagger(\boldsymbol{v}_k)\right] - \mathbb{E}\left[\delta\left(1 - \frac{L_{2\dagger,\boldsymbol{v}}\delta}{2}\right)\left\|\nabla_{\boldsymbol{v}}\tilde{J}_\dagger(\boldsymbol{v}_k)\right\|_2^2 - \frac{L_{2\dagger,\boldsymbol{v}}\mathcal{V}_{\dagger,\boldsymbol{v}}\delta^2}{2Nf(\xi_k)^2}\right] \tag{194}$$

$$= \tilde{J}_\dagger(\boldsymbol{v}^*) - \mathbb{E}\left[\tilde{J}_\dagger(\boldsymbol{v}_k)\right] - \delta\left(1 - \frac{L_{2\dagger,\boldsymbol{v}}\delta}{2}\right)\mathbb{E}\left[\left\|\nabla_{\boldsymbol{v}}\tilde{J}_\dagger(\boldsymbol{v}_k)\right\|_2^2\right] + \mathbb{E}\left[\frac{L_{2\dagger,\boldsymbol{v}}\mathcal{V}_{\dagger,\boldsymbol{v}}\delta^2}{2Nf(\xi_k)^2}\right] \tag{195}$$

$$\leqslant \tilde{J}_\dagger(\boldsymbol{v}^*) - \mathbb{E}\left[\tilde{J}_\dagger(\boldsymbol{v}_k)\right] - \delta\left(1 - \frac{L_{2\dagger,\boldsymbol{v}}\delta}{2}\right)\mathbb{E}\left[\left\|\nabla_{\boldsymbol{v}}\tilde{J}_\dagger(\boldsymbol{v}_k)\right\|_2^2\right] + \frac{L_{2\dagger,\boldsymbol{v}}\mathcal{V}_{\dagger,\boldsymbol{v}}\delta^2}{2N\sigma_{\min}^2}, \tag{196}$$

where we employed the fact that the stochasticity mapping function $f$ allows the mapped $\xi$ to assume as minimum value $\sigma_{\min}$.

We can now employ Assumption 7.3 as:

$$\left\|\nabla_{\boldsymbol{v}}\tilde{J}_\dagger(\boldsymbol{v})\right\|_2 \geqslant \frac{1}{\alpha_{\boldsymbol{v}}}\max\left\{0, \tilde{J}_\dagger(\boldsymbol{v}^*) - \tilde{J}_\dagger(\boldsymbol{v}) - \beta_{\boldsymbol{v}}\right\}. \tag{197}$$

Thus, we have the following result:

$$\tilde{J}_\dagger(\boldsymbol{v}^*) - \mathbb{E}\left[\tilde{J}_\dagger(\boldsymbol{v}_{k+1})\right] \tag{198}$$

$$\leqslant \tilde{J}_\dagger(\boldsymbol{v}^*) - \mathbb{E}\left[\tilde{J}_\dagger(\boldsymbol{v}_k)\right] - \mu\delta\left(1 - \frac{L_{2\dagger,\boldsymbol{v}}\delta}{2}\right)\mathbb{E}\left[\max\left\{0, \tilde{J}_\dagger(\boldsymbol{v}^*) - \tilde{J}_\dagger(\boldsymbol{v}_k) - \beta_{\boldsymbol{v}}\right\}^2\right] + \frac{L_{2\dagger,\boldsymbol{v}}\mathcal{V}_{\dagger,\boldsymbol{v}}\delta^2}{2N\sigma_{\min}^2} \tag{199}$$

$$\leqslant \tilde{J}_\dagger(\boldsymbol{v}^*) - \mathbb{E}\left[\tilde{J}_\dagger(\boldsymbol{v}_k)\right] - \mu\delta\left(1 - \frac{L_{2\dagger,\boldsymbol{v}}\delta}{2}\right)\max\left\{0, \mathbb{E}\left[\tilde{J}_\dagger(\boldsymbol{v}^*) - \tilde{J}_\dagger(\boldsymbol{v}_k) - \beta_{\boldsymbol{v}}\right]\right\}^2 + \frac{L_{2\dagger,\boldsymbol{v}}\mathcal{V}_{\dagger,\boldsymbol{v}}\delta^2}{2N\sigma_{\min}^2}, \tag{200}$$

where we applied twice the Jensen's inequality and where we defined $\mu := \alpha_{\boldsymbol{v}}^{-2}$.

We now define $r_k := \tilde{J}_\dagger(\boldsymbol{v}^*) - \mathbb{E}\left[\tilde{J}_\dagger(\boldsymbol{v}_k)\right] - \beta_{\boldsymbol{v}}$, thus we can rewrite the previous result as:

$$r_{k+1} \leqslant r_k - \mu\delta\left(1 - \frac{L_{2\dagger,\boldsymbol{v}}\delta}{2}\right)\max\{0, r_k\}^2 + \frac{L_{2\dagger,\boldsymbol{v}}\mathcal{V}_{\dagger,\boldsymbol{v}}\delta^2}{2N\sigma_{\min}^2}. \tag{201}$$

Selecting $\delta \leqslant 1/L_{2\dagger,\boldsymbol{v}}$, we have the following recursion:

$$r_{k+1} \leqslant r_k - \frac{\mu\delta}{2}\max\{0, r_k\}^2 + \frac{L_{2\dagger,\boldsymbol{v}}\mathcal{V}_{\dagger,\boldsymbol{v}}\delta^2}{2N\sigma_{\min}^2}. \tag{202}$$

From this point on, the proof is the same as the one of Theorem F.1 by Montenegro et al. (2024). In particular, selecting a constant step size of

$$\delta = \frac{\epsilon^2 N\mu\sigma_{\min}^2}{4L_{2\dagger,\boldsymbol{v}}\mathcal{V}_{\dagger,\boldsymbol{v}}}, \tag{203}$$

and running SL-PG for a number of iterations $K$ such that

$$K \geqslant \frac{16L_{2\dagger,\boldsymbol{v}}\mathcal{V}_{\dagger,\boldsymbol{v}}}{N\mu^2\epsilon^3\sigma_{\min}^2}\log\frac{\tilde{J}_\dagger(\boldsymbol{v}^*) - \tilde{J}_\dagger(\boldsymbol{v}_0)}{\epsilon}, \tag{204}$$

it is guaranteed that

$$\tilde{J}_\dagger(\boldsymbol{v}^*) - \mathbb{E}\left[\tilde{J}_\dagger(\boldsymbol{v}_K)\right] \leqslant \epsilon + \beta_{\boldsymbol{v}}. \tag{205}$$

$\square$

G.4.3. ON THE INHERITANCE OF WEAK GRADIENT DOMINATION ON $\widetilde{J}_\dagger$

Instead of assuming Assumption 7.3 to holds, it is possible to recover the same result by exploiting Theorem 4.2. However, it is necessary to introduce the following assumptions.

**Assumption G.1** (WGD $J_\dagger$ w.r.t. $\xi$). *With a stochasticity mapping $f(\cdot)$ satisfying Assumption 7.1, there exist $\mathcal{A}_f > 0$ and $\mathcal{B}_f \geqslant 0$ such that, for every $\xi \in \mathbb{R}$, the following holds:*

$$J_\dagger^* - J_\dagger(\boldsymbol{\theta}^*(\sigma), \sigma) \leqslant \mathcal{A}_f \left\| \frac{\partial}{\partial \xi} J_\dagger(\boldsymbol{\theta}^*(\sigma), \sigma) \right\|_2 + \mathcal{B}_f,$$

*where $\sigma = f(\xi)$ and $J_\dagger^* = \max_{\boldsymbol{\theta}, \xi} J_\dagger(\boldsymbol{\theta}, f(\xi))$.*

**Assumption G.2** (Bounded Distance of to the optimal $\boldsymbol{\theta}$). *There exists $\iota \geqslant 0$ such that, for every $\boldsymbol{\theta} \in \Theta$, the following holds:*

$$\|\boldsymbol{\theta}^*(\sigma) - \boldsymbol{\theta}\|_2 \leqslant \iota(J_\dagger(\boldsymbol{\theta}^*(\sigma), \sigma) - J_\dagger(\boldsymbol{\theta}, \sigma)).$$

Sufficient conditions are requiring that the function $J_\dagger(\cdot, \sigma)$ is quasi-convex in its optimum $\boldsymbol{\theta}^*(\sigma)$ (e.g., star-convex functions, Lee & Valiant 2016).

The following result shows that under Assumptions G.1 and G.2, and the assumptions of Theorem 4.2 and Lemma G.9, it is possible to recover the same condition imposed by Assumption 7.3.

**Theorem G.14** (($\boldsymbol{\theta}, \xi$) Joint Weak Gradient Domination). *Consider a (hyper)policy complying with Definitions 2.2 (**AB**) or 2.3 (**PB**), and assume to be under regularity assumptions for the MDP (Asm. D.1) and the deterministic policy $\mu_{\boldsymbol{\theta}}$ (Asm. D.2). Under Assumptions 4.1, 4.2, 5.1 (**PB**) or 5.2 (**AB**), 7.1, G.1, G.2, and 7.2, for every parameterization $\boldsymbol{\theta} \in \Theta$ and exploration parameterization $\xi \in \mathbb{R}$, it holds that:*

$$\widetilde{J}_\dagger(\boldsymbol{v}^*) - \widetilde{J}_\dagger(\boldsymbol{v}) \leqslant \mathcal{A}_{\dagger, \boldsymbol{v}, f} \left\| \nabla_{\boldsymbol{v}} \widetilde{J}_\dagger(\boldsymbol{v}) \right\|_2 + \mathcal{B}_{\dagger, \boldsymbol{v}, f}(f(\xi)).$$

*where $\mathcal{A}_{\dagger, \boldsymbol{v}, f} := 2\alpha_D \left(1 + \mathcal{A}_f L_{3\dagger, \xi}\iota\right)$ and $\mathcal{B}_{\dagger, \boldsymbol{v}, f}(f(\xi)) := \beta_D + \beta_\dagger(f(\xi)) \left(1 + \mathcal{A}_f L_{3\dagger, \xi}\iota\right)$.*

*Proof.* Under this set of Assumptions, we have that Theorem 4.2 holds, that is:

$$J_\dagger(\boldsymbol{\theta}^*(\sigma), \sigma) - J_\dagger(\boldsymbol{\theta}, \sigma) \leqslant \alpha_D \|\nabla_{\boldsymbol{\theta}} J_\dagger(\boldsymbol{\theta}, \sigma)\|_2 + \beta_\dagger(\sigma), \tag{206}$$

where $\beta_\dagger(\sigma) = \beta_D + D_\dagger \sigma$, and $\alpha_D$ and $\beta_D$ are the WGD constants of Assumption 4.2.

Moreover, we report Assumption G.1:

$$J_\dagger(\boldsymbol{\theta}^*(\sigma^*), \sigma^*) - J_\dagger(\boldsymbol{\theta}^*(\sigma), \sigma) \leqslant \mathcal{A}_f \left| \frac{\partial}{\partial \xi} J_\dagger(\boldsymbol{\theta}^*(\sigma), \sigma) \right| + \mathcal{B}_f, \tag{207}$$

where we use $\frac{\partial}{\partial \xi} J_\dagger(\boldsymbol{\theta}^*(\sigma), \sigma)$ as a shortcut for $\frac{\partial}{\partial \xi} J_\dagger(\boldsymbol{\theta}, \sigma)|_{\boldsymbol{\theta} = \boldsymbol{\theta}^*(\sigma)}$.

Our goal is to find an upper bound to the quantity:

$$\widetilde{J}_\dagger(\boldsymbol{v}^*) - \widetilde{J}_\dagger(\boldsymbol{v}) = J_\dagger(\boldsymbol{\theta}^*(f(\xi^*)), f(\xi^*)) - J_\dagger(\boldsymbol{\theta}, f(\xi)), \tag{208}$$

for every $\boldsymbol{v} \in \mathbb{R}^{d_\Theta + 1}$.

Thus, we can start from the latter quantity:

$$J_\dagger(\boldsymbol{\theta}^*(f(\xi^*)), f(\xi^*)) - J_\dagger(\boldsymbol{\theta}, f(\xi)) \tag{209}$$

$$= J_\dagger(\boldsymbol{\theta}^*(f(\xi^*)), f(\xi^*)) - J_\dagger(\boldsymbol{\theta}, f(\xi)) \pm J_\dagger(\boldsymbol{\theta}^*(f(\xi)), f(\xi)) \tag{210}$$

$$\leqslant \mathcal{A}_f \left| \frac{\partial}{\partial \xi} J_\dagger(\boldsymbol{\theta}^*(f(\xi)), f(\xi)) \right| + \alpha_D \|\nabla_{\boldsymbol{\theta}} J_\dagger(\boldsymbol{\theta}, f(\xi))\|_2 + \beta_D + \beta_\dagger(f(\xi)), \tag{211}$$

where we just applied Assumption G.1 and the result of Theorem 4.2.

Now, we focus on the quantity $\left| \frac{\partial}{\partial \xi} J_\dagger(\boldsymbol{\theta}^*(f(\xi)), f(\xi)) \right|$. In particular, the following holds:

$$\left| \frac{\partial}{\partial \xi} J_\dagger(\boldsymbol{\theta}^*(f(\xi)), f(\xi)) \right| \tag{212}$$

$$\leqslant \left| \frac{\partial}{\partial \xi} J_\dagger(\boldsymbol{\theta}^*(f(\xi)), f(\xi)) - \frac{\partial}{\partial \xi} J_\dagger(\boldsymbol{\theta}, f(\xi)) \right| + \left| \frac{\partial}{\partial \xi} J_\dagger(\boldsymbol{\theta}, f(\xi)) \right| \tag{213}$$

$$\leqslant L_{3\dagger,\xi} \left\| \boldsymbol{\theta}^*(f(\xi)) - \boldsymbol{\theta} \right\|_2 + \left| \frac{\partial}{\partial \xi} J_\dagger(\boldsymbol{\theta}, f(\xi)) \right| \tag{214}$$

$$\leqslant L_{3\dagger,\xi}\iota \left| J_\dagger(\boldsymbol{\theta}^*(f(\xi)), f(\xi)) - J_\dagger(\boldsymbol{\theta}, f(\xi)) \right| + \left| \frac{\partial}{\partial \xi} J_\dagger(\boldsymbol{\theta}, f(\xi)) \right| \tag{215}$$

$$\leqslant L_{3\dagger,\xi}\iota\alpha_{\mathrm{D}} \left\| \nabla_{\boldsymbol{\theta}} J_\dagger(\boldsymbol{\theta}, \sigma) \right\|_2 + L_{3\dagger,\xi}\iota\beta_\dagger(f(\xi)) + \left| \frac{\partial}{\partial \xi} J_\dagger(\boldsymbol{\theta}, f(\xi)) \right|, \tag{216}$$

where in the second inequality we employed Lemma G.9, in the third inequality we employed Assumption G.2, and in the last inequality we exploited the result of Theorem 4.2.

Given this last result, we have what follows:

$$J_\dagger(\boldsymbol{\theta}^*(f(\xi^*)), f(\xi^*)) - J_\dagger(\boldsymbol{\theta}, f(\xi)) \tag{217}$$

$$\leqslant \mathcal{A}_f \left| \frac{\partial}{\partial \xi} J_\dagger(\boldsymbol{\theta}^*(f(\xi)), f(\xi)) \right| + \alpha_{\mathrm{D}} \left\| \nabla_{\boldsymbol{\theta}} J_\dagger(\boldsymbol{\theta}, f(\xi)) \right\|_2 + \beta_{\mathrm{D}} + \beta_\dagger(f(\xi)) \tag{218}$$

$$\leqslant \mathcal{A}_f \left| \frac{\partial}{\partial \xi} J_\dagger(\boldsymbol{\theta}, f(\xi)) \right| + (\mathcal{A}_f L_{3\dagger,\xi}\iota\alpha_{\mathrm{D}} + \alpha_{\mathrm{D}}) \left\| \nabla_{\boldsymbol{\theta}} J_\dagger(\boldsymbol{\theta}, \sigma) \right\|_2 + \beta_{\mathrm{D}} + \beta_\dagger(f(\xi)) + \mathcal{A}_f L_{3\dagger,\xi}\iota\beta_\dagger(f(\xi)) \tag{219}$$

$$\leqslant \alpha_{\mathrm{D}} \left(1 + \mathcal{A}_f L_{3\dagger,\xi}\iota\right) \left( \left| \frac{\partial}{\partial \xi} J_\dagger(\boldsymbol{\theta}, f(\xi)) \right| + \left\| \nabla_{\boldsymbol{\theta}} J_\dagger(\boldsymbol{\theta}, \sigma) \right\|_2 \right) + \beta_{\mathrm{D}} + \beta_\dagger(f(\xi)) \left(1 + \mathcal{A}_f L_{3\dagger,\xi}\iota\right). \tag{220}$$

The last step to conclude the proof is to bound the term $\left| \frac{\partial}{\partial \xi} J_\dagger(\boldsymbol{\theta}, f(\xi)) \right| + \left\| \nabla_{\boldsymbol{\theta}} J_\dagger(\boldsymbol{\theta}, \sigma) \right\|_2$. To this end, we recall that $\boldsymbol{v} = (\boldsymbol{\theta}, \xi)^\top$ and that:

$$\nabla_{\boldsymbol{v}} \tilde{J}_\dagger(\boldsymbol{v}) := \left( \frac{\partial \tilde{J}_\dagger(\boldsymbol{v})}{\partial \theta_1}, ..., \frac{\partial \tilde{J}_\dagger(\boldsymbol{v})}{\partial \theta_{d_\Theta}}, \frac{\partial \tilde{J}_\dagger(\boldsymbol{v})}{\partial \xi} \right)^\top, \quad \nabla_{\boldsymbol{\theta}} \tilde{J}_\dagger(\boldsymbol{v}) := \left( \frac{\partial \tilde{J}_\dagger(\boldsymbol{v})}{\partial \theta_1}, ..., \frac{\partial \tilde{J}_\dagger(\boldsymbol{v})}{\partial \theta_{d_\Theta}}, 0 \right)^\top, \quad \nabla_\xi \tilde{J}_\dagger(\boldsymbol{v}) := \left( \mathbf{0}_{d_\Theta}, \frac{\partial \tilde{J}_\dagger(\boldsymbol{v})}{\partial \xi} \right)^\top.$$

Thus, the following hold:

$$\left| \frac{\partial}{\partial \xi} J_\dagger(\boldsymbol{\theta}, f(\xi)) \right| = \left\| \nabla_\xi \tilde{J}_\dagger(\boldsymbol{v}) \right\|_2 \leqslant \left\| \nabla_{\boldsymbol{v}} \tilde{J}_\dagger(\boldsymbol{v}) \right\|_2. \tag{221}$$

because of the form of the treated gradients. Analogously, we have:

$$\left\| \nabla_{\boldsymbol{\theta}} J_\dagger(\boldsymbol{\theta}, \sigma) \right\|_2 = \left\| \nabla_{\boldsymbol{\theta}} \tilde{J}_\dagger(\boldsymbol{v}) \right\|_2 \leqslant \left\| \nabla_{\boldsymbol{v}} \tilde{J}_\dagger(\boldsymbol{v}) \right\|_2. \tag{222}$$

In light of these results, we can conclude what follows:

$$J_\dagger(\boldsymbol{\theta}^*(f(\xi^*)), f(\xi^*)) - J_\dagger(\boldsymbol{\theta}, f(\xi)) \tag{223}$$

$$= \tilde{J}_\dagger(\boldsymbol{v}^*) - \tilde{J}_\dagger(\boldsymbol{v}) \tag{224}$$

$$\leqslant \alpha_{\mathrm{D}} \left(1 + \mathcal{A}_f L_{3\dagger,\xi}\iota\right) \left( \left| \frac{\partial}{\partial \xi} J_\dagger(\boldsymbol{\theta}, f(\xi)) \right| + \left\| \nabla_{\boldsymbol{\theta}} J_\dagger(\boldsymbol{\theta}, \sigma) \right\|_2 \right) + \beta_{\mathrm{D}} + \beta_\dagger(f(\xi)) \left(1 + \mathcal{A}_f L_{3\dagger,\xi}\iota\right) \tag{225}$$

$$\leqslant 2\alpha_{\mathrm{D}} \left(1 + \mathcal{A}_f L_{3\dagger,\xi}\iota\right) \left\| \nabla_{\boldsymbol{v}} \tilde{J}_\dagger(\boldsymbol{v}) \right\|_2 + \beta_{\mathrm{D}} + \beta_\dagger(f(\xi)) \left(1 + \mathcal{A}_f L_{3\dagger,\xi}\iota\right), \tag{226}$$

which concludes the proof. $\qquad\square$

### G.4.4. CONVERGENCE UNDER THE CONDITIONS FOR WGD INHERITANCE ON $\tilde{J}_\dagger$

**Theorem G.15** (SL-PG Convergence Under WGD Inheritance). *If the (hyper)policy satisfies Definitions 2.2 or 2.3, the output parameterization $\boldsymbol{v}_K$ of* SL-PG *is such that: under Assumptions 4.1, 4.2, 4.3, 4.4, 5.1 (**PB**) or 5.2 (**AB**), 5.3, 7.1, 7.2, G.1, and G.2, running* SL-PG *for $K$ iterations, with a suitable choice for the learning rate $\delta$, the output parameterization $\boldsymbol{v}_K$ is such that:*

$$\tilde{J}_\dagger^* - \mathbb{E}[\tilde{J}_\dagger(\boldsymbol{v}_K)] \leqslant \epsilon + \mathcal{B}_{\dagger,\boldsymbol{v},f}(\sigma_{\max}),$$

*with a total sample complexity of:*

$$NK = \tilde{\mathcal{O}} \left( \frac{16 \mathcal{A}_{\dagger,\boldsymbol{v},f}^4 L_{2\dagger,\boldsymbol{v}} \mathcal{V}_{\dagger,\boldsymbol{v}}}{\epsilon^3 \sigma_{\min}^2} \right),$$

*where $L_{2\dagger,\boldsymbol{v}} := L_2 + L_{2\dagger,\xi}$ and $\mathcal{V}_{\dagger,\boldsymbol{v}} := \mathcal{V}_{\dagger,\boldsymbol{\theta}} + \mathcal{V}_{\dagger,\xi}$.*

*Proof.* The proof is the same as the one of Theorem 7.3, except for the fact that, instead employing Assumption 7.3 to bound $\|\nabla_{\boldsymbol{v}}\tilde{J}_{\dagger}(\boldsymbol{v})\|^2$, we use the result of Theorem G.14 as:

$$\left\|\nabla_{\boldsymbol{v}}\tilde{J}_{\dagger}(\boldsymbol{v})\right\|_2 \geqslant \frac{1}{\mathcal{A}_{\dagger,\boldsymbol{v},f}} \max\left\{0, \tilde{J}_{\dagger}^* - \tilde{J}_{\dagger}(\boldsymbol{v}) - \mathcal{B}_{\dagger,\boldsymbol{v},f}(f(\xi))\right\} \tag{227}$$

$$\geqslant \frac{1}{\mathcal{A}_{\dagger,\boldsymbol{v},f}} \max\left\{0, \tilde{J}_{\dagger}^* - \tilde{J}_{\dagger}(\boldsymbol{v}) - \mathcal{B}_{\dagger,\boldsymbol{v},f}(\sigma_{\max})\right\}, \tag{228}$$

having exploited the fact that, by definition, the stochasticity mapping function $f(\cdot)$ ensures that the maximum allowed value for $\sigma$ is $\sigma_{\max}$. Now, by defining $r_k := \tilde{J}_{\dagger}^* - \mathbb{E}[\tilde{J}_{\dagger}(\boldsymbol{v}_k)] - \mathcal{B}_{\dagger,\boldsymbol{v},f}(\sigma_{\max})$, the proof continues as the one of Theorem 7.3. $\quad\square$

**Comment.** It is worth noticing that this result comes at the cost of the very demanding Assumption G.2. Moreover, the result suggests that in the final suboptimality there is always a term which is directly proportional to the maximum value that the $\sigma$ can assume due to the mapping $f(\cdot)$. With this analysis, this term, which comes from Theorems 4.2 and G.14, would not disappear even under the assumption that all the $\epsilon$-optimal stochastic (hyper)policies have the stochasticity less than $\epsilon$, thus setting $\sigma_{\min} = \epsilon$. To make this additional quantity disappear, one has to assume the (very) demanding strong gradient domination on $J_{\dagger}$ w.r.t. both $\boldsymbol{\theta}$ and $\xi$.

| AB and PB Configurations for *Swimmer-v5* | |
|---|---|
| Environment | *Swimmer-v5* (Todorov et al., 2012) |
| Horizon | $T = 200$ |
| Dimensions | $d_\mathcal{A} = 2 \quad d_\mathcal{S} = 8 \quad d_\Theta = 16$ |
| (Hyper)policy | Linear Gaussian |
| Learning rates (Adam, Kingma & Ba 2014) | 0.01 |
| Batch size | $N = 100$ |
| PES Schedule A | $\sigma_{\max} = 1 \quad y = 1 \quad P = 25 \quad K_p = 200$ |
| PES Schedule B | $\sigma_{\max} = 1 \quad y = 0.5 \quad P = 5000 \quad K_p = 1$ |
| SL-PG | $\sigma_{\max} = 1 \quad \sigma_{\min} = 0 \quad K = 5000$ |
| Static $\sigma$ (AB and PB) | $\sigma \in \{1, 0.5, 0.04, 0.014\} \quad K = 5000$ |

*Table 2.* Experimental details for the numerical validation presented in Section 9.

## H. Additional Experiments

### H.1. Employed Policies and Hyperpolicies

**Linear Deterministic Policy.** A *linear parametric deterministic* policy $\mu_{\boldsymbol{\theta}} : \mathcal{S} \to \mathcal{A}$ samples the actions as $\boldsymbol{a}_t = \boldsymbol{\theta}^\top \boldsymbol{s}_t$, where $\boldsymbol{s}_t$ is the observed state at time $t$ and $\boldsymbol{\theta}$ is the parameter vector.

**Linear Gaussian Policy.** A *linear parametric gaussian* policy $\pi_{\boldsymbol{\theta},\sigma} : \mathcal{S} \times \mathcal{A} \to \Delta(\mathcal{A})$ with variance $\sigma^2$ samples the actions as $\boldsymbol{a}_t \sim \mathcal{N}(\boldsymbol{\theta}^\top \boldsymbol{s}_t, \sigma^2 I_{d_\mathcal{A}})$, where $\boldsymbol{s}_t$ is the observed state at time $t$ and $\boldsymbol{\theta}$ is the parameter vector. Notice that in this case we consider $\boldsymbol{\theta} \in \mathbb{R}^{d_\mathcal{S} \times d_\mathcal{A}}$. For this policy, the scores are the following:

$$\nabla_{\boldsymbol{\theta}} \log \pi_{\boldsymbol{\theta},\sigma}(\boldsymbol{a}) = \frac{\left((\boldsymbol{a} - \boldsymbol{\theta}^\top \boldsymbol{s})\boldsymbol{s}^\top\right)^\top}{\sigma^2}; \quad \frac{\partial}{\partial \sigma} \log \pi_{\boldsymbol{\theta},\sigma}(\boldsymbol{a}) = \frac{\|\boldsymbol{a} - \boldsymbol{\theta}^\top \boldsymbol{s}\|_2^2 - d_\mathcal{A}\sigma^2}{\sigma^3}; \quad \frac{\partial}{\partial \xi} \log \pi_{\boldsymbol{\theta},\sigma}(\boldsymbol{a}) = \frac{\|\boldsymbol{a} - \boldsymbol{\theta}^\top \boldsymbol{s}\|_2^2 - d_\mathcal{A}\sigma^2}{\sigma^2},$$

where the considered parameterization for the stochasticity, i.e., $\sigma = e^\xi$.

**Gaussian Hyperpolicy.** A *parametric gaussian* hyperpolicy $\nu_{\boldsymbol{\theta},\sigma} \in \Delta(\Theta)$ with variance $\sigma^2$ samples the parameters $\boldsymbol{\theta}'$ for the underlying linear deterministic parametric policy $\mu_{\boldsymbol{\theta}}$ as $\boldsymbol{\theta}' \sim \mathcal{N}(\boldsymbol{\theta}, \sigma^2 I_{d_\Theta})$, where $\boldsymbol{\theta}$ is the parameter vector for the hyperpolicy. For this hyperpolicy, the scores are the following:

$$\nabla_{\boldsymbol{\theta}} \log \nu_{\boldsymbol{\theta},\sigma}(\boldsymbol{\theta}') = \frac{\boldsymbol{\theta}' - \boldsymbol{\theta}}{\sigma^2}; \quad \frac{\partial}{\partial \sigma} \log \nu_{\boldsymbol{\theta},\sigma}(\boldsymbol{\theta}') = \frac{\|\boldsymbol{\theta}' - \boldsymbol{\theta}\|_2^2 - d_\Theta\sigma^2}{\sigma^3}; \quad \frac{\partial}{\partial \xi} \log \nu_{\boldsymbol{\theta},\sigma}(\boldsymbol{\theta}') = \frac{\|\boldsymbol{\theta}' - \boldsymbol{\theta}\|_2^2 - d_\Theta\sigma^2}{\sigma^2},$$

where the considered parameterization for the stochasticity, i.e., $\sigma = e^\xi$.

### H.2. Experiment: Swimmer (Details of Section 9)

Here, we report the details for the experiment presented in Section 9. In particular, Table 2 reports the experimental details, Table 3 report the deterministic policy deployment scores, and Figures 3a and 3b report the values of $J_\dagger$ and $\sigma$ during the learning process.

### H.3. Experiment: Inverted Pendulum

Here, we present a similar experiment w.r.t. the one shown in Section 9. In particular, we analyze the behavior of PES and SL-PG in both the AB and PB exploration domains, comparing them with their static stochasticity counterparts (GPOMDP and PGPE) in the context of deploying deterministic policies. We conduct the evaluations in the *InvertedPendulum-v5* environment, which is part of the MuJoCo (Todorov et al., 2012) control suite, using a horizon of $T = 200$. All learning rates are managed by the Adam (Kingma & Ba, 2014) optimizer.

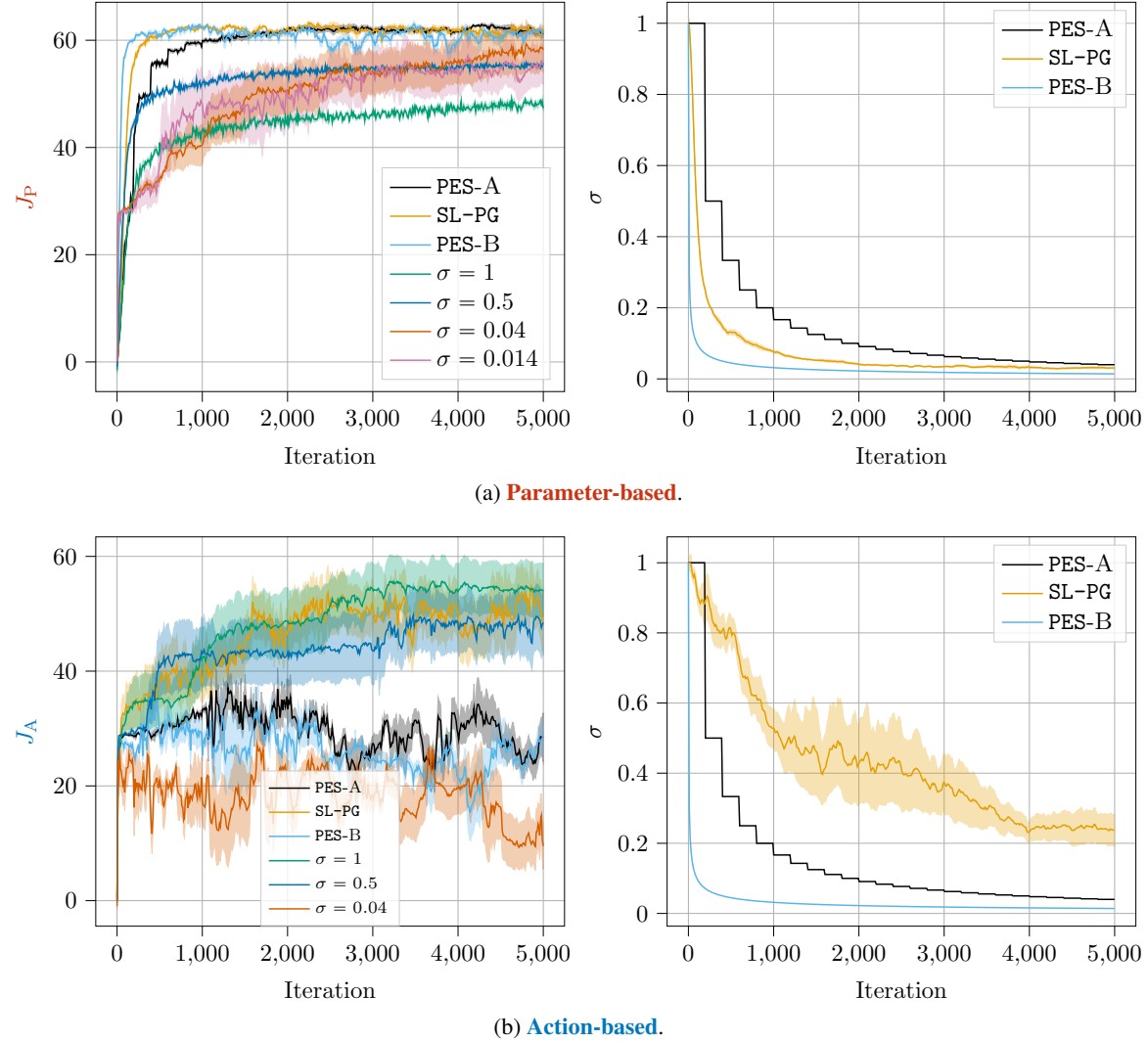

(a) **Parameter-based**.

(b) **Action-based**.

Figure 3. $J_\dagger$ and $\sigma$ behavior for **PB** and **AB** in *Swimmer-v5* (5 runs, mean $\pm 95\%$ C.I.).

| Method | PB | $\Delta_{\mathrm{P}}$ | AB | $\Delta_{\mathrm{A}}$ |
|---|---|---|---|---|
| PES-A | $61.14 \pm 1.1$ | $-0.12$ | $28.93 \pm 9.89$ | $0.44$ |
| SL-PG | $61.43 \pm 2.73$ | $-0.21$ | $49.83 \pm 13.38$ | $-1.19$ |
| PES-B | $60.40 \pm 2.31$ | $-0.27$ | $29.24 \pm 8.74$ | $0.33$ |

Table 3. *Swimmer-v5*: deterministic deployment performance (5 runs over 100 trajectories, mean $\pm$ std). $\Delta_\dagger = J_{\mathrm{D}}(\boldsymbol{\theta}_K) - J_\dagger(\boldsymbol{\theta}_K)$.

For both exploration paradigms, we present PES with two different schedules, both starting with $\sigma = 1$. The first (A) schedule consists of $P = 25$ phases, each lasting $K_p = 50$ iterations, with a schedule exponent of $y = 1$. The second (B) schedule includes $P = 1250$ phases, each lasting $K_p = 1$ iteration, with a schedule exponent of $y = 0.5$. SL-PG is executed for $K = 1250$ iterations, using the common exponential parameterization for $\sigma$ (i.e., $\sigma = e^\xi$). The static stochasticity counterparts are also run for $K = 1250$ iterations, employing stochasticity levels $\sigma \in \{1, 0.5, 0.04, 0.028\}$. Here, $\sigma = 1$ represents the maximum stochasticity in the PES schedules, while $\sigma = 0.04$ and $\sigma = 0.028$ correspond to the minima of the first and second PES schedules, respectively.

**Parameter-based**. As shown by Figure 4a, in the **PB** exploration domain, PES-B (the one with the continuous schedule) outperforms the other methods in terms of convergence to optimal performance, while PES-A (the one who mimics what

| AB and PB Configurations for *InvertedPendulum-v5* | |
|---|---|
| Environment | *InvertedPendulum-v5* (Todorov et al., 2012) |
| Horizon | $T = 200$ |
| Dimensions | $d_\mathcal{A} = 1 \quad d_\mathcal{S} = 4 \quad d_\Theta = 4$ |
| (Hyper)policy | Linear Gaussian |
| Learning rates (Adam, Kingma & Ba 2014) | 0.01 |
| Batch size | $N = 100$ |
| PES Schedule A | $\sigma_{\max} = 1 \quad y = 1 \quad P = 25 \quad K_p = 50$ |
| PES Schedule B | $\sigma_{\max} = 1 \quad y = 0.5 \quad P = 1250 \quad K_p = 1$ |
| SL-PG | $\sigma_{\max} = 1 \quad \sigma_{\min} = 0 \quad K = 1250$ |
| Static $\sigma$ (AB and PB) | $\sigma \in \{1, 0.5, 0.04, 0.028\} \quad K = 1250$ |

*Table 4.* Experimental details for the numerical validation presented in Appendix H.3.

| Method | PB | $\Delta_P$ | AB | $\Delta_A$ |
|---|---|---|---|---|
| PES-A | $200 \pm 0$ | 0.08 | $200 \pm 0$ | 0.21 |
| SL-PG | $199.97 \pm 0.06$ | 0.6 | $200 \pm 0$ | 0.018 |
| PES-B | $198.37 \pm 2.3$ | 0.53 | $200 \pm 0$ | 0 |

*Table 5. InvertedPendulum-v5*: deterministic deployment performance (5 runs over 100 trajectories, mean $\pm$ std). Setting of Table 4. $\Delta_\dagger = J_D(\boldsymbol{\theta}_K) - J_\dagger(\boldsymbol{\theta}_K)$.

prescribed by theory) exhibits similar behavior to SL-PG. In terms of deterministic deployment, PES-A is the one outputting the best performing deterministic policy. In this case, the worst performing one PES-B, which employs a schedule not compliant with what prescribed by the PES theoretical tractation.

**Action-based.** This pattern does not hold in the AB exploration domain, as illustrated in Figure 4b. In this case, the GPOMDP instances with the lowest static stochasticity levels converge faster than their dynamic $\sigma$ counterparts. However, this faster convergence comes at the expense of manually tuning the stochasticity levels, which is not required for approaches that employ a dynamic $\sigma$. For what concerns deterministic deployment, all the dynamic stochasticity methods reach the optimal deterministic policy (see Table 5).

**Final $\sigma$.** In general, while the final stochasticity level of PES is controlled by the imposed schedule, the final stochasticity of SL-PG remains uncertain until the end of the learning process, as it is learned via stochastic gradient ascent. In this specific case, the final parameterization learned by PES-A, that follows what suggested by theory since $K_p > 1$, ensures that the loss incurred when switching off the noise is smaller compared to that incurred when using SL-PG or PES-B. This difference, which surprisingly is not holding for the AB case, is highlighted in Table 5.

## H.4. Reacher-v5 Study

Similarly to what done in Section 9, here we analyze the behavior of PES and SL-PG in both the AB and PB exploration domains, comparing them with their static stochasticity counterparts (GPOMDP and PGPE) in the context of deploying deterministic policies. This time, the evaluations are conducted in the *Reacher-v5* environment, always part of the MuJoCo (Todorov et al., 2012) control suite, using its default horizon $T = 50$. All learning rates are managed by the Adam (Kingma & Ba, 2014) optimizer.

For both exploration paradigms, we present PES with two different schedules, both starting with $\sigma = 0.5$. The first schedule (A) consists of $P = 25$ phases, each lasting $K_p = 200$ iterations, with a schedule exponent of $y = 1$. The second schedule (B) includes $P = 5000$ phases, each lasting $K_p = 1$ iteration, with a schedule exponent of $y = 0.5$. SL-PG is executed for $K = 5000$ iterations, using the common exponential parameterization for $\sigma$ (i.e., $\sigma = e^\xi$). The static stochasticity

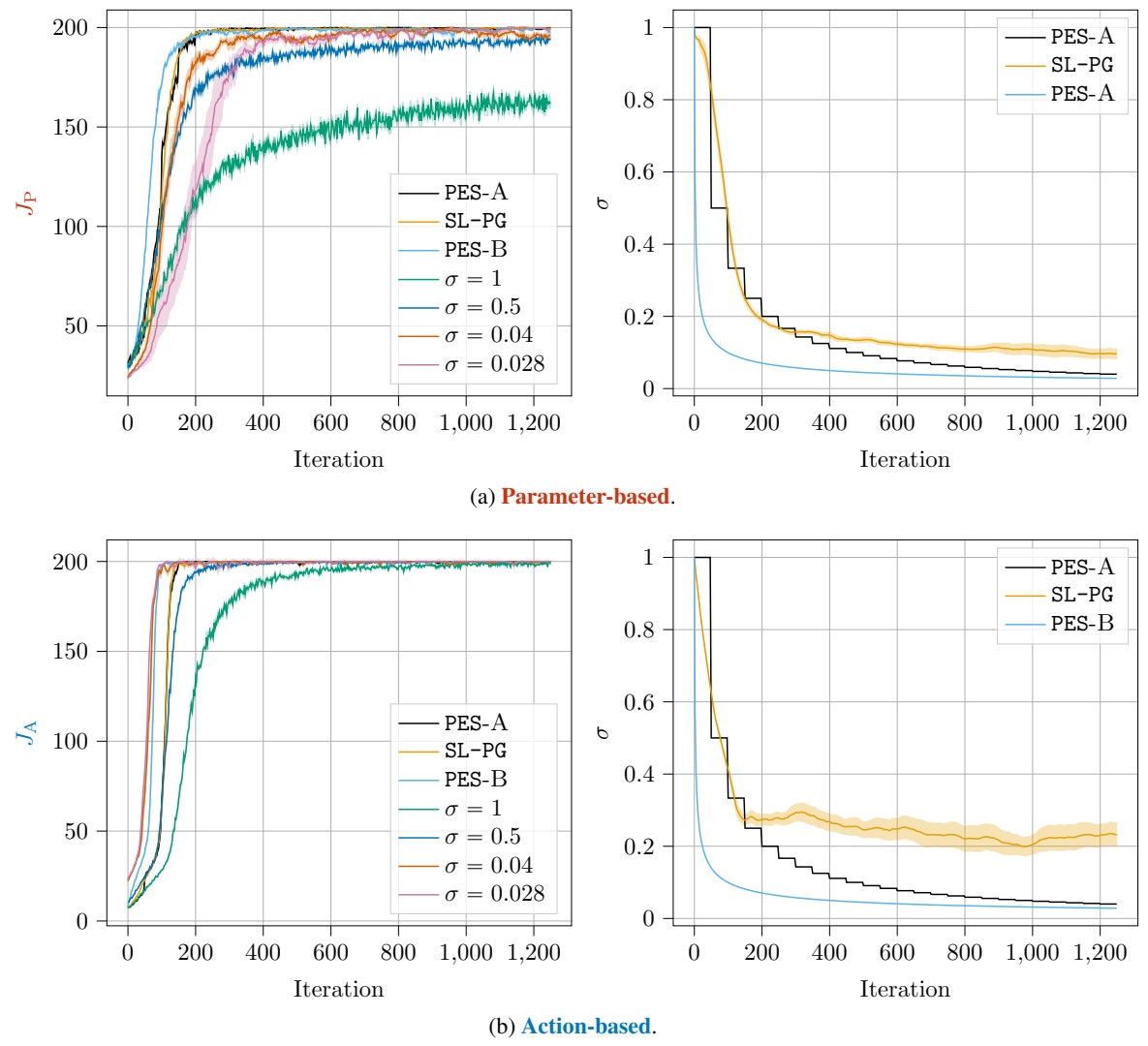

(a) **Parameter-based**.

(b) **Action-based**.

Figure 4. $J_{\dagger}$ and $\sigma$ behavior for **PB** and **AB** in *InvertedPendulum-v5* (5 runs, mean $\pm 95\%$ C.I.).

counterparts are also run for $K = 5000$ iterations, employing stochasticity levels $\sigma \in \{0.5, 0.08, 0.002\}$. Here, $\sigma = 1$ represents the maximum stochasticity in the PES schedules, while $\sigma = 0.002$ correspond to the minimum of the first PES schedule. All the experimental details are presented in Table 6.

**Parameter-based.** As shown in Figure 5a, in the **PB** exploration domain, PES-B (with the continuous schedule) outperforms the other methods in terms of convergence to optimal performance, but the PGPE instance with fixed stochasticity $\sigma = \sigma_{\min}$. Also in this scenario, PES-A exhibits a slower performance convergence, this can be explained by the fact that, by construction of the stochasticity schedule, it keeps the same $\sigma$ value fore more iterations. This behavior is more similar to the one prescribed by theoretical algorithm construction (Sec. 3). It is worth noticing that SL-PG outputs a parameterization associated with a $\sigma_K$ which is $\approx 12$ times larger than the final stochasticity of PES-A and $\approx 35$ times the final $\sigma$ of PES-B.

**Action-based.** Figure 5b shows the same results in the **AB** exploration context. The figure highlights the need of having a dynamic stochasticity schedule, since GPOMDP instances with fixed stochasticity levels do not manage to optimize the parameters of the linear policy, except the one with $\sigma = 0.02$. This last instance shows a similar behavior to PES-B, which, at the end of the learning procedure, ensures to have a good parameterization for a smaller stochasticity $\sigma = 0.007$. The algorithm reaching the largest performance is PES-A, which benefits from keeping the stochasticity level fixed for more than one iteration, however resulting in a good parameterization for a stochasticity $\sigma = 0.02$. The slowest one to converge is SL-PG, which output a good parameterization for a stochasticity $\sigma \approx 0.08$.

**Final $\sigma$.** In general, while the final stochasticity level of PES is controlled by the imposed schedule, the final stochasticity

| AB and PB Configurations for *Reacher-v5* | |
|---|---|
| Environment | *Reacher-v5* (Todorov et al., 2012) |
| Horizon | $T = 50$ |
| Dimensions | $d_{\mathcal{A}} = 2 \quad d_{\mathcal{S}} = 10 \quad d_{\Theta} = 20$ |
| (Hyper)policy | Linear Gaussian |
| Learning rates (Adam, Kingma & Ba 2014) | 0.001 |
| Batch size | $N = 100$ |
| PES Schedule A | $\sigma_{\max} = 0.5 \quad y = 1 \quad P = 25 \quad K_p = 200$ |
| PES Schedule B | $\sigma_{\max} = 0.5 \quad y = 0.5 \quad P = 5000 \quad K_p = 1$ |
| SL-PG | $\sigma_{\max} = 0.5 \quad \sigma_{\min} = 0 \quad K = 5000$ |
| Static $\sigma$ (AB and PB) | $\sigma \in \{0.5, 0.08, 0.002\} \quad K = 5000$ |

*Table 6.* Experimental details for the numerical validation presented in Appendix H.4.

| Method | PB | $\Delta_{\mathrm{P}}$ | AB | $\Delta_{\mathrm{A}}$ |
|---|---|---|---|---|
| PES-A | $-11.47 \pm 0.35$ | 0.37 | $-7.38 \pm 0.2$ | 0.02 |
| SL-PG | $-102.85 \pm 8.1$ | 187.13 | $-6.83 \pm 0.34$ | 0.66 |
| PES-B | $-8.73 \pm 0.29$ | 0.06 | $-9.66 \pm 0.74$ | $-0.01$ |

*Table 7.* Deterministic deployment performance (5 runs over 100 trajectories, mean $\pm$ std). Setting of Table 6. $\Delta_{\dagger} = J_{\mathrm{D}}(\boldsymbol{\theta}_K) - J_{\dagger}(\boldsymbol{\theta}_K)$.

of SL-PG remains uncertain until the end of the learning process, as it is learned via stochastic gradient ascent. Table 7 shows the deterministic deployment performances of the learned parameterization by PES-A, PES-B, and SL-PG. For the AB case the best performing is PES-B, which ensures the lowest level of stochasticity at the end of the learning process. Switching to the AB scenario, the best performing is SL-PG. This fact enforces the consideration that SL-PG cannot ensure always to converge to a particular value of the stochasticity level, which can be crucial for the deterministic deployment performance, as it is the case for this environment.

**On the Semantic Difference between PB and AB.** Finally, in this experiment it is possible to appreciate the semantic difference between the PB and AB exploration paradigms. Indeed, Figures 5a and 5b highlight the different scale of performance values $J_{\mathrm{P}}$ and $J_{\mathrm{A}}$ corresponding to hyperpolicies and policies w.r.t. the same stochasticity level $\sigma$.

### H.5. PES Sensitivity Analysis: $y$ Parameter

In this section, we conduct an experiment on the sensitivity of PES to the $y$ parameter. We run PES in the *InvertedPendulum-v5* environment with a horizon $T = 100$. The PES instances employ a schedule with $\sigma_{\max} = 1$, $K_p = 40$, and $P = 25$. Learning rates are managed by Adam and initialized at $\zeta = 0.01$. We recall that the parameter $y$ controls the smoothness of the schedule, determining, having fixed $K_p$, $P$, and $\sigma_{\max}$, the final value for the $\sigma$. In these experiments, we employed three schedules, differing in the $y$ value: (A) has $y = 1$, (B) has $y = 0.5$, finally (C) has $y = 0.1$. The details for this experimental setting are summarized in Table 8.

Figures 6a and 6b show that, in both AB and PB exploration domains, it is possible to see that PES-C is the one showing the poorest convergence speed. Moreover, its final performance is the lowest one. The best performing one i PES-A, which mimics the behavior prescribed by theory. As also observed in the experiments conducted in Section 9, and Appendices H.3 and H.4, a smaller value for $y$ should be associated with phases running a smaller amount of iterations. This is in order to provide a final value for $\sigma$ that is the same across schedules, given a total amount of iterations $K = \sum_{p=0}^{P-1} K_p$.

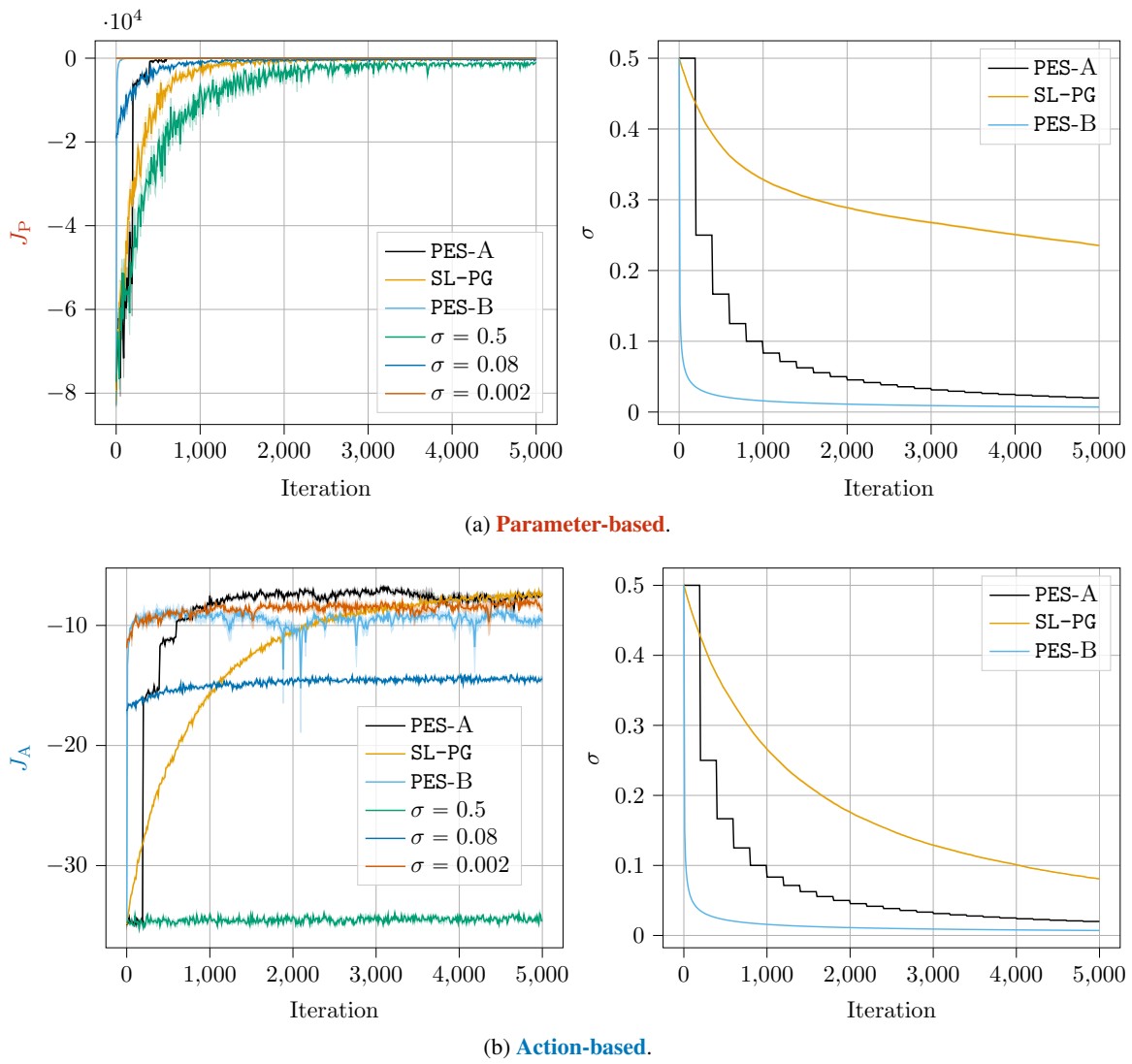

(a) **Parameter-based**.

(b) **Action-based**.

*Figure 5.* $J_\dagger$ and $\sigma$ behavior for **PB** and **AB** in *Reacher-v5* (5 runs, mean $\pm 95\%$ C.I.).

| AB and PB Configurations for PES Sensitivity Study on $y$ | |
|---|---|
| Environment | *InvertedPendulum-v5* (Todorov et al., 2012) |
| Horizon | $T = 100$ |
| Dimensions | $d_\mathcal{A} = 1 \quad d_\mathcal{S} = 4 \quad d_\Theta = 4$ |
| (Hyper)policy | Linear Gaussian |
| Learning rates (Adam, Kingma & Ba 2014) | 0.01 |
| Batch size | $N = 100$ |
| PES Schedule A | $\sigma_{\max} = 1 \quad y = 1 \quad P = 25 \quad K_p = 40$ |
| PES Schedule B | $\sigma_{\max} = 1 \quad y = 0.5 \quad P = 25 \quad K_p = 40$ |
| PES Schedule C | $\sigma_{\max} = 1 \quad y = 0.1 \quad P = 25 \quad K_p = 40$ |

*Table 8.* Experimental details for the numerical validation presented in Appendix H.5.

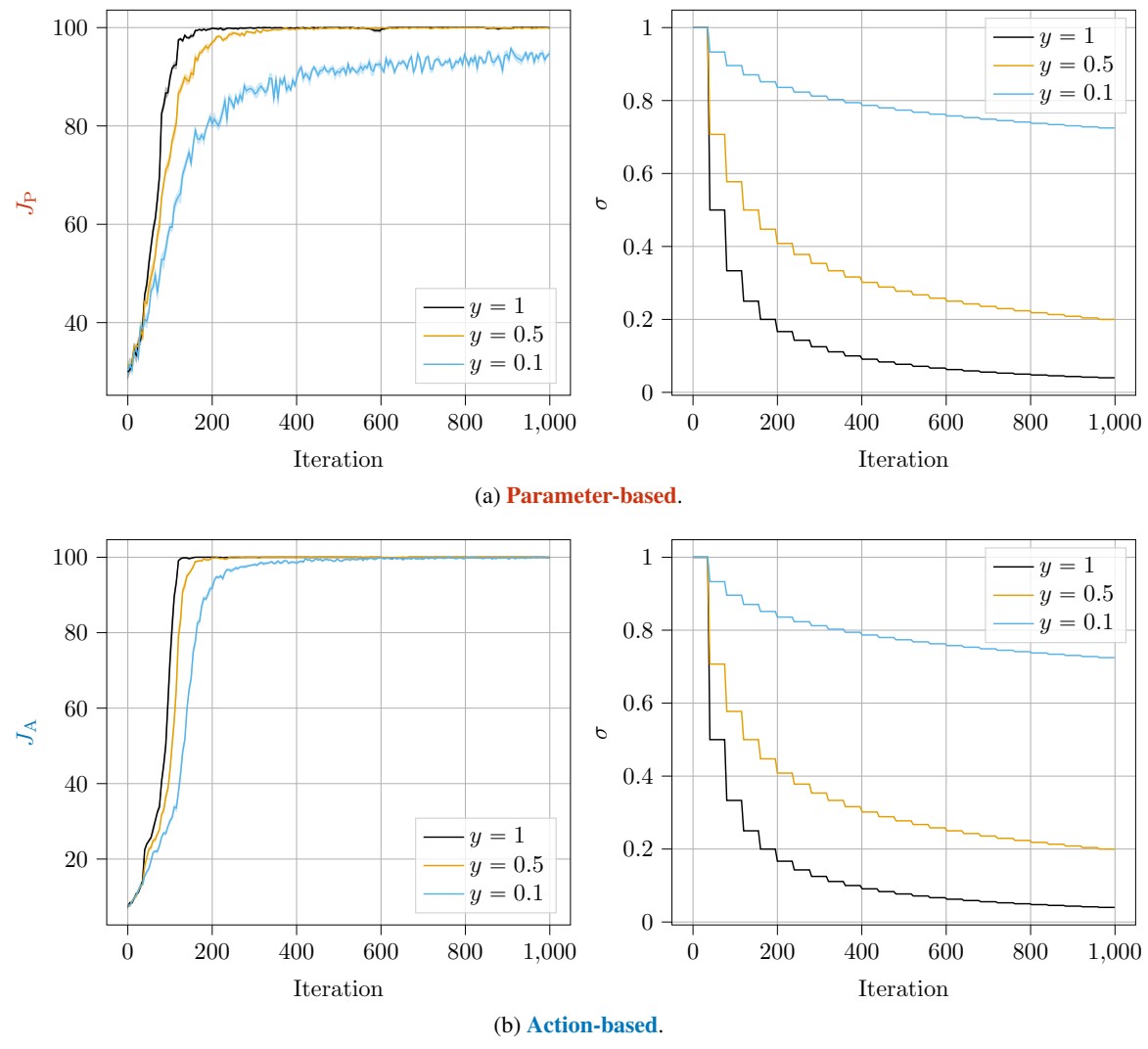

(a) **Parameter-based**.

(b) **Action-based**.

*Figure 6.* $J_\dagger$ and $\sigma$ behavior for **PB** and **AB** in *InvertedPendulum-v5* (5 runs, mean $\pm 95\%$ C.I.).

### H.6. `PES` Sensitivity Analysis: $K_p$ and $P$ Parameters

In this section, we conduct an experiment on the sensitivity of `PES` to the $K_p$ and $P$ parameters. We run `PES` in the *InvertedPendulum-v5* environment with a horizon $T = 100$. The `PES` instances employ a schedule with $\sigma_{max} = 1$ and $y = 1$. Learning rates are managed by Adam and initialized at $\zeta = 0.01$. We recall that the parameters $K_p$ and $P$ control the total amount of iterations done by the algorithm. Specifically, $K_p$ specifies the number of iterations in which PG subroutines are run with a fixed value of stochasticity $\sigma$. Such parameters, together with the selection of $y$ and $\sigma_{max}$, regulate the smoothness of the schedule and the final value for the $\sigma$. In these experiments, we employed three schedules, differing in the $(K_p, P)$ values: (A) has $(40, 25)$, (B) has $(10, 100)$, finally (C) has $(1, 1000)$. The details for this experimental setting are summarized in Table 9.

Figures 7a and 7b show that, in both **AB** and **PB** exploration domains, it is possible to see that `PES`-C is the one showing the poorest convergence speed. In particular, such a schedule presents an unstable behavior at convergence. The best performing one i `PES`-A, which mimics the behavior prescribed by theory. As also observed in the experiments conducted in Section 9, Appendices H.3 and H.4, a smaller value for $K_p$ should be paired with a smoother schedule (i.e., smaller $y$). Indeed, by frequently reducing the stochasticity, it has to be reduced smoothly, since `PES` has not the opportunity to find a good parameterization for every fixed $\sigma$.

| AB and PB Configurations for `PES` Sensitivity Study on $K_p$ and $P$ | |
|---|---|
| Environment | *InvertedPendulum-v5* (Todorov et al., 2012) |
| Horizon | $T = 100$ |
| Dimensions | $d_{\mathcal{A}} = 1 \quad d_{\mathcal{S}} = 4 \quad d_{\Theta} = 4$ |
| (Hyper)policy | Linear Gaussian |
| Learning rates (Adam, Kingma & Ba 2014) | 0.01 |
| Batch size | $N = 100$ |
| `PES` Schedule A | $\sigma_{\max} = 1 \quad y = 1 \quad P = 25 \quad K_p = 40$ |
| `PES` Schedule B | $\sigma_{\max} = 1 \quad y = 1 \quad P = 10 \quad K_p = 100$ |
| `PES` Schedule C | $\sigma_{\max} = 1 \quad y = 1 \quad P = 1000 \quad K_p = 1$ |

*Table 9.* Experimental details for the numerical validation presented in Appendix H.6.

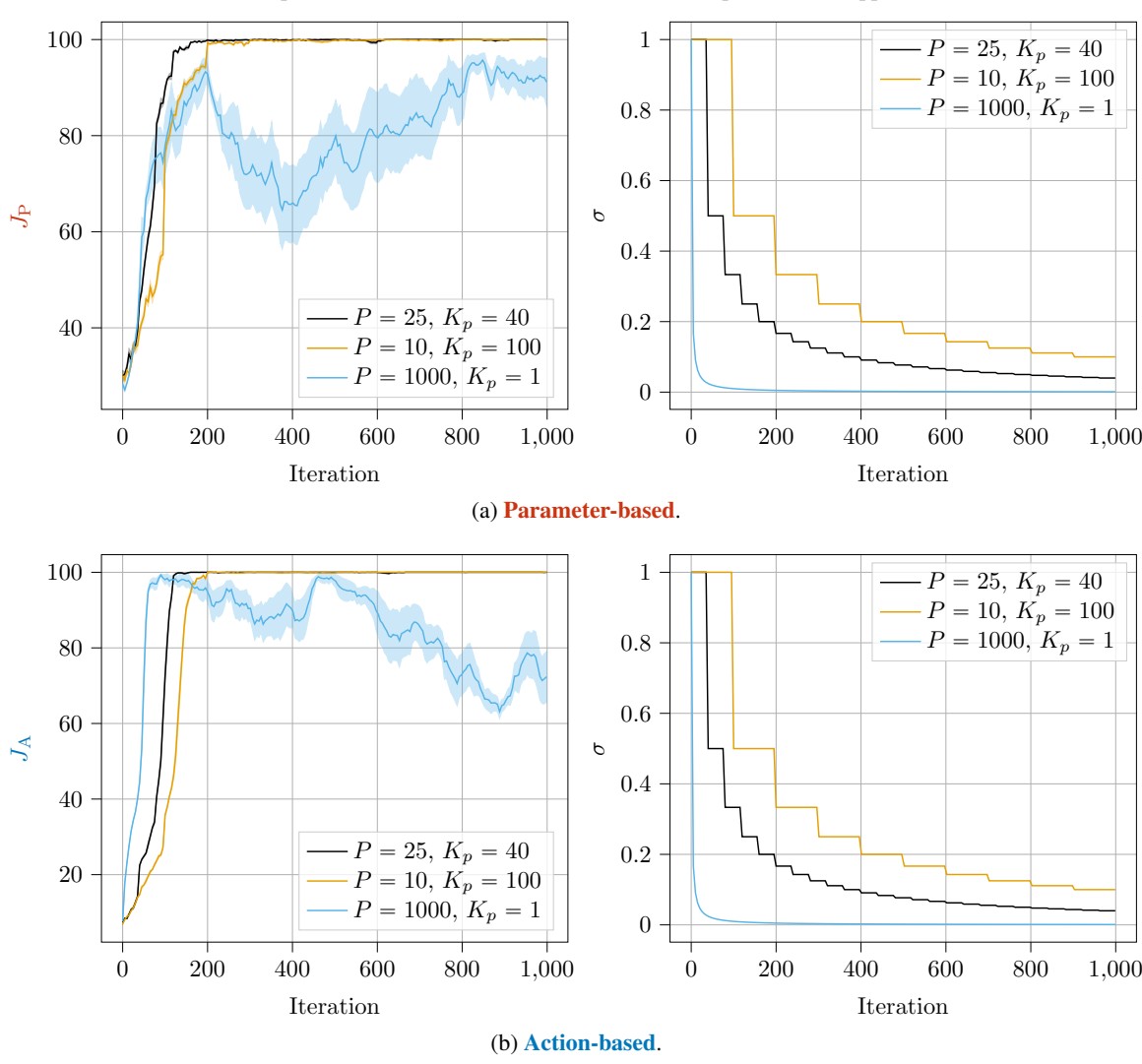

(a) **Parameter-based**.

(b) **Action-based**.

*Figure 7.* $J_\dagger$ and $\sigma$ behavior for **PB** and **AB** in *InvertedPendulum-v5* (5 runs, mean $\pm 95\%$ C.I.).

### H.7. Computational Resources

All the experiments were run on a 2019 16-inches MacBook Pro. The machine was equipped as follows:

| CPU | RAM | GPU |
|-----|-----|-----|
| Intel Core i7 (6 cores, 2.6 GHz) | 16 GB 2667 MHz DDR4 | Intel UHD Graphics 630 1536 MB |

In particular, $N = 100$ trajectories of the MuJoCo environments with $T = 100$ scored $\approx 2$ iterations per second. All the performances are to be considered with a parallelization over 10 CPU cores.

