# OpenReview forum: "Convergence Analysis of Policy Gradient Methods with Dynamic Stochasticity"
_ICML.cc/2025/Conference — ICML 2025 poster_

### Official Review · Reviewer_TELj · 2025-03-12

**Overall Recommendation:** 3

**Summary:**

This paper introduces PES, a phase-based policy gradient algorithm for optimizing (hyper-)policies. Throughout the phases, the stochasticity required for exploration is gradually reduced. However, within a single phase, the stochasticity level remains constant. This allows existing convergence analyses for constant stochasticity to be leveraged to establish convergence within each phase of the PES algorithm. Subsequently, phase-wise convergence to the optimal deterministic policy is proven.
Furthermore, the widely used SL-PG algorithm is examined. In this approach, the stochasticity level is introduced as an additional parameter in the policy gradient algorithm and is optimized jointly with the deterministic policy. Under stronger assumptions, convergence is also demonstrated, but to the optimal stochastic policy. Both algorithms are evaluated in different simulation studies and compared to algorithms with static stochasticity.

## update after rebuttal: Thanks for the answers to clarify some questions.

**Claims And Evidence:**

All Theorems, Lemmas and Corollaries are proven in the appendix. I could not find unclear claims or statements in the text.

**Essential References Not Discussed:**

None. But parts of the paper are very close (also in words) to Montenegro et.al. 2024.

**Experimental Designs Or Analyses:**

I did not implement the experiments to check the correctness. But the design seems fine to me.

**Methods And Evaluation Criteria:**

In theoretical analyses of PG algorithms, a static stochasticity level has almost always been considered. PES is one of the first approaches where stochasticity is gradually reduced, reflecting a common practice in real-world applications. Additionally, SL-PG investigates an algorithm in which the stochasticity is trained alongside the policy and automatically adapted. In my view, this represents a practically relevant algorithm whose convergence rates had not been analyzed before. Therefore, both methods appear well-founded.
The algorithms were evaluated in various examples in Appendix H, which can be considered standard for comparing PG-based methods.

**Other Comments Or Suggestions:**

My judgement is in the middle between weak accept and weak reject. The article is interesting but a pretty close continuation of Montenegra et al. I rated weak accept but note that it is a rather weak weak accept.

**Other Strengths And Weaknesses:**

Strengths:
* The authors analyze two different versions of PG methods with a non-static stochasticity setting
* They verify the performance of the theoretically analyzed algorithms in various applications

Weaknesses:
* The authors do not present examples of policy parametrizations which verify the assumptions in Sec. 4 or Sec. 7.
* The proofs are primarily based on already known methods and results. No new proof strategy is evident.

**Questions For Authors:**

* Could you provide some examples of common policy parametrizations such that the assumptions in Sec. 4 and Sec. 7 are verified?
* Could you compare your convergence results with the known rates for constant stochasticity?
* Would applying momentum-based methods, as in Fatkhullin et al. (2023), also lead to a faster convergence rate in your case?

**Relation To Broader Scientific Literature:**

No direct comparisons is made regarding the convergence rate with previously analyzed algorithms that use static stochasticity. The (upper bound) convergence rates of PES with O(\epsilon{-5}) and SL-PG with O(\epsilon{-3}) are not as favorable as, for example, O(\epsilon^{-2}) in Fatkhullin et.al. 2023 using HARPG.

The authors should address this aspect in more detail in the main part of the paper.

**Theoretical Claims:**

The authors include the case $T=\infty$ when introducing the setting in Sec. 2. However, the convergence results in Sec. 7 sometimes rely on the constant $T$. The authors should explain how this can be possible or delete the possibility of $T=\infty$. Claims in Sec. 4 are all based on Montenegro et.al. 2024, I did not check the correctness in this reference. I checked the Claims in Sec. 5, 6 and 7. They seem solid.

---

> ### Author Rebuttal · Authors · 2025-04-01
>
> We thank the Reviewer for reviewing our work and for recognizing that the proposed algorithms are practically relevant, have not been analyzed before, and appear theoretically well-founded. Below, we address the reviewer’s concerns.
>
> ### 1. On the dependence on $T$
> For the AB results, the correct quantity that should appear is the **effective horizon $\overline{T} = (1-\gamma^T)/(1-\gamma)$**. We will clarify this point and correct the notation accordingly.
>
> ### 2. Theoretical comparison with static stochasticity PGs
> We provide a detailed comparison on this in **Appendix A**, which we plan to move and expand in the main paper, leveraging the additional page.
> The best known rate is achieved by HARPG [2] , a momentum-aided PG method, converging to the **optimal stochastic policy** (AB only) with $\tilde{\mathcal{O}}(\epsilon^{-2})$. However, HARPG differs from vanilla PG due to a Hessian correction step and performs poorly (Fig. 1 of [2]).
> Our focus is on **vanilla PG methods**, thus a comparison with [1,3] is more appropriate. Both provide convergence to the **optimal stochastic (hyper)policy** with rate $\tilde{\mathcal{O}}(\epsilon^{-3} \sigma^{-2})$, and only the latter considers the PB setting.
> That being said, **SL-PG** achieves a rate $\tilde{\mathcal{O}}(\epsilon^{-3} \sigma_{\min}^{-2})$, where $\sigma_{\min}$ is the minimum value for $\sigma$ ensured by its parameterization.
> In contrast, **PES** is designed to guarantee convergence to the **optimal deterministic policy**, with a rate $\tilde{\mathcal{O}}(\epsilon^{-5})$, matching [3] but without requiring a fixed $\sigma = \epsilon$.
>
> ### 3. Differences with [3] and technical novelty
> While our analysis builds upon the framework of [3], we introduce several **non-trivial contributions** that go beyond their results. For a summary on the key contributions and the differences with [3], please refer to the answer to Reviewer UaJQ. Here we focus on the technical novelty which is mainly represented by:
> 1. Theorem 5.1: there we **quantify the loss incurred in terms of performance index $J$ when varying $\sigma$.** This represents the first result of this kind and it highlights the need of employing an additive noise as presented in Section 2 which maintains the same distribution over the training.
> 2. Theorem 6.1: there we assess the **convergence rate of PES**, leveraging the results of [3] for each phase, but introducing a deterministic schedule leading to a final $\sigma$ allowing for a safe deterministic deployment.
> 3. Lemmas G.1--G.12, 7.1, 7.2: there we characterize **crucial quantities for proving Theorem 7.3 (SL-PG convergence)** which relies on standard procedures for convergence study. However, the preliminary results, which we believe are of independent interest, help in better understanding the conditions under which SL-PG converges, especially concerning the kind of $\sigma$ parameterization to be employed.
> 4. Theorems G.14 and G.15: there we study **whether and under which conditions WGD can be inherited by the SL-PG objective**, additionally deriving its sample complexity under WGD inheritance.
>
> In conclusion, we believe our contribution is *not a mere simple extension, providing for the first time a formal foundation for the common practice of dynamically adjusting the stochasticity in PG methods*.
>
> ### 4. Examples of parameterizations
> All the assumptions solely regarding the policy parameterization and the AB and PB noise are met by linear parameterized deterministic policies $\mu_{\theta}(s) = \theta^{\top} s$ with noises sampled from $0$-mean gaussians:  (AB) $\varepsilon \sim \mathcal{N}(0, \sigma I_{d_{\mathcal{A}}})$; (PB) $\varepsilon \sim \mathcal{N}(0, \sigma I_{d_{\theta}})$. Assumptions regarding the $\sigma$ parameterization are met by $\sigma = \sigma_{\min} + 1/(1 + e^{-\xi})$, being $\xi$ the optimization variable. An example of MDP allowing to meet also the remaining MDP-dependent assumptions is LQR, in which the chosen parameterization exhibit WGD (Asm. 4.2, 7.3) [4, Lemma 3].
>
> ### 5. PES and SL-PG over momentum-based PGs
> Recovering the convergence result HARPG (Thr. 3 of [2]), it exhibits a sample complexity of order $\tilde{\mathcal{O}}(\epsilon^{-2})$. In our setting, this translates into a rate $\tilde{\mathcal{O}}(\epsilon^{-2} \sigma^{-2})$, when considering a fixed $\sigma$ (we have the bound to the policy score being $\mathcal{O}(\sigma^{-2})$).
> Thus, **PES over HARPG** converges to the **optimal deterministic policy** with a rate $\tilde{\mathcal{O}}(\epsilon^{-4})$, thus improving our current result at the cost of considering PG subroutines leveraging Hessian estimation. Similarly, **SL-PG over HARPG** converges to the **optimal stochastic policy** with a rate $\tilde{\mathcal{O}}(\epsilon^{-2} \sigma_{\min}^{-2})$, with the same issue. We will add a comment on this.
>
> **References**
> [1] Yuan et al. (2021)
> [2] Fatkhullin et al. (2023)
> [3] Montenegro et al. (2024)
> [4] Fazel et al. (2019)

---

### Official Review · Reviewer_Uuyp · 2025-03-13

**Overall Recommendation:** 2

**Summary:**

The studies effects of exploration on the convergence of the policy gradient in RL. It proposes PES method that reduces the stochasticity with iteration, allowing sufficient exploration in the beginning and the convergence to the the optimal policy in the end. Further, it proposes another SL-PG method, and shows sample complexity of both the methods.

**Claims And Evidence:**

I didn't verify the results, but it seems believable.

**Essential References Not Discussed:**

[1] is not discussed.

**Experimental Designs Or Analyses:**

No.

**Methods And Evaluation Criteria:**

I am not completely satisfied with comparison of the methods with existing litreature.

**Other Comments Or Suggestions:**

Suggestions: SL-PG algorithm is crucial to the paper, a peudocode would be helpful for the reader.

**Other Strengths And Weaknesses:**

Strengths: The paper studies the exploration in policy gradient algorithms which are close to the practice. And provides theoretical bounds.

Weakness: The paper is difficult to read and understand. The advantages of PES algorithm over vanilla actor-critic algorithm (global convergence with sample complexity O(\epsilon^{-4}), see Proposition 1 of [1] ) is not clear. .


[1] @misc{kumar2024improvedsamplecomplexityglobal,
      title={Improved Sample Complexity for Global Convergence of Actor-Critic Algorithms},
      author={Navdeep Kumar and Priyank Agrawal and Giorgia Ramponi and Kfir Yehuda Levy and Shie Mannor},
      year={2024},
      eprint={2410.08868},
      archivePrefix={arXiv},
      primaryClass={cs.LG},
      url={https://arxiv.org/abs/2410.08868},
}

**Questions For Authors:**

Q1: We can obtain global convergence of PG (actor-critic) with an sample complexity of $O(\epsilon^{-4}$ ( local convergence in [1] combined with gradient dominal lemma, see Proposition 1 of [2]). However, it requires mis-match coefficient to be finite, equivalent it assumes sufficient state-space coverage.
I assume, the paper in the review, wants to alleviate this issue with adding exploration in policy?   However, I see, the paper still makes sufficient coverage assumptions? Could the author compare and contrast the results and setting, with [1], what are theoretical benefits of adding exploration, how does this explorations relaxes the assumptions in [1] ?


Q2: The return in RL satisfies gradient domination condition (Agrawal et. al.), then why it is taken as a assumption in the paper?






[1] @misc{chen2024finitetimeanalysissingletimescaleactorcritic,
      title={Finite-time analysis of single-timescale actor-critic},
      author={Xuyang Chen and Lin Zhao},
      year={2024},
      eprint={2210.09921},
      archivePrefix={arXiv},
      primaryClass={cs.LG},
      url={https://arxiv.org/abs/2210.09921},
}


[2] @misc{kumar2024improvedsamplecomplexityglobal,
      title={Improved Sample Complexity for Global Convergence of Actor-Critic Algorithms},
      author={Navdeep Kumar and Priyank Agrawal and Giorgia Ramponi and Kfir Yehuda Levy and Shie Mannor},
      year={2024},
      eprint={2410.08868},
      archivePrefix={arXiv},
      primaryClass={cs.LG},
      url={https://arxiv.org/abs/2410.08868},
}

**Relation To Broader Scientific Literature:**

To my understanding, exploration in RL is a very central topic. The theoretical understanding of effects of exploration on convergence is of great importance to the community.

**Theoretical Claims:**

No.

---

> ### Author Rebuttal · Authors · 2025-04-01
>
> We thank the Reviewer for reviewing our work and for recognizing its practical relevance. Next, we address the reviewer’s questions.
>
> ### 1. Clarification on the paper contribution
> Our paper focuses on **actor-only PG methods** in **continuous state and action spaces**, using (hyper)policies with **dynamically varying stochasticity**, as commonly done in practice. Our goal is to provide a **theoretical foundation** to this practice, which had not been rigorously analyzed before.
> We study **SL-PG**, where $\sigma$ is learned via gradient ascent. Under a gradient domination assumption, we prove convergence to the **optimal stochastic policy** with a rate $\tilde{\mathcal{O}}(\epsilon^{-3} \sigma_{\min}^{-2})$, but SL-PG does not allow to quantify the final $\sigma$, hence no guarantee on the deployed deterministic policy.
> To address this, we propose **PES**, which **deterministically decreases $\sigma$** within phases. This allows control over the final stochasticity and guarantees **last-iterate convergence to the optimal deterministic policy** with rate $\tilde{\mathcal{O}}(\epsilon^{-5})$, matching [1], but without requiring to fix $\sigma = \epsilon$ for the entire training.
>
> ### 2. Lack of comparison with [2]
> We acknowledge the relevance of the contribution of [2], but we believe it lies **outside the scope of our study**, since their **framework is different from the one adopted in our work**.
> Specifically, [2] analyze **finite state and action spaces and employ a softmax policy parameterization**. Their theoretical results establish last-iterate global convergence guarantees for **actor-critic methods with AB exploration**.
> In contrast, our setting considers **continuous state and action spaces and general (hyper)policy parameterization**. We address **both AB and PB explorations for actor-only PG methods**, and, most importantly, we explicitly analyze scenarios in which the **stochasticity $\sigma$ varies during the learning** process, being interested in providing guarantees on the goodness of the **deployed deterministic policy**.
> For these reasons the methodology of [2] is not directly comparable to ours, either in terms of assumptions or scope.
> However, we thank the reviewer for highlighting the lack of discussion on the actor-critic convergence literature, which we plan to include in the paper.
>
> ### 3. Answer to Q1
> As previously said, our setting is different w.r.t. the one of [2,3]. Specifically for [3], we highlight the following:
> 1. We are in **continuous state and action spaces**, [3] consider finite actions;
> 2. We provide **last-iterate global convergence to the optimal stochastic (SL-PG) and deterministic (PES) policies** for actor-only PG methods, [3] provide convergence to the stationary point under stochastic policies for actor-critic methods;
> 3. We consider **both PB and AB explorations**, [3] only AB one;
> 4. We consider a setting in which the **(hyper)policy stochasticity varies** while learning, [3] consider static stochasticity;
> 5. We are interested in **deploying deterministic policies**, [3] stochastic ones;
> 6. For convergence, we need **$J_D$ to be smooth, the variance of the estimators to be finite, and weak gradient domination (WGD) on $J_D$**, [3] assume sufficient exploration ($A$ matrix to be negative definite, e.g., in tabular MDPs this requires to the policy to explore all the state-action pairs), uniform ergodicity of the stationary distribution, and the regularity of the policy.
>
> Our set of assumptions is more general, which is required by the more general setting of our work. Notice that the kind of exploration we propose combined with the WGD do not require to assume sufficient state coverage, allowing to treat continuous spaces and general parameterizations, permitting to ensure last-iterate convergence (stronger w.r.t. the one to stationary points).
>
> ### 4. Why assuming weak gradient domination (WGD)
> In general settings, as the one considered in our work, WGD does not hold and has to be assumed [1,7,8,9]. [4] prove it in **tabular MDPs** with **direct policy parameterization only**. However, WGD can be derived under different sets of assumptions. For instance, [5] show that, in **AB exploration**, **WGD is induced** on $J_A$ when the **policy satisfies the Fisher non-degeneracy condition**, while it remains an open question whether a similar result holds for PB exploration. In our framework, [1] show that **if it holds for** $J_D$, then it is **inherited by both $J_A$ and $J_P$**.
> Similarly, for **SL-PG**, we show in Appendix G.4.3 that **WGD is inherited by $J_A$ and $J_P$**, even when including the additional learned variable $\sigma$, provided it holds for $J_D$ and under additional assumptions.
>
> **References**
> [1] Montenegro et al. (2024)
> [2] Kumar et al. (2024)
> [3] Chen and Zhao (2024)
> [4] Agrawal et al. (2020)
> [5] Liu et al., (2020)
> [6] Mei et al. (2022)
> [7] Yuan et al. (2021)
> [8] Fathkullin et al. (2023)
> [9] Bhandari & Russo (2024b)

---

### Official Review · Reviewer_vEmL · 2025-03-13

**Overall Recommendation:** 4

**Summary:**

This paper provides a global last-iterate convergence analysis for a widely used class of reinforcement learning algorithms, specifically deterministic policy gradient methods with dynamic stochasticity. It considers two common types of dynamic stochasticity: phased exploration scheduling (PES) and stochasticity learning scheduling. The paper establishes global convergence under standard assumptions and offers a thorough discussion of theoretical connections with prior work. Additionally, experimental results are presented to support the analysis.

**Claims And Evidence:**

The claims made in the submission are well supported by evidence.

**Essential References Not Discussed:**

No specificly.

**Experimental Designs Or Analyses:**

Some of the experiments are unclear. Please refer to the questions for more details.

**Methods And Evaluation Criteria:**

Methods And Evaluation Criteria make sense for the problem.

**Other Comments Or Suggestions:**

Please see questions.

**Other Strengths And Weaknesses:**

Strengths:
Broad Scenarios: The paper considers both action-based stochasticity and parameter-based stochasticity for phased-based dymanic and learning-based dynamic.
The theoretical analysis is robust and effectively compared with prior work.

Weaknesses:
Please refer to the questions.

**Questions For Authors:**

Overall, I find this paper to be well-written and solid. However, I have a few questions:

1. How do the experimental results help to illustrate the theoretical findings? While reviewing the results for Swimmer-v5 and InvertedPendulum-v5, the connection between the global convergence rate and the levels of stochasticity is not clear.

2. Given that dynamic stochasticity is a common scenario in practice and of interest to a broader audience, I would appreciate seeing more experimental results from environments with higher dimensions.

3. While the Weak Gradient Domination is addressed as assumptions (4.2) in this paper, I wonder if this WGD can be derived by other assumptions?

**Relation To Broader Scientific Literature:**

This work addresses a common and highly relevant scenario in reinforcement learning, offering valuable insights for both theorists and practitioners. It is particularly pertinent to those utilizing deterministic policy gradient methods with noise-based exploration, such as in DDPG.

**Theoretical Claims:**

I reviewed the overall structure of the theories (though I did not check the proof details line by line) and found that the theory is well-written and sound.

---

> ### Author Rebuttal · Authors · 2025-04-01
>
> We thank the Reviewer for reviewing our work, and for recognizing that we address a highly relevant scenario in RL, providing valuable insights for both theorists and practitioners, and offering a strong theoretical analysis. Below, we address the Reviewer’s concerns.
>
> ### 1. Relation between theoretical results and empirical learning curves
> We are happy to clarify this point. Our aim is to provide, for the first time, a **theoretical foundation to the common practice of reducing the (hyper)policy stochasticity** during the learning process. To this end, we analyze the practice of learning $\sigma$, named in our work as SL-PG, and we propose PES, which decreases deterministically $\sigma$. The former converges to the **optimal stochastic policy** with a rate $\tilde{\mathcal{O}}(\epsilon^{-3} \sigma_{\min}^{-2})$, the latter converges to the **optimal deterministic policy** with a rate $\tilde{\mathcal{O}}(\epsilon^{-5})$, both matching SOTA results employing a fixed stochasticity during the whole learning [1,2,4] whose prescribe, for deterministic convergence, to set $\sigma=\mathcal{O}(\epsilon)$, resulting in an impractical setting. In general, we expect that methods showing a small dominating term in the sample complexity bound to converge fast also in practice. Here, PES and SL-PG **provide the same guarantees of the ones employing a static $\sigma$**. This happens since in the theoretical analysis the dominant term in the variance of the estimator is the one related to the smallest $\sigma$ seen during training, thus providing guarantees as if we use a static $\sigma=\sigma_{\min}$ for the entire learning. However, as seen in our numerical validation, PES and SL-PG **may perform better in practice since they expand the exploration possibilities by employing various $\sigma$ values, without requiring to select one and to keep it for the whole learning.** We conclude by stressing that by employing PES there is the additional advantage of knowing the final $\sigma$ which can be set small as the user wants, guaranteeing a proportional loss in performance when deploying a deterministic policy.
>
> ### 2. Experiments on high-dimensional environments
> We emphasize that the **primary focus** of our work is **theoretical**. Indeed, our goal is to **provide formal foundations for practices** that are already common in the use of **SL-PG** [6,7,8], and to propose **PES** as an alternative when the user aims to **obtain an almost deterministic final policy.**
> This is why our experiments remain close to the theoretical setting, while, regarding the **learning of** $\sigma$, the **practical literature has already adopted this approach**, and most experiments in the deep PG field involve learning the stochasticity.
> That said, for the purpose of this rebuttal, we have also **conducted additional experiments using deep RL** methods to assess the behavior of PES and SL-PG in high-dimensional settings.
> (https://drive.google.com/file/d/1149yLdiUaLudIfGEA8yRZK6FjHmY9fJG/view?usp=sharing)
> As shown in the paper, **PES and SL-PG perform at least as well as their static $\sigma$ counterparts**, without requiring prior tuning. With PES, the final $\sigma$ is predefined, yielding an **almost deterministic policy**. As predicted by theory, parameter-based methods may struggle with large parameter spaces.
>
> ### 3. Weak gradient domination (WGD) and its derivation
> WGD is a **customary assumption** in the PG literature when establishing convergence guarantees [1,2,3,4]. Fundamentally, it enables **last-iterate convergence guarantees** by **characterizing the objective function without requiring concavity** in the parameters and while allowing for the presence of (at most $\beta$-near) local optima.
> In the same setting we adopt, [1] show that WGD is **inherited by both AB and PB objectives**, denoted respectively by $J_A$ and $J_P$, whenever it is **assumed to hold for the deterministic** objective $J_D$.
> Additionally, [5] show that, for AB exploration, **WGD is induced on the objective** $J_A$ when the **stochastic policy satisfies the Fisher non-degeneracy condition**, i.e., when there exists $\lambda > 0$ s.t. $\mathbb{E}[\nabla \log \pi_{\theta}(a|s) \nabla \log \pi_{\theta}^{\top}(a|s)] \succeq \lambda I$ for any $\theta, a, s$. While this result is well established for AB exploration, it **remains an open question** whether WGD **can be similarly induced** on $J_P$ through a specific class of hyperpolicies **in the PB setting**.
> Finally, for **SL-PG**, we show in Appendix G.4.3 how WGD **can be inherited by the stochastic objectives** $J_A$ and $J_P$, even when **including the additional learning variable** associated with $\sigma$, provided it holds for $J_D$, along with supplementary assumptions.
>
> **References**
> [1] Montenegro et al. (2024)
> [2] Yuan et al. (2021)
> [3] Fathkullin et al. (2023)
> [4] Bhandari & Russo (2024b)
> [5] Liu et al. (2020)
> [6] Duan et al. (2016)
> [7] Schulman et al. (2017)
> [8] Tirinzoni et al. (2024)

---

### Official Review · Reviewer_UaJQ · 2025-03-18

**Overall Recommendation:** 3

**Summary:**

The paper focuses on the convergence analysis of policy gradient methods in RL with dynamic stochasticity. It introduces PES, a phase-based algorithm that reduces stochasticity through a deterministic schedule while running policy gradient subroutines with fixed stochasticity in each phase. The paper demonstrates that PES achieves last-iterate convergence to the optimal deterministic policy. Additionally, it analyzes the common practice of jointly learning stochasticity and policy parameters (SL-PG), showing that this approach also ensures last-iterate convergence but to the optimal stochastic policy only, requiring stronger assumptions compared to PES. Experimental results on a toy simulation environment demonstrate that the theoretical claims are indeed satisfied.

**Claims And Evidence:**

The claims regarding convergence of PG algorithms under dynamic stochastic noises are proven thoroughly with theoretical proofs.

**Essential References Not Discussed:**

Distinction between the proposed work and Montenegroetal. (2024) [from which most of the theories are derived] could have been explained better.

**Experimental Designs Or Analyses:**

Experiments are conducted on a toy simulation environment to demonstrate the impact of different number of phases and their duration, and to compare between proposed PES and SL-PG. While these analysis makes sense to demonstrate the theoretical properties, a detailed analysis by comparing the performance with existing deep PG algorithms would have been more appreciated.

**Methods And Evaluation Criteria:**

This is more of a theoretical paper for convergence analysis of PG approaches in RL. Experiments and evaluation are conducted on toy simulation setting just to demonstrate the theoretical properties.

**Other Comments Or Suggestions:**

Majority of the space is allocated towards theorem and analysis. The paper is convoluted with symbols and theorems which is hard to follow in the current format. Some restructuring would be helpful for the readers. For example, adding related work section in main body, and highlighting the high-level takeaways from each theorem.

**Other Strengths And Weaknesses:**

The paper is theoretically strong and solves an important problem of convergence analysis of PG algorithms through a novel phase-based iterative algorithm. It also analyzes common practices of Stochasticity-Learning Policy Gradient (SL-PG). Most importantly, the paper provides a solid theoretical foundation for the use of dynamic stochasticity in policy gradient methods, bridging the gap between theory and practice. Having said that, there are a few concerns in terms of practical application:
1. Complexity of PES: While PES provides strong convergence guarantees, its phase-based approach and deterministic schedule might be complex to implement and tune in practice. The need to carefully select parameters such as the number of phases and the learning rate schedule adds to this complexity. How to select the optimal parameter configuration needs to be explained in more details.
2. Limited Practical Validation: The paper primarily focuses on theoretical analysis and does not provide extensive practical validation of the proposed methods. More empirical results and experiments in real-world scenarios would strengthen the paper's contributions. Furthermore, benchmarking the performance against SOTA deep PG methods would be a nice addition.

**Questions For Authors:**

1. Why PES and SL-PG are not compared with SOTA deep PG algorithms?
2. How to identify optimal number of phases and their length in a practical setting?
2. What is the main difference between the proposed analysis and the work of Montenegroetal. (2024)?

**Relation To Broader Scientific Literature:**

While policy gradient in RL is an important and broader topic, the convergence analysis under white noise hyper-policies would of interest to a smaller portion.

**Theoretical Claims:**

The paper is theoretically very strong. Although majority of the ideas are derived from Montenegroetal. (2024), the authors have proved the convergence properties step by step with several theorems.

---

> ### Author Rebuttal · Authors · 2025-04-01
>
> We thank the Reviewer for reviewing our work and for recognizing the strengths of our theoretical analysis, and the relevance of addressing an open problem in the convergence of PG methods with dynamic stochasticity. Below, we address the Reviewer’s concerns.
>
> ### 1. Comparison against SOTA Deep RL techniques
> We wish to emphasize that the **exploration schedules** adopted in PES (i.e., a deterministic decay of $\sigma$) and SL-PG (i.e., learning $\sigma$ via gradient ascent) can, in principle, be **applied to any PG method** (also actor-critic).
> Thus, we can consider deep PG methods, as PPO, to be incorporated with the dynamic stochasticity methods of PES and SL-PG.
> We stress that the convergence analyses of PES and SL-PG over actor-critic methods require different analyses.
> Nonetheless, we now present an **empirical comparison of three variants of PPO**: $(i)$ the version with a fixed $\sigma$; $(ii)$ a PES-inspired version deterministically decreasing $\sigma$; $(iii)$ an SL-PG-inspired version where $\sigma$ is learned.
> (https://drive.google.com/file/d/1149yLdiUaLudIfGEA8yRZK6FjHmY9fJG/view?usp=sharing) As shown in the paper, **PES and SL-PG perform at least as well as their static $\sigma$ counterparts**, without requiring prior tuning. With PES, the final $\sigma$ is predefined, yielding an **almost deterministic policy**. As predicted by theory, parameter-based methods may struggle with large parameter spaces.
>
> ### 2. On the selection of schedule's parameters
> For PES, the selection of the schedule's parameters is crucial. In general, following the theory, Theorems 4.4 and 6.1 suggest that longer phases allow to better converge to the optimum of a specific $\sigma$, while a higher number of phases $P$ allows a smooth transition among objective functions ($\sigma$-dependent). When designing the actual schedule, considering equal length phases, the user may decide the total number of iterations $K$ and the desired final $\sigma$. Then, selecting the smoothness of the schedule $y$, $P$ is identified by inverting $\sigma_P = \sigma_{\max}P^{-y}$. If $y$ is large, then the schedule will not be smooth and $P$ will be small, while $K_p$ large. As $y\to0$, then the schedule will be smoother and $P \to K$, leading to a continuous schedule ($K_p=1$) which often works properly (Fig. 1a, 4a, 4b, 5a, 5b). SL-PG may overcome these by adapting $\sigma$ automatically, avoiding manual tuning. However, since the final $\sigma$ is not controlled, **no guarantees can be provided on the resulting deterministic policy** after switching off the noise.
>
> ### 3. Differences with [1]
> Our paper **builds upon the foundation established by [1]**, which consider fixed $\sigma$ throughout their analysis. [1] show that the convergence to the optimal deterministic policy is achieved with $\tilde{\mathcal{O}}(\epsilon^{-5})$, by setting $\sigma = \mathcal{O}(\epsilon)$. This is impractical since it means to keep $\sigma$ fixed to a small value for the whole training, leading to slow convergence in practice, and requiring to run multiple trainings to tune $\sigma$. In contrast, for the first time in theory, we consider a setting in which $\sigma$ is variable, studying methods eliminating the issues of [1] and managing to provide the same convergence guarantees for PES and similar for SL-PG.
> We explicitly **include the results of [1]** (Sec.4, Apx. D), as their framework introduces key concepts necessary to fully appreciate our contribution. Nevertheless, **all additional results that explicitly model and analyze the role of varying stochasticity are novel**:
> 1. The design of PES (Sec. 3);
> 2. The characterization of the loss incurred in the objectives $J_A$ and $J_P$ when the stochasticity level is changed (Thr. 5.1);
> 3. The derivation of the sample complexity required by PES to guarantee last-iterate global convergence to the optimal deterministic policy (Thr. 6.1);
> 4. The study on the conditions and the sample complexity under which SL-PG ensure last-iterate global convergence to the optimal stochastic policy (Lem. G.1--G.12, 7.1, 7.2, Thr. 7.3);
> 5. The study on whether the weak gradient domination (WGD) is inherited by $J_A$ and $J_P$ considering also the additional learning variable related to the $\sigma$ (Thr. G.14) and the resulting sample complexity of SL-PG under WGD inheritance (Thr. G.15).
>
> ### 4. Relevance of white-noise-based exploration
> White-noise (hyper)policies encompass a large variety of controllers. Indeed **most stochastic (hyper)policies used in continuous control**, such as Gaussian with fixed variance, **fall in this class** [2,3,4].
>
> ### 5. On the clarity of the manuscript
> We agree that moving the **related works section to the main text** will improve clarity, and we plan to do so in the final version. We also aim to better highlight the core ideas of each result to enhance readability.
>
> **References**
> [1] Montenegro et al. (2024)
> [2] Duan et al. (2016)
> [3] Schulman et al. (2017)
> [4] Tirinzoni et al. (2024)

---

### Decision · Program_Chairs · 2025-05-01

**Decision:**

Accept (poster)

**Comment:**

This paper analyzes the convergence of the deterministic policy gradient method when exploration noise is systematically decreased across multiple phases. The reviewers assessed the paper as a valuable contribution to the understanding of related algorithms, and I concur with their evaluation.